# A NON-LORENTZIAN PRIMER

ERIC A. BERGSHOEFF, JOSÉ M. FIGUEROA-O'FARRILL, AND JOAQUIM GOMIS

ABSTRACT. We review both the kinematics and dynamics of non-lorentzian theories and their associated geometries. First, we introduce non-lorentzian kinematical spacetimes and their symmetry algebras. Next, we construct actions describing the particle dynamics in some of these kinematical spaces using the method of nonlinear realisations. We explain the relation with the coadjoint orbit method. We continue discussing three types of non-lorentzian gravity theories: Galilei gravity, Newton-Cartan gravity and Carroll gravity. Introducing matter, we discuss electric and magnetic non-lorentzian field theories for three different spins: spin-0, spin-1/2 and spin-1, as limits of relativistic theories.

## CONTENTS

EMPG-22-08.

## 1. INTRODUCTION

The recent interest in non-lorentzian theories and their associated geometries is, among other things, due to the following developments:

(i) *non-relativistic holography* [1] [2], which has applications in condensed matter physics. In particular it allows to describe non-relativistic strongly coupled field theories in terms of dual non-relativistic gravity theories;

(ii) *flat space holography*, see for example [3–5] (general), [6, 7] (BMS symmetries) and [8, 9] (Carroll symmetries), which allows us to understand soft theorems [10, 11] and the symmetries of black-hole horizons [12];

(iii) *non-relativistic string theories* [13, 14], carrollian string theories [15] and tensionless string theories [16–18] as corners of the moduli space of solvable string theories (see the review [19]);

(iv) *post-Newtonian corrections* [20–24] in the experimental and theoretical investigations of gravitational waves [25]; and

(v) *fractons* [26] [27] which are condensed matter configurations with restricted mobility which display infrared/ultraviolet mixing with subsystem symmetries (see the review [28]).

In this review we introduce some of the basic concepts and tools to study these theories. We first introduce kinematical Lie algebras their associated homogeneous spacetimes. Some of these Lie algebras arise as contractions of the isometry algebras of (anti) de Sitter spacetimes, following the pioneering work of Bacry and Lévy-Leblond [29], but by far not all of them are obtained in this way. We restrict ourselves to kinematical Lie algebras which preserve space isotropy and hence the kinematical spacetimes we consider are also spatially isotropic. They are adequate to describe particle dynamics. In particular this means that we are not considering so-called p-brane kinematical Lie algebras and their associated spacetimes in which to describe non-lorentzian p-brane actions. We refer the interested reader to [30–33].

We present a classification of (spatially isotropic) kinematical and aristotelian[1] Lie algebras in generic dimension [34, 35]. Generic means that they exist in all dimensions. There are additional kinematical Lie algebras in two, three and four spacetime dimensions, to which we refer the reader to the classic work of Bacry and Nuyts [36] (reviewed in [34]) for dimension $3 + 1$, [37] for dimension $2 + 1$ and the classic Bianchi classification of three-dimensional Lie algebras [38, 39] for $1 + 1$. After a brief review of homogeneous geometry and the infinitesimal description of homogeneous spaces in terms of Klein pairs, we present the classification of spatially isotropic homogeneous spacetimes of kinematical Lie groups. Again we list those which exist in generic dimension, which here it means we are omitting some $1 + 1$ and $2 + 1$ dimensional spacetimes, which can be found in [40, 41].

In the study of particle dynamics on homogeneous kinematical spacetimes, one meets homogeneous spaces of the kinematical groups other that the actual spacetimes: namely, coadjoint orbits and their associated evolution spaces. We review the rôle played by these homogeneous spaces in the construction of lagrangians describing particle dynamics on the homogeneous spacetimes.

We review the method of nonlinear realisations and coadjoint orbits in the construction of particle lagrangians and apply it in several examples, among them the well-known relativistic massive and massless particles. In the non-relativistic case we construct the harmonic oscillator as a nonlinear realisation of a centrally extended Newton–Hooke group. We also consider the massless galilean particle introduced by Souriau [42, 43].

In the case of Caroll due to its causal structure we consider a massive timelike and tachyonic particles. Using the conformal algebra in one dimension we derive the action of conformal mechanics of Alfaro, Fubini and Furlan [44] and the Schwarzian action [45–47].

The analogues of contractions for Lie algebras in dynamical systems are limits of actions, such as non-relativistic, carrollian and flat limits. The actions constructed using the nonlinear realisation method are also obtained as nonrelativistic limits of the relativistic actions. In general these limits produce terms that are divergent, these unwanted terms can be eliminated by a suitable coupling of a relativistic dynamical system to a gauge field in the case of a particle or a B-field in the case of a string [13] [31]. In some of these cases the divergent terms are total derivatives. One can also eliminate divergences by a redefinition of the parameters appearing in the first term of the expansion see [48] [49].

As for the case of non-lorentzian particles, we continue discussing different aspects of non-lorentzian gravity theories. We first review how general relativity can be described by a gauging procedure applied to the Poincare algebra. Next, we extend the discussion by applying the same gauging procedure to the following non-lorentzian algebras: Galilei, Bargmann and Carroll. These gaugings lead to Galilei gravity,

---

[1]These are kinematical Lie algebras without boosts.

Newton-Cartan gravity [2] and Carroll gravity, respectively. We show how these same gravity theories can be obtained by taking particular (Galilean, Bargmann and Carroll) limits of general relativity. For recent work on electric and magnetic theories of gravity see [51–54].

Besides taking non-relativistic limits there are two more ways to obtain non-relativistic theories that we do not explore any further in this review. First, instead of taking the Inönü–Wigner contraction of a Lie algebra, one may also consider a Lie algebra expansion [55–58] where the number of generators corresponding to the nonrelativistic symmetries is increased. Second, one may obtain a non-relativistic theory by the null reduction of a relativistic theory in one spatial dimension higher, see, e.g., [59, 60]. This null reduction is based upon the fact that the Bargmann algebra allows a null-embedding into a Poincare algebra in one spatial dimension higher.

Having discussed non-lorentzian gravity, we continue introducing matter and discussing non-lorentzian field theories. We will do this for a for a complex and real massive spin-0 particle, a massive spin-1/2 particle and a massless spin-1 particle. In particular, for spin-0, we will discuss the Galilei, Bargmann and Carroll limits while for spin-1/2 and spin-1 we will only discuss the Bargmann limit.

## 2. MOTIVATION

Let us motivate our discussion of kinematical symmetries and their spacetimes by contrasting two classical models of the universe: the Galilei spacetime of newtonian mechanics and Minkowski spacetime of special relativity. As we will see, both spacetimes are described by a four-dimensional affine space, homogeneous under the action of a kinematical Lie group; that is, a transformation group consisting of rotations, boosts and translations in both space and time. We will contrast the invariant structures of the two spacetimes: a clock and a ruler in the Galilei spacetime and a proper distance in Minkowski spacetime. The latter defines a lorentzian metric and the former, as we will see, a (weak) Newton–Cartan structure. We will also contrast their Lie algebras of symmetries: the finite-dimensional Lie algebra of isometries in Minkowski spacetime and the infinite-dimensional Coriolis algebra in the Galilei spacetime.

2.1. **Affine space.** Let $\mathbb{A}^4$ denote the four-dimensional affine space. It is modelled on the vector space $\mathbb{R}^4$ in the sense that given any two points $a, b \in \mathbb{A}^4$ there exists a unique translation $v \in \mathbb{R}^4$ such that $b = a + v$. We often refer to $v$ as $b - a$ and identify translations with differences of points. We will use an explicit model for $\mathbb{A}^4$ as the affine hyperplane in $\mathbb{R}^5$ consisting of points $(x^1, x^2, x^3, x^4, x^5 = 1) \in \mathbb{R}^5$, but we should emphasise that the fifth dimension is an auxiliary construct and has no physical meaning. One cannot add points in $\mathbb{A}^4$ (their last entry would not equal 1), but one can add differences, since those lie in the hyperplane $x^5 = 0$. In this model, the group $\mathrm{Aff}(4, \mathbb{R})$ of affine transformations of $\mathbb{A}^4$ is the subgroup of $\mathrm{GL}(5, \mathbb{R})$ which preserves the hyperplane $x^5 = 1$. It consists of matrices of the form

$$\begin{pmatrix} L & v \\ 0 & 1 \end{pmatrix} \tag{2.1}$$

where $v \in \mathbb{R}^4$ and $L \in \mathrm{GL}(4, \mathbb{R})$. We will see that the relativity groups of the Galilei and Minkowski spacetimes are subgroups of the affine group containing all the translations $v \in \mathbb{R}^4$ but with a restricted subgroup of linear transformations consisting of rotations and boosts.

It follows from matrix multiplication that the affine group is the semidirect product $\mathrm{GL}(4, \mathbb{R}) \ltimes \mathbb{R}^4$, with $\mathrm{GL}(4, \mathbb{R})$ acting on $\mathbb{R}^4$ by matrix multiplication. Multiplying $(x, 1) \in \mathbb{R}^5$ by the matrix in equation (2.1) gives $(Lx + v, 1)$, which is the effect of an affine transformation. Both the Galilei and Minkowski spacetimes are described by $\mathbb{A}^4$, only that their invariant structures differ. Points in $\mathbb{A}^4$ are called **(spacetime) events**.

2.2. **Galilei spacetime.** The following description of Galilean spacetime is essentially due to Weyl [61]. Galilei spacetime is defined by $\mathbb{A}^4$ together with two invariant notions:

- a **clock** $\tau : \mathbb{R}^4 \to \mathbb{R}$, sending $b - a \mapsto \tau(b - a)$ and measuring the time interval between two events $a, b \in \mathbb{A}^4$. If $a = (x, 1)$ and $b = (y, 1)$, then $\tau(b - a) = y^4 - x^4$. Two events $a, b \in \mathbb{A}^4$ are said to be **simultaneous** if $\tau(b - a) = 0$. In other words, simultaneous events are related by translations in the kernel of $\tau$. If we fix an event $a$, the set of events simultaneous to $a$ defines a three-dimensional affine subspace

$$a + \ker \tau = \{a + v \mid \tau(v) = 0\} \tag{2.2}$$

of $\mathbb{A}^4$. As the notation suggests, it is a coset of the subgroup $\ker \tau$ of the translation group $\mathbb{R}^4$. The quotient $\mathbb{A}^4 / \ker \tau$ is an affine line $\mathbb{A}^1$, so that the clock gives a fibration $\pi : \mathbb{A}^4 \to \mathbb{A}^1$ whose

---

[2]Newton-Cartan gravity in the context of non-relativistic holography was studied in [50].

fibre at $\pi(\mathfrak{a})$ consists of all those events simultaneous to $\mathfrak{a}$, which constitute an affine hypersurface $\mathbb{A}^3_\mathfrak{a}$ of $\mathbb{A}^4$. This is illustrated in Figure 1.

- a **ruler** $\lambda : \ker \tau \to \mathbb{R}$, sending $\mathfrak{b} - \mathfrak{a} \mapsto \lambda(\mathfrak{b} - \mathfrak{a})$ and measuring the euclidean distance between simultaneous events. Explicitly, if $\mathfrak{a} = (\mathbf{x}, x^4, 1)$ and $\mathfrak{b} = (\mathbf{y}, y^4, 1)$ with $\mathbf{x}, \mathbf{y} \in \mathbb{R}^3$ and $x^4 = y^4$ are simultaneous events, then $\lambda(\mathfrak{b} - \mathfrak{a}) = \|\mathbf{y} - \mathbf{x}\| = \sqrt{(\mathbf{y} - \mathbf{x}) \cdot (\mathbf{y} - \mathbf{x})}$, which is the euclidean distance between $\mathbf{x}$ and $\mathbf{y}$.

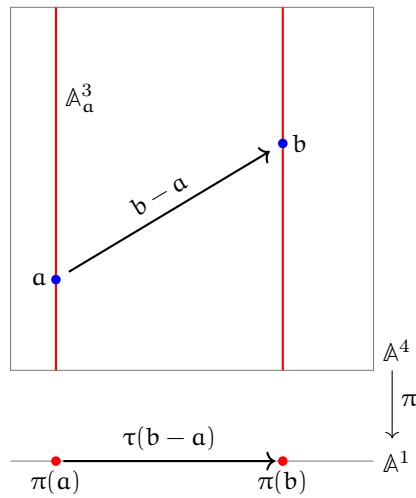

FIGURE 1. The clock fibration $\pi : \mathbb{A}^4 \to \mathbb{A}^1$

The kinematical group of Galilei spacetime is called the **Galilei group** and it consists of those affine transformations of $\mathbb{A}^4$ which preserve the clock and the ruler. It embeds in $\mathrm{GL}(5, \mathbb{R})$ as those matrices of the form

$$\begin{pmatrix} R & \boldsymbol{v} & \mathfrak{a} \\ 0 & 1 & s \\ 0 & 0 & 1 \end{pmatrix}, \tag{2.3}$$

where $R \in \mathrm{O}(3)$, $\mathfrak{a}, \boldsymbol{v} \in \mathbb{R}^3$ and $s \in \mathbb{R}$. This matrix is of the form (2.1), but where the general linear transformation $L$ is of the form $\begin{pmatrix} R & \boldsymbol{v} \\ 0 & 1 \end{pmatrix}$.

The action of the matrix in equation (2.3) on an event $(\mathbf{x}, t, 1)$ gives the event $(R\mathbf{x} + t\boldsymbol{v} + \mathfrak{a}, t + s, 1)$ which we interpret as the composition of an orthogonal transformation $\mathbf{x} \mapsto R\mathbf{x}$, a **Galilei boost** $\mathbf{x} \mapsto \mathbf{x} + t\boldsymbol{v}$, a spatial translation $\mathbf{x} \mapsto \mathbf{x} + \mathfrak{a}$ and a temporal translation $t \mapsto t + s$:

$$\begin{pmatrix} R & \boldsymbol{v} & \mathfrak{a} \\ 0 & 1 & s \\ 0 & 0 & 1 \end{pmatrix} = \begin{pmatrix} I & 0 & 0 \\ 0 & 1 & s \\ 0 & 0 & 1 \end{pmatrix} \begin{pmatrix} I & 0 & \mathfrak{a} \\ 0 & 1 & 0 \\ 0 & 0 & 1 \end{pmatrix} \begin{pmatrix} I & \boldsymbol{v} & 0 \\ 0 & 1 & 0 \\ 0 & 0 & 1 \end{pmatrix} \begin{pmatrix} R & 0 & 0 \\ 0 & 1 & 0 \\ 0 & 0 & 1 \end{pmatrix}. \tag{2.4}$$

Its Lie algebra is the **Galilei algebra**, which is isomorphic to the subalgebra of $\mathfrak{gl}(5, \mathbb{R})$ consisting of matrices of the form

$$\begin{pmatrix} A & \boldsymbol{v} & \mathfrak{a} \\ 0 & 0 & s \\ 0 & 0 & 0 \end{pmatrix}, \tag{2.5}$$

where $A \in \mathfrak{so}(3)$, $\boldsymbol{v}, \mathfrak{a} \in \mathbb{R}^3$ and $s \in \mathbb{R}$. We may introduce a basis $L_{ab} = -L_{ba}, B_a, P_a, H$ by

$$\begin{pmatrix} A & \boldsymbol{v} & \mathfrak{a} \\ 0 & 0 & s \\ 0 & 0 & 0 \end{pmatrix} = \tfrac{1}{2} A^{ab} L_{ab} + v^a B_a + a^a P_a + s H. \tag{2.6}$$

We can easily work out the Lie brackets of the Galilei algebra in this basis. The nonzero brackets are given by

$$\begin{aligned} [L_{ab}, L_{cd}] &= \delta_{bc} L_{ad} - \delta_{ac} L_{bd} - \delta_{bd} L_{ac} + \delta_{bd} L_{ac} \\ [L_{ab}, B_b] &= \delta_{bc} B_a - \delta_{ac} B_b \\ [L_{ab}, P_b] &= \delta_{bc} P_a - \delta_{ac} P_b \\ [B_a, H] &= P_a. \end{aligned} \tag{2.7}$$

This shows that $L_{ab}$ span an $\mathfrak{so}(3)$ subalgebra, relative to which $B_a, P_a$ transform according to the three-dimensional vector representation (which is also the adjoint representation in this dimension) and H transforms as the one-dimensional scalar representation. We shall see that all kinematical Lie algebras (with spatial isotropy) share these properties, which are strong enough to allow for their classification.

2.3. **Minkowski spacetime.** Minkowski spacetime is also described by $\mathbb{A}^4$, but the invariant notion is now that of a **proper distance** $\Delta : \mathbb{R}^4 \to \mathbb{R}$, sending $b - a \mapsto \Delta(b - a)$, where if $a = (x, 1)$ and $b = (y, 1)$,

$$\Delta(b - a) = (y - x)^\mathsf{T} \eta (y - x), \tag{2.8}$$

where

$$\eta = \begin{pmatrix} -1 & 0 & 0 & 0 \\ 0 & 1 & 0 & 0 \\ 0 & 0 & 1 & 0 \\ 0 & 0 & 0 & 1 \end{pmatrix}. \tag{2.9}$$

We no longer have a separate clock and ruler, or as Minkowski himself put it [62]:

> Von Stund' an sollen Raum für sich und Zeit für sich völlig zu Schatten herabsinken und nur noch eine Art Union der beiden soll Selbständigkeit bewahren.[3]

In particular, there is no longer an invariant notion of simultaneity between events, so instead of affine subspaces of simultaneity, we have lightcones at every spacetime event $a$: the **lightcone** $\mathbb{L}_a$ of $a$ being defined as those events which are a zero proper distance away from $a$:

$$\mathbb{L}_a = \left\{ b \in \mathbb{A}^4 \middle| \Delta(b - a) = 0 \right\}. \tag{2.10}$$

The kinematical group of Minkowski spacetime is the **Poincaré group** and consists of those affine transformations which preserve the proper distance between events. It embeds in $\mathrm{GL}(5, \mathbb{R})$ as those matrices

$$\begin{pmatrix} L & v \\ 0 & 1 \end{pmatrix} \tag{2.11}$$

where $L^\mathsf{T} \eta L = \eta$ and $v \in \mathbb{R}^4$. Matrix multiplication shows that the Poincaré group is isomorphic to the semidirect product $\mathrm{O}(3,1) \ltimes \mathbb{R}^4$, where $\mathrm{O}(3,1)$ is the **Lorentz group**. Acting on an event $(x, 1)$ with the matrix in equation (2.11), we obtain the event $(Lx + v, 1)$, which is the effect of a Lorentz transformation $(x \mapsto Lx)$ and a (spatiotemporal) translation $x \mapsto x + v$; that is,

$$\begin{pmatrix} L & v \\ 0 & 1 \end{pmatrix} = \begin{pmatrix} I & v \\ 0 & 1 \end{pmatrix} \begin{pmatrix} L & 0 \\ 0 & 1 \end{pmatrix}. \tag{2.12}$$

The Lie algebra of the Poincaré group embeds in $\mathfrak{gl}(5, \mathbb{R})$ as those matrices of the form

$$\begin{pmatrix} X & v \\ 0 & 0 \end{pmatrix} \tag{2.13}$$

where $X^\mathsf{T} \eta + \eta X = 0$ and $v \in \mathbb{R}^4$. Introducing a basis $L_{AB} = -L_{BA}, P_A$, where now $A, B = 0, 1, 2, 3$, by

$$\begin{pmatrix} X & v \\ 0 & 0 \end{pmatrix} = \tfrac{1}{2} X^{AB} L_{AB} + v^A P_A, \tag{2.14}$$

it is easy to calculate the nonzero Lie brackets:

$$[L_{AB}, L_{CD}] = \eta_{BC} L_{AD} - \eta_{AC} L_{BD} - \eta_{BD} L_{AC} + \eta_{AD} L_{BC}$$
$$[L_{AB}, P_C] = \eta_{BC} P_A - \eta_{AC} P_B. \tag{2.15}$$

To ease comparison with the Galilei algebra (2.7), we will let $P_A = (H = P_0, P_a)$ and $L_{AB} = (B_a = L_{0a}, L_{ab})$, relative to which the brackets become

$$[L_{ab}, L_{cd}] = \delta_{bc} L_{ad} - \delta_{ac} L_{bd} - \delta_{bd} L_{ac} + \delta_{bd} L_{ac}$$
$$[L_{ab}, B_b] = \delta_{bc} B_a - \delta_{ac} B_b$$
$$[L_{ab}, P_b] = \delta_{bc} P_a - \delta_{ac} P_b$$
$$[B_a, B_b] = L_{ab}$$
$$[B_a, P_b] = \delta_{ab} H$$
$$[B_a, H] = P_a. \tag{2.16}$$

We see that again $L_{ab}$ span an $\mathfrak{so}(3)$ subalgebra relative to which $B_a, P_a$ transform according to the three-dimensional vector representation and H transforms according to the one-dimensional scalar representation.

---

[3]Henceforth space by itself, and time by itself, are doomed to fade away into mere shadows, and only a kind of union of the two will preserve an independent reality.

What sets the Poincaré and Galilei algebras apart are the Lie brackets which do not involve the $L_{ab}$: the last bracket in equation (2.7) and the last three brackets in equation (2.16).

2.4. **Lie algebra of symmetries.** Minkowski spacetime is a lorentzian manifold, diffeomorphic to $\mathbb{R}^4$ with lorentzian metric

$$g = -dt^2 + dx^2 + dy^2 + dz^2 = \eta_{\mu\nu}dx^\mu dx^\nu \tag{2.17}$$

relative to cartesian coordinates $x^\mu = (t, x, y, z)$. The Poincaré Lie algebra is isomorphic to the Lie algebra of Killing vector fields of the metric $g$. Let $\xi = \xi^\mu \partial_\mu$ denote a vector field of Minkowski spacetime. It is a Killing vector field if $\mathscr{L}_\xi g = 0$, which translates into

$$\eta_{\rho\nu}\partial_\mu\xi^\rho + \eta_{\mu\rho}\partial_\nu\xi^\rho = 0, \tag{2.18}$$

or, defining $\xi_\mu = \eta_{\mu\rho}\xi^\rho$, into Killing's equation[4]:

$$\partial_\mu\xi_\nu + \partial_\nu\xi_\mu = 0. \tag{2.19}$$

Notice that $\partial_\mu\partial_\nu\xi_\rho$ is clearly symmetric in $\mu \leftrightarrow \nu$ and, from Killing's equation, also skewsymmetric in $\nu \leftrightarrow \rho$. Therefore $\partial_\mu\partial_\nu\xi_\rho = 0$ and hence $\xi_\mu = \Lambda_{\mu\nu}x^\nu + a_\mu$. Re-inserting this into Killing's equation, we find that $\Lambda_{\mu\nu} = -\Lambda_{\nu\mu}$ and we may write the general solution of Killing's equation as

$$\xi = \tfrac{1}{2}\Lambda^{\mu\nu}\xi_{L_{\mu\nu}} + a^\mu\xi_{P_\mu}, \tag{2.20}$$

where

$$\xi_{L_{\mu\nu}} = x_\nu\partial_\mu - x_\mu\partial_\nu \qquad \text{and} \qquad \xi_{P_\mu} = \partial_\mu. \tag{2.21}$$

One can check that these vector fields obey the opposite (i.e., negative) brackets of those of the Poincaré Lie algebra:

$$[\xi_{L_{\mu\nu}}, \xi_{L_{\rho\sigma}}] = -\eta_{\nu\rho}\xi_{L_{\mu\sigma}} + \eta_{\mu\rho}\xi_{L_{\nu\sigma}} + \eta_{\nu\sigma}\xi_{L_{\mu\rho}} - \eta_{\mu\sigma}\xi_{L_{\nu\rho}}$$
$$[\xi_{L_{\mu\nu}}, \xi_{P_\rho}] = -\eta_{\nu\rho}\xi_{P_\mu} + \eta_{\mu\rho}\xi_{P_\nu}. \tag{2.22}$$

The fact that we have an antihomomorphism of Lie algebras might seem counter-intuitive, but we will see that it is natural in the context of homogeneous spaces, where the group action is induced from left multiplication in the group. The infinitesimal generators of left multiplication are the right-invariant vector fields whose Lie brackets are opposite to those of the left-invariant vector fields defining the Lie algebra.

In contrast, Galilei spacetime is a non-lorentzian geometry: there is no invariant metric, but rather an invariant Newton–Cartan structure.[5] Relative to cartesian coordinates $(x, y, z, t)$, the clock defines a one-form $\tau = dt$. Indeed, as shown in Figure 1, the clock is the linear projection $\mathbb{R}^4 \to \mathbb{R}$ taking $b - a$ to $\tau(b - a)$. This is nothing but the derivative of the projection $\pi: \mathbb{A}^4 \to \mathbb{A}^1$, which in this model of the affine space is given by $\pi(x, y, z, t) = t$; in other words, $dt$. We will see later that in a general Newton–Cartan manifold, the clock one-form need not be exact or even closed. The ruler defines an invariant symmetric $(2, 0)$-tensor field $\lambda = \partial_x \otimes \partial_x + \partial_y \otimes \partial_y + \partial_y \otimes \partial_y$. Interpreting $\lambda$ as a symmetric bilinear form on one-forms, we notice that $\lambda$ is degenerate along $dt$. It is often called the "spatial cometric". In analogy with a lorentzian spacetime, let us say that a vector field $\xi$ is "Killing", if it preserves the clock one-form $\tau$ and the spatial cometric $\lambda$; that is,

$$\mathscr{L}_\xi\tau = 0 \qquad \text{and} \qquad \mathscr{L}_\xi\lambda = 0. \tag{2.23}$$

For Galilei spacetime, and introducing coordinates $x^a = (x, y, z)$, the general solution of equations (2.23) is given by

$$\xi = \alpha\partial_t + \nu^a(t)\partial_a + \tfrac{1}{2}T^{ab}(t)(x_b\partial_a - x_a\partial_b), \tag{2.24}$$

where $\alpha \in \mathbb{R}$ and where $\nu^a$ and $T^{ab} = -T^{ba}$ are smooth functions of $t$. In contrast to the Lie algebra of isometries of a lorentzian manifold, the Lie algebra of symmetries of the (weak) Newton–Cartan structure of Galilei spacetime is infinite-dimensional, and is known as the **Coriolis algebra** [64]. It contains (the opposite of) the Galilei algebra as a subalgebra, spanned by

$$\xi_H = \partial_t, \qquad \xi_{P_a} = \partial_a, \qquad \xi_{B_a} = t\partial_a \qquad \text{and} \qquad \xi_{L_{ab}} = -x_a\partial_b + x_b\partial_a. \tag{2.25}$$

---

[4]Solutions of equation (2.19) are the Noether charges for point symmetries of the geodesic equation. Indeed, if we consider the variational problem with lagrangian $\mathscr{L} = \tfrac{1}{2}\eta_{\mu\nu}\dot{x}^\mu\dot{x}^\nu$ and ask which point transformations $\delta x^\mu = \xi^\mu(x)$ leave $\mathscr{L}$ invariant, we find that $\xi^\mu$ must satisfy equation (2.19).

[5]Some authors (e.g., [63]) refer to this structure as a "weak" Newton–Cartan structure, reserving the unqualified name for the structure which results by an additional choice of an adapted connection; that is a connection relative to which the clock one-form and the spatial cometric are parallel.

Had we considered (strict) Newton–Cartan structures, including the adapted connection as part of the data, then the Lie algebra of symmetries would be finite-dimensional.[6]

## 3. Symmetry

In Section 2 we discussed two models of the universe: the Galilei and Minkowski spacetimes. Both are four-dimensional affine space homogeneous under the action of a kinematical Lie group: the Galilei and Poincaré groups, respectively. In this section we will discuss the notion of a kinematical Lie group in more formally and will discuss the classification of kinematical Lie algebras.

3.1. **Kinematical Lie algebras.** In a landmark paper [29] written more than half a century ago, Bacry and Lévy-Leblond asked themselves the question of which were the possible kinematics, rephrasing the question mathematically as the classification of kinematical Lie algebras. A careful comparison of the Poincaré and Galilei algebras we met in Section 2 suggests the following definition for four-dimensional spacetimes.[7]

**Definition 1.** A **kinematical Lie algebra** is a ten-dimensional (real) Lie algebra $\mathfrak{k}$ with generators $L_{ab} = -L_{ba}, B_a, P_a, H$ with $a, b = 1, 2, 3$ satisfying the following conditions:

- the generators $L_{ab}$ span an $\mathfrak{so}(3)$-subalgebra $\mathfrak{r}$ of $\mathfrak{g}$:

$$[L_{ab}, L_{cd}] = \delta_{bc}L_{ad} - \delta_{ac}L_{bd} - \delta_{bd}L_{ac} + \delta_{bd}L_{ac}, \tag{3.1}$$

- the generators $B_a, P_a$ transform as vectors under $\mathfrak{r}$:

$$[L_{ab}, B_b] = \delta_{bc}B_a - \delta_{ac}B_b$$
$$[L_{ab}, P_b] = \delta_{bc}P_a - \delta_{ac}P_b \tag{3.2}$$

- and the generator $H$ transforms as a scalar:

$$[L_{ab}, H] = 0. \tag{3.3}$$

In addition, Bacry and Lévy-Leblond initially also imposed that the Lie brackets should be invariant under parity $P_a \mapsto -P_a$ and time-reversal $H \mapsto -H$; although they did point out that those restrictions were "by no means compelling" and indeed twenty years later, Bacry and Nuyts [36] lifted those conditions arriving at a classification of four-dimensional kinematical Lie algebras. This classification was recovered using deformation theory in [34] and extended to arbitrary dimension in [35, 37]. The definition of kinematical algebra in $d + 1$ dimensions is formally as the one above, except that $a, b = 1, \ldots, d$ and the subalgebra $\mathfrak{r}$ spanned by $L_{ab}$ is now isomorphic to $\mathfrak{so}(d)$. The case of $d = 1$ corresponds to the Bianchi classification of three-dimensional real Lie algebras [38, 39], here re-interpreted as kinematical Lie algebras for two-dimensional spacetimes. The cases of $d = 2$ and $d = 3$ are the most complicated due to the existence of $\epsilon_{ab}$ and $\epsilon_{abc}$ which are $\mathfrak{r}$-invariant and can thus appear in the Lie brackets and, indeed, there are kinematical Lie algebras in dimension $2 + 1$ and $3 + 1$ which have no higher-dimensional analogues. We will refer the interested reader to the papers cited above and will concentrate here on those kinematical Lie algebras which exist in generic dimensions.

Before we state the classification, let us make an important remark. Although the notation for the generators of a kinematical Lie algebra suggests a physical interpretation: namely, $L_{ab}$ generate rotations, $B_a$ boots, $P_a$ spatial translations and $H$ temporal translations, it would be imprudent to take this very seriously. The physical interpretation of the generators can only be determined once we realise them geometrically as vector fields in a spacetime. In the two examples we have seen in Section 2, it is indeed the case that the generators can be interpreted as above, but this is certainly not true in most cases.

One way to approach the classification is to write down the most general $\mathfrak{r}$-invariant Lie brackets for the generators $B_a, P_a, H$ and impose the Jacobi identity. The Jacobi identity cuts out an algebraic variety $\mathscr{J}$ in the vector space of possible brackets: i.e., the vector space of linear maps $\wedge^2 W \to \mathfrak{k}$, where $W \subset \mathfrak{k}$ is the vector subspace spanned by $B_a, P_a, H$. Two points in $\mathscr{J}$ define isomorphic kinematical Lie algebras if and only if they are related by a change of basis in $W$. We take care of this ambiguity by quotienting $\mathscr{J}$ by the action of the subgroup of $GL(W)$ which commutes with the action of $\mathfrak{r}$. In practice, one selects a unique representative for each isomorphism class of kinematical Lie algebras.

---

[6]The reason is that Newton–Cartan structures are Cartan geometries and the infinitesimal automorphisms of a Cartan geometry form a finite-dimensional Lie algebra.

[7]Strictly speaking, the definition is for spatially isotropic spacetimes. There are generalisations where the rotational subalgebra $\mathfrak{r}$ in the definition is replaced by a Lorentz subalgebra. Such homogeneous spaces do occur in nature. Indeed, as shown in [65], the blow-up of spatial infinity of Minkowski spacetime is a homogeneous space of the Poincaré group with lorentzian isotropy. There are other homogeneous spaces of the Poincaré group occurring at the asymptotic infinities of Minkowski spacetime, as discussed in [66].

Table 1 lists the kinematical Lie algebras in generic dimension $d + 1$. For $d \leqslant 2$, there are some degeneracies (e.g., if $d = 2$, the Galilei algebra $\mathfrak{g}$ is isomorphic to the Carroll algebra $\mathfrak{c}$), but for general $d$ the table below lists non-isomorphic kinematical Lie algebras and for $d > 3$ the table is complete. The table lists the nonzero Lie brackets except for the common ones in every kinematical Lie algebra. It also uses a shorthand notation omitting indices. The only $\mathfrak{r}$-invariant tensor which can appear is $\delta_{ab}$ and hence there is an unambiguous way to add indices. For example, $[H, \mathbf{B}] = \mathbf{B} + \mathbf{P}$ unpacks as $[H, B_a] = B_a + P_a$, whereas $[\mathbf{B}, \mathbf{P}] = H + L$ stands for $[B_a, P_b] = \delta_{ab} H + L_{ab}$, et cetera. There is no standard notation for all the kinematical Lie algebras, so we have made some choices.

TABLE 1. Kinematical Lie algebras in generic dimension

| Name | Nonzero Lie brackets in addition to (3.1)–(3.3) | | | | Comments |
|---|---|---|---|---|---|
| $\mathfrak{s}$ | | | | | |
| $\mathfrak{g}$ | $[H, \mathbf{B}] = -\mathbf{P}$ | | | | |
| $\mathfrak{n}^0$ | $[H, \mathbf{B}] = \mathbf{B} + \mathbf{P}$ | $[H, \mathbf{P}] = \mathbf{P}$ | | | |
| $\mathfrak{n}_\gamma^+$ | $[H, \mathbf{B}] = \gamma\mathbf{B}$ | $[H, \mathbf{P}] = \mathbf{P}$ | | | $\gamma \in [-1, 1]$ |
| $\mathfrak{n}_\chi^-$ | $[H, \mathbf{B}] = \chi\mathbf{B} + \mathbf{P}$ | $[H, \mathbf{P}] = \chi\mathbf{P} - \mathbf{B}$ | | | $\chi \geqslant 0$ |
| $\mathfrak{c}$ | | | $[\mathbf{B}, \mathbf{P}] = H$ | | |
| $\mathfrak{iso}(d,1)$ $\mathfrak{iso}(d+1)$ | $[H, \mathbf{B}] = -\varepsilon\mathbf{P}$ | | $[\mathbf{B}, \mathbf{B}] = \varepsilon L$ $[\mathbf{B}, \mathbf{P}] = H$ | | $\varepsilon = \pm 1$ |
| $\mathfrak{so}(d+1, 1)$ | $[H, \mathbf{B}] = \mathbf{B}$ | $[H, \mathbf{P}] = -\mathbf{P}$ | $[\mathbf{B}, \mathbf{P}] = H + L$ | | |
| $\mathfrak{so}(d,2)$ $\mathfrak{so}(d+2)$ | $[H, \mathbf{B}] = -\varepsilon\mathbf{P}$ | $[H, \mathbf{P}] = \varepsilon\mathbf{B}$ | $[\mathbf{B}, \mathbf{B}] = \varepsilon L$  $[\mathbf{B}, \mathbf{P}] = H$ | $[\mathbf{P}, \mathbf{P}] = \varepsilon L$ | $\varepsilon = \pm 1$ |

We now describe each of the algebras in turn:
- The Lie algebra $\mathfrak{s}$ is the **static** kinematical Lie algebra: all additional brackets are zero. Therefore every kinematical Lie algebras is a deformation of $\mathfrak{s}$.
- The Galilei algebra is denoted $\mathfrak{g}$ and we have denoted by $\mathfrak{n}^0$ a closely related algebra. In $\mathfrak{g}$ and $\mathfrak{n}^0$, the adjoint action of $H$ is not diagonalisable over the complex numbers, but has a nontrivial Jordan block:

$$\text{ad}_H^{\mathfrak{g}} \begin{pmatrix} \mathbf{B} \\ \mathbf{P} \end{pmatrix} = \begin{pmatrix} 0 & -1 \\ 0 & 0 \end{pmatrix} \begin{pmatrix} \mathbf{B} \\ \mathbf{P} \end{pmatrix} \qquad \text{and} \qquad \text{ad}_H^{\mathfrak{n}^0} \begin{pmatrix} \mathbf{B} \\ \mathbf{P} \end{pmatrix} = \begin{pmatrix} 1 & 1 \\ 0 & 1 \end{pmatrix} \begin{pmatrix} \mathbf{B} \\ \mathbf{P} \end{pmatrix}. \tag{3.4}$$

- There are two one-parameter families of algebras: $\mathfrak{n}_\gamma^+$, with $\gamma \in [-1, 1]$, which for $\gamma = -1$ is one of the two **Newton–Hooke** algebras; and $\mathfrak{n}_\chi^-$, with $\chi \geqslant 0$, which for $\chi = 0$ is the other Newton–Hooke algebra. These two families correspond to the cases where the adjoint action of $H$ is diagonalisable over the complex numbers: in $\mathfrak{n}_\gamma^+$, the eigenvalues are real, whereas in $\mathfrak{n}_\chi^-$ they are complex:

$$\text{ad}_H^{\mathfrak{n}^+} \begin{pmatrix} \mathbf{B} \\ \mathbf{P} \end{pmatrix} = \begin{pmatrix} \gamma & 0 \\ 0 & 1 \end{pmatrix} \begin{pmatrix} \mathbf{B} \\ \mathbf{P} \end{pmatrix} \qquad \text{and} \qquad \text{ad}_H^{\mathfrak{n}^-} \begin{pmatrix} \mathbf{B} \\ \mathbf{P} \end{pmatrix} = \begin{pmatrix} \chi & 1 \\ -1 & \chi \end{pmatrix} \begin{pmatrix} \mathbf{B} \\ \mathbf{P} \end{pmatrix}. \tag{3.5}$$

- The Carroll algebra is denoted $\mathfrak{c}$.
- The Poincaré algebra is $\mathfrak{iso}(d, 1)$ and the euclidean algebra is $\mathfrak{iso}(d + 1)$.
- The remaining algebras are semisimple (for $d \geqslant 2$) and consist of $\mathfrak{so}(d + 2)$, $\mathfrak{so}(d + 1, 1)$ and $\mathfrak{so}(d, 2)$. Finite-dimensional semisimple Lie algebras are rigid, so they cannot be deformed further. However they can be contracted. Not all the kinematical Lie algebras in the table can be obtained as contractions of the simple ones: those which can are the Poincaré, euclidean, (both) Newton–Hooke, Galilei, Carroll and static algebras. These are precisely the algebras which admit parity and time-reversal automorphisms; that is, the ones originally classified in [29].

3.2. **Aristotelian Lie algebras.** A closely related family of Lie algebras are the **aristotelian algebras**, defined just like in Definition 1, but dropping the boosts.

**Definition 2.** An **aristotelian Lie algebra** is a real Lie algebra $\mathfrak{a}$ with generators $L_{ab} = -L_{ba}, P_a, H$, with $a, b = 1, \dots, d$, satisfying the following conditions:
- the generators $L_{ab}$ span an $\mathfrak{so}(d)$-subalgebra $\mathfrak{r}$ of $\mathfrak{g}$:

$$[L_{ab}, L_{cd}] = \delta_{bc} L_{ad} - \delta_{ac} L_{bd} - \delta_{bd} L_{ac} + \delta_{bd} L_{ac}, \tag{3.6}$$

- the generators $P_a$ transform as vectors under $\mathfrak{r}$:

$$[L_{ab}, P_b] = \delta_{bc} P_a - \delta_{ac} P_b \tag{3.7}$$

- and the generator H transforms as a scalar:

$$[\mathsf{L}_{ab}, \mathsf{H}] = 0. \tag{3.8}$$

Aristotelian Lie algebras are easy to classify in any dimension and the result is contained in [40, Appendix B] and summarised in the Table 2, which lists the nonzero Lie brackets in addition to those fixed by the definition. We omit aristotelian Lie algebras which do not exist in general dimension.

TABLE 2. Aristotelian Lie algebras

| Name | Nonzero Lie brackets | Comments |
|---|---|---|
| $\mathfrak{iso}(d) \oplus \mathbb{R}$ | | |
| $\mathfrak{sim}(d)$ | $[\mathsf{H}, \mathsf{P}_a] = \mathsf{P}_a$ | |
| $\mathfrak{so}(d,1)\oplus\mathbb{R}$ $\mathfrak{so}(d+1)\oplus\mathbb{R}$ | $[\mathsf{P}_a, \mathsf{P}_b] = \varepsilon \mathsf{L}_{ab}$ | $\varepsilon = \pm 1$ |

Let us describe each of the aristotelian Lie algebras in turn:

- The aristotelian Lie algebra with no additional nonzero Lie brackets, which we could term the "static" aristotelian Lie algebra, is isomorphic to $\mathfrak{iso}(d) \oplus \mathbb{R}$, with $\mathfrak{iso}(d)$ spanned by $\mathsf{L}_{ab}, \mathsf{P}_a$ and the one-dimensional Lie subalgebra spanned by H, which is central.
- If instead of being central, we think of H as dilatations, we obtain a Lie algebra isomorphic to the similitude algebra of $d$-dimensional euclidean space: $\mathfrak{sim}(d)$. This is also denoted $\mathfrak{co}(d) \ltimes \mathbb{R}^d$, where $\mathfrak{co}(d) = \mathfrak{so}(d) \oplus \mathbb{R}$ is the extension of the rotation algebra by dilatations and $\mathbb{R}^d$ transforms as a vector under rotations but with nonzero conformal weight.
- If H remains central, but now the translations do not commute, we obtain trivial central extensions of $\mathfrak{so}(d,1)$ or $\mathfrak{so}(d+1)$.

3.3. **Central extensions.** Central extensions of Lie algebras arise naturally in Physics. In quantum Physics they arise due to the fact that the state space of a quantum system is a projective space (the space of rays of a Hilbert space) so that the action of a group $\mathscr{G}$ on the projective space may only lift to a projective representation on the Hilbert space, and hence an honest representations of a one-dimensional central extension of $\mathscr{G}$. In classical Physics they arise due to the fact that homogeneous symplectic manifolds of a Lie group $\mathscr{G}$ are (up to covering) coadjoint orbits of $\mathscr{G}$ or perhaps a one-dimensional central extension of $\mathscr{G}$, as we will discuss in Section 4.4.

Mathematically, a central extension of a Lie algebra $\mathfrak{k}$ is a special case of a Lie algebra extension. A Lie algebra $\widetilde{\mathfrak{k}}$ is said to be a an **extension** of a Lie algebra $\mathfrak{k}$ by a Lie algebra $\mathfrak{a}$ if they fit in an exact sequence of Lie algebras

$$0 \longrightarrow \mathfrak{a} \longrightarrow \widetilde{\mathfrak{k}} \longrightarrow \mathfrak{k} \longrightarrow 0, \tag{3.9}$$

This is equivalent to the following conditions: $\widetilde{\mathfrak{k}} = \mathfrak{k} \oplus \mathfrak{a}$ as a vector space, $\mathfrak{a}$ is an ideal of $\widetilde{\mathfrak{k}}$ (i.e., $[\widetilde{\mathfrak{k}}, \mathfrak{a}] \subset \mathfrak{a}$) and the quotient Lie algebra $\widetilde{\mathfrak{k}}/\mathfrak{a}$ is isomorphic to $\mathfrak{k}$. If $\mathfrak{a}$ is central, so that $[\mathfrak{a}, \widetilde{\mathfrak{k}}] = 0$, then we have a **central extension**. Notice that $\mathfrak{k}$ is not necessarily a Lie subalgebra of $\widetilde{\mathfrak{k}}$. If that is the case, the sequence is said to be split and we have that $\widetilde{\mathfrak{k}}$ is the semidirect product of $\mathfrak{k}$ with $\mathfrak{a}$. A special case of semidirect products are the trivial extensions, when $\widetilde{\mathfrak{k}} = \mathfrak{k} \oplus \mathfrak{a}$ as a Lie algebra; that is, $\mathfrak{k}$ and $\mathfrak{a}$ are subalgebras (actually ideals) and $[\mathfrak{k}, \mathfrak{a}] = 0$.

Whereas every Lie algebra admits trivial extensions, the only kinematical Lie algebras in Table 1 admitting nontrivial central extensions (in dimension $d > 2$) are the static ($\mathfrak{s}$), Newton–Hooke ($\mathfrak{n}^\pm$) and Galilei ($\mathfrak{g}$) algebras. To describe them, we introduce a new generator Z with $[\mathsf{Z}, -] = 0$ and modify the Lie brackets of the kinematical Lie algebra by $[\mathsf{B}_a, \mathsf{P}_b] = \delta_{ab}\mathsf{Z}$. The central extension of the Galilei algebra is called the **Bargmann algebra**.

A one-dimensional extension (not necessarily central) of a kinematical Lie algebra is called a **generalised Bargmann algebra**. Apart from the central extensions listed above and the trivial extensions, there is a small list (some with parameters). Those for which $[\mathsf{B}_a, \mathsf{P}_b] = \delta_{ab}\mathsf{Z}$ are deformations of the central extension of the static kinematical Lie algebra and have been classified in [34] (for $d = 3$) and in [35] (for $d > 3$). Those for which $[\mathsf{B}_a, \mathsf{P}_b] = 0$ are listed here for the first time.

Table 3 lists the (nontrivial) generalised Bargmann algebras in dimension $d > 2$. In the table $\mathfrak{k}$ stands for the kinematical Lie algebra being extended and the brackets listed are the ones which involve the additional generator Z, so they are either new or modifications of the brackets in $\mathfrak{k}$. The (nonzero) parameter $\alpha$ in the last three rows is effective: different values of $\alpha$ give non-isomorphic Lie algebras.

TABLE 3. Generalised Bargmann algebras in $\mathfrak{d} > 2$

| $\mathfrak{k}$ | Brackets involving $\mathsf{Z}$ | | Comments |
|---|---|---|---|
| $\mathfrak{s}$ | $[\mathbf{B}, \mathbf{P}] = \mathsf{Z}$ | | |
| $\mathfrak{n}^+$ | $[\mathbf{B}, \mathbf{P}] = \mathsf{Z}$ | | |
| $\mathfrak{n}^-$ | $[\mathbf{B}, \mathbf{P}] = \mathsf{Z}$ | | |
| $\mathfrak{g}$ | $[\mathbf{B}, \mathbf{P}] = \mathsf{Z}$ | | |
| $\mathfrak{n}^+_\gamma$ | $[\mathbf{B}, \mathbf{P}] = \mathsf{Z}$ | $[\mathsf{H}, \mathsf{Z}] = (\gamma + 1)\mathsf{Z}$ | $\gamma \in (-1, 1]$ |
| $\mathfrak{n}^0$ | $[\mathbf{B}, \mathbf{P}] = \mathsf{Z}$ | $[\mathsf{H}, \mathsf{Z}] = 2\mathsf{Z}$ | |
| $\mathfrak{n}^-_\chi$ | $[\mathbf{B}, \mathbf{P}] = \mathsf{Z}$ | $[\mathsf{H}, \mathsf{Z}] = 2\chi\mathsf{Z}$ | $\chi > 0$ |
| $\mathfrak{s}$ | | $[\mathsf{H}, \mathsf{Z}] = \mathsf{Z}$ | |
| $\mathfrak{g}$ | | $[\mathsf{H}, \mathsf{Z}] = \mathsf{Z}$ | |
| $\mathfrak{n}^+_\gamma$ | | $[\mathsf{H}, \mathsf{Z}] = \alpha\mathsf{Z}$ | $\gamma \in [-1, 1]$ and $\alpha \neq 0$ |
| $\mathfrak{n}^0$ | | $[\mathsf{H}, \mathsf{Z}] = \alpha\mathsf{Z}$ | $\alpha \neq 0$ |
| $\mathfrak{n}^-_\chi$ | | $[\mathsf{H}, \mathsf{Z}] = \alpha\mathsf{Z}$ | $\chi \geqslant 0$ and $\alpha \neq 0$ |

Aristotelian Lie algebras (if $\mathfrak{d} > 2$) admit no nontrivial central extensions: there are no $\mathfrak{r}$-invariant cochains, let alone cocycles. They do, however, admit nontrivial non-central extensions, which are listed in Table 4, which lists the aristotelian Lie algebra being extended and the brackets involving the additional generator $\mathsf{Z}$. Again the (nonzero) parameter $\alpha$ is effective.

TABLE 4. One-dimensional extensions of aristotelian Lie algebras in $\mathfrak{d} > 2$

| $\mathfrak{a}$ | Brackets involving $\mathsf{Z}$ | Comments |
|---|---|---|
| $\mathfrak{iso}(\mathfrak{d}) \oplus \mathbb{R}$ | $[\mathsf{H}, \mathsf{Z}] = \mathsf{Z}$ | |
| $\mathfrak{sim}(\mathfrak{d})$ | $[\mathsf{H}, \mathsf{Z}] = \alpha\mathsf{Z}$ | $\alpha \neq 0$ |
| $\mathfrak{so}(\mathfrak{d}, 1) \oplus \mathbb{R}$ $\mathfrak{so}(\mathfrak{d}+1) \oplus \mathbb{R}$ | $[\mathsf{H}, \mathsf{Z}] = \mathsf{Z}$ | |

Changing notation: $(\mathsf{H}, \mathsf{Z}) \mapsto (\mathsf{D}, \mathsf{H})$, the Lie algebras in Table 4 are examples of Lifshitz Lie algebras (see, e.g., [67]). The extension of $\mathfrak{sim}(\mathfrak{d})$ is the original Lifshitz algebra, where the parameter $\alpha$ is typically denoted $z$:

$$[\mathsf{D}, \mathbf{P}] = \mathbf{P} \qquad \text{and} \qquad [\mathsf{D}, \mathsf{H}] = z\mathsf{H}, \tag{3.10}$$

in addition to the brackets (3.6)–(3.8) common to all aristotelian Lie algebras.

## 4. GEOMETRY

In this section we discuss non-lorentzian geometries. In the spirit of Klein's Erlangen Programme, we start by discussing the homogeneous spacetimes associated to the kinematical Lie algebras discussed in Section 3. We will see that these spaces fall into different families depending on the structure which the kinematical group preserves: metric (riemannian or lorentzian), Newton–Cartan, carrollian or aristotelian. Each of these geometries is a Cartan geometry modelled on a homogeneous spacetime and we will discuss them in turn. In a sense, all we are doing is extending to the non-lorentzian context the standard sequence of ideas:

$$\text{Poincaré symmetry} \longrightarrow \text{Minkowski spacetime} \longrightarrow \text{lorentzian geometry.} \tag{4.1}$$

Of course, Minkowski spacetime is not the only homogeneous space of the Poincaré group, so the passage from the Poincaré group to Minkowski spacetime requires a choice, whereas the passage from Minkowski spacetime to lorentzian geometry is more or less forced.

4.1. **Homogeneous spaces.** In this section we review the basic notions of homogeneous geometry.

4.1.1. *Group actions on manifolds.* Let $\mathscr{G}$ be a Lie group. A (linear) representation of $\mathscr{G}$ on a vector space $V$ is a Lie group homomorphism $\rho : \mathscr{G} \to \mathrm{GL}(V)$; that is, $\rho$ is a smooth map and a group homomorphism $\rho(ab) = \rho(a)\rho(b)$ for all $a, b \in \mathscr{G}$. We are also interested in nonlinear realisations[8] of $\mathscr{G}$ on a manifold $M$. It would be tempting by analogy with the case of a linear representation to define a nonlinear realisation as a Lie group homomorphism $\rho : \mathscr{G} \to \mathrm{Diff}(M)$, except for the fact that the diffeomorphism group $\mathrm{Diff}(M)$ of a manifold is not typically a Lie group. Instead we define nonlinear realisations as *actions*. One has to distinguish between left and right actions; although it is easy to go between them. By a **(left) action** of $\mathscr{G}$ on a manifold $M$ we mean a smooth map $\alpha : \mathscr{G} \times M \to M$, written simply as $\alpha(g, p) = g \cdot p$, satisfying two properties:

- for all $g_1, g_2 \in \mathscr{G}$ and $p \in M$, $(g_1 g_2) \cdot p = g_1 \cdot (g_2 \cdot p)$; and
- for all $p \in M$, $e \cdot p = p$ where $e \in \mathscr{G}$ is the identity element.

If we fix $g \in \mathscr{G}$, $\alpha(g, -) : M \to M$ is a diffeomorphism which we typically denote $\alpha_g$. On the other hand, if we fix $p \in M$, we get a map $\alpha(-, p) : \mathscr{G} \to M$ known as the **orbit map**, as its image is the orbit of $p$ under $\mathscr{G}$.

Let $\mathfrak{g}$ denote the Lie algebra of $\mathscr{G}$. An action of $\mathscr{G}$ on $M$ gives rise to a Lie algebra antihomomorphism $\xi : \mathfrak{g} \to \mathscr{X}(M)$, assigning to every $X \in \mathfrak{g}$ a vector field $\xi_X$ and such that for all $X, Y \in \mathfrak{g}$, $[\xi_X, \xi_Y] = -\xi_{[X,Y]}$, where the bracket on the LHS is the Lie bracket of vector fields and that on the RHS is the bracket on $\mathfrak{g}$. The vector fields in the image of $\xi$ are called the **fundamental vector fields** of the group action. Although one can redefine the fundamental vector fields in such a way that the new map $\mathfrak{g} \to \mathscr{X}(M)$ is a Lie algebra homomorphism, it turns out not to be natural, as we will see shortly.

A group action $\alpha : \mathscr{G} \times M \to M$ is said to be **effective** if the only element $g \in \mathscr{G}$ which acts trivially (i.e., which obeys $g \cdot p = p$ for all $p \in M$) is the identity element. A weaker condition is for the action to be **locally effective**, which says that the elements of $\mathscr{G}$ which act trivially form a discrete subgroup of $\mathscr{G}$. This is equivalent to the map $\xi : \mathfrak{g} \to \mathscr{X}(M)$ being injective, so that no nonzero element in $\mathfrak{g}$ is sent to the zero vector field.

A group action $\alpha : \mathscr{G} \times M \to M$ is said to be **transitive** if given any two points $p, q \in M$, there is some $g \in \mathscr{G}$ with $q = g \cdot p$. Equivalently, if the $\mathscr{G}$-orbit of any point is the whole manifold. This is the analogue for nonlinear realisations of irreducibility for linear representations. A linear representation of $\mathscr{G}$ on $V$ is irreducible if there are no proper subspaces of $V$ which are stable under $\mathscr{G}$. Similarly, an action of $\mathscr{G}$ on $M$ is transitive if there are no proper submanifolds of $M$ stable under the action of $\mathscr{G}$.

A manifold $M$ is said to be a **homogeneous space** of a Lie group $\mathscr{G}$ if $\mathscr{G}$ acts transitively on $M$. The **stabiliser subgroup** of a point $p \in M$ is the subgroup $\mathscr{H} \subset \mathscr{G}$ which fixes $p$: $\mathscr{H} = \{g \in \mathscr{G} | g \cdot p = p\}$. It is a closed subgroup of $\mathscr{G}$. Its Lie algebra $\mathfrak{h}$ consists of those fundamental vector fields which vanish at $p$. If $M$ is a homogeneous space of $\mathscr{G}$, then the stabiliser subgroups of all of its points are conjugate in $\mathscr{G}$. Indeed, let $\mathscr{H}_p$ denote the stabiliser subgroup of $p \in M$ and $\mathscr{H}_q$ that of $q \in M$. Since $\mathscr{G}$ acts transitively, there is some $g \in \mathscr{G}$ such that $q = g \cdot p$ and hence $h \in \mathscr{H}_q$ if and only if $h = gh'g^{-1}$ for some $h' \in \mathscr{H}_p$. Often one picks an "origin" $o \in M$ and lets $\mathscr{H}$ denote the stabiliser subgroup of $o$. Then $M$ is diffeomorphic to the space of left cosets $\mathscr{G}/\mathscr{H}$. This is why homogeneous spaces are often referred to as *coset spaces* or *coset manifolds*. Of course the choice of origin is immaterial, since from the point of $\mathscr{G}$ all points in a homogeneous space "look the same".

It is not just that $M$ and $\mathscr{G}/\mathscr{H}$ are diffeomorphic, but that they are $\mathscr{G}$-equivariantly so: the diffeomorphism $M \to \mathscr{G}/\mathscr{H}$ intertwines between the left action of $\mathscr{G}$ on $M$ and the left action of $\mathscr{G}$ on $\mathscr{G}/\mathscr{H}$ which is induced from left multiplication in $\mathscr{G}$: if $g'\mathscr{H} \in \mathscr{G}/\mathscr{H}$ and $g \in \mathscr{G}$, we have that $g \cdot g'\mathscr{H} = (gg')\mathscr{H}$. Now recall that the vector fields which generate left multiplication on $\mathscr{G}$ are the right-invariant vector fields and they satisfy the opposite Lie algebra. This explains why it is natural for the map $\xi : \mathfrak{g} \to \mathscr{X}(M)$ to be an antihomomorphism.

4.1.2. *Linear isotropy representation and invariant tensors.* Let $M$ be a homogeneous space of $\mathscr{G}$ and $\mathscr{H} \subset \mathscr{G}$ the stabiliser of the origin $o \in M$. Since every $h \in \mathscr{H}$ preserves $o$, the derivative at $o$ of the diffeomorphism $\alpha_h : M \to M$ defines a linear transformation $\lambda(h)$ of the tangent space $T_o M$. Since $\alpha$ is an action, in particular, $\alpha_{h_1} \circ \alpha_{h_2} = \alpha_{h_1 h_2}$ for all $h_1, h_2 \in \mathscr{H}$ and, by the chain rule, $\lambda : \mathscr{H} \to \mathrm{GL}(T_o M)$ is a representation, known as the **linear isotropy representation**.

The linear isotropy representation plays a very important rôle in determining the $\mathscr{G}$-invariant tensor fields on a homogeneous space $M$. An important result, which is a special case of the *fundamental*

---

[8]In Physics it is customary to reserve the name "nonlinear realisation" only to transitive actions (see later), when $M$ is diffeomorphic to a coset space $\mathscr{G}/\mathscr{H}$. A manifold admitting a transitive action of a Lie group is the nonlinear analogue of an *irreducible* representation. In the same way that it is useful to consider representations which are not necessarily irreducible, we shall consider nonlinear realisations where the action is not necessarily transitive.

*principle of holonomy* (see, e.g., [68, Para. 10.19] in the riemannian case, but holds more generally for any connection), states that there is a one-to-one correspondence between $\mathscr{H}$-invariant tensors on $\mathsf{T_o M}$ and $\mathscr{G}$-invariant tensor fields on $\mathsf{M}$. Briefly, it goes as follows. If $\Phi$ is a $\mathscr{G}$-invariant tensor field on $\mathsf{M}$, its value at the origin is a tensor $\Phi_o$ on $\mathsf{T_o M}$ which is invariant under the linear isotropy representation of $\mathscr{H}$. Conversely, given an $\mathscr{H}$-invariant tensor $\Phi_o$ on $\mathsf{T_o M}$ we may extend it to a tensor field on $\mathsf{M}$ via the $\mathscr{G}$ action. Its value $\Phi_p$ at $p \in \mathsf{M}$ is defined by picking $g \in \mathscr{G}$ with $g \cdot o = p$ and acting on $\Phi_o$ with $g$: $\Phi_p = g \cdot \Phi_o$. The problem is that there is typically not a unique $g \in \mathscr{G}$ connecting $o$ to $p$, so which one do we choose? It turns out that the choice is immaterial: if $g' \in \mathscr{G}$ is any other such element, then $g' = gh$ for some $h \in \mathscr{H}$ and precisely because $\Phi_o$ is $\mathscr{H}$-invariant, $g' \cdot \Phi_o = g \cdot \Phi_o$ and it does not matter whether we use $g$ or $g'$ to calculate $\Phi_p$.

If in addition, $\mathscr{H}$ is a connected subgroup with Lie algebra $\mathfrak{h} \subset \mathfrak{g}$, then $\mathscr{G}$-invariant tensor fields on $\mathsf{M}$ are in one-to-one correspondence with $\mathfrak{h}$-invariant tensors on $\mathsf{T_o M}$. Determining the $\mathfrak{h}$-invariant tensors is a reasonably simple linear algebra problem in most cases.

4.1.3. *Klein pairs.* Let $\mathsf{M}$ be a homogeneous space of $\mathscr{G}$ with typical stabiliser $\mathscr{H}$. Let $\mathfrak{g}$ and $\mathfrak{h}$ denote the Lie algebras of $\mathscr{G}$ and $\mathscr{H}$, respectively. Then we may associate to $\mathsf{M}$ the **Klein pair** $(\mathfrak{g}, \mathfrak{h})$. Not every pair $(\mathfrak{g}, \mathfrak{h})$ consisting of a Lie algebra $\mathfrak{g}$ and a Lie subalgebra $\mathfrak{h}$ is a Klein pair. It has to be *geometrically realisable*, which says that there exists some Lie group $\mathscr{G}$ with Lie algebra $\mathfrak{g}$ such that the connected subgroup $\mathscr{H}$ generated by $\mathfrak{h}$ is closed. As explained, for example, in [40, Appendix B], there is a one-to-one correspondence between (effective, geometrically realisable) Klein pairs and simply-connected homogeneous spaces of $\mathscr{G}$. (See [40, Appendix B.3] for a simple example of a Klein pair which is not geometrically realisable.) Paraphrasing slightly, (effective, geometrically realisable) Klein pairs classify homogeneous spaces up to covering, in the same way that Lie algebras classify Lie groups up to covering.

There is a notion of isomorphism between Klein pairs which is crucial in classifications. We say that two Klein pairs $(\mathfrak{g}_1, \mathfrak{h}_1)$ and $(\mathfrak{g}_2, \mathfrak{h}_2)$ are *isomorphic*, if there is a Lie algebra isomorphism $\varphi : \mathfrak{g}_1 \to \mathfrak{g}_2$ with $\varphi(\mathfrak{h}_1) = \mathfrak{h}_2$. Isomorphic Klein pairs, if geometrically realisable, give rise to locally isomorphic homogeneous spaces.

Let $\mathsf{M}$ be a homogeneous $\mathscr{G}$-space with Klein pair $(\mathfrak{g}, \mathfrak{h})$. We say that the Klein pair is **reductive** if there exists a complementary subspace $\mathfrak{m}$ with $\mathfrak{g} = \mathfrak{h} \oplus \mathfrak{m}$ which is stable under the restriction to $\mathscr{H}$ of the adjoint action of $\mathscr{G}$ on $\mathfrak{g}$. If $\mathscr{H}$ is connected, reductivity says that $[\mathfrak{h}, \mathfrak{m}] \subset \mathfrak{m}$. In the reductive case, the vector space isomorphism $\mathsf{T_o M} \cong \mathfrak{m}$ intertwines between the linear isotropy representation on $\mathsf{T_o M}$ and the restriction to $\mathscr{H}$ of the $\mathscr{G}$-adjoint representation on $\mathfrak{m}$. In the non-reductive case, there is a vector space isomorphism $\mathsf{T_o M} \cong \mathfrak{g}/\mathfrak{h}$, where the quotient vector space $\mathfrak{g}/\mathfrak{h}$ is naturally a representation of $\mathscr{H}$. In practice we work with $\mathfrak{g}/\mathfrak{h}$ by working with $\mathfrak{g}$ and just dropping any terms belonging to $\mathfrak{h}$ at the end.

A reductive Klein pair $(\mathfrak{g} = \mathfrak{h} \oplus \mathfrak{m}, \mathfrak{h})$ is said to be **symmetric** if $[\mathfrak{m}, \mathfrak{m}] \subset \mathfrak{h}$. Symmetric Klein pairs are the infinitesimal description (up to coverings) of symmetric spaces.

It may be convenient to write the reductive and symmetry conditions in a basis. Let $\mathsf{X}_i$ denote a basis for $\mathfrak{h}$. The Klein pair $(\mathfrak{g}, \mathfrak{h})$ is reductive if we can complete to a basis $\mathsf{X}_i, \mathsf{Y}_I$ for $\mathfrak{g}$ such that $[\mathsf{X}_i, \mathsf{Y}_I] = c_{iI}{}^J \mathsf{Y}_J$; that is, no $\mathsf{X}_i$ appear in the RHS. For a reductive Klein pair, $[\mathsf{Y}_I, \mathsf{Y}_J] = c_{IJ}{}^i \mathsf{X}_i + c_{IJ}{}^K \mathsf{Y}_K$ in general, but if it is symmetric then $c_{IJ}{}^K = 0$ and hence $[\mathsf{Y}_I, \mathsf{Y}_J] = c_{IJ}{}^i \mathsf{X}_i$.

4.1.4. *Exponential coordinates.* Let us now discuss coordinates on homogeneous spaces, but first we review exponential coordinates on a Lie group.

In the neighbourhood of any point $g$ in a Lie group $\mathscr{G}$, we have exponential coordinates associated to every choice of basis for the Lie algebra $\mathfrak{g}$. Recall that the exponential map $\exp : \mathfrak{g} \to \mathscr{G}$, which for a matrix Lie group is just the matrix exponential, is a diffeomorphism between a neighbourhood of $0 \in \mathfrak{g}$ and a neighbourhood of the identity $e \in \mathscr{G}$. Let $\mathsf{X}_1, \ldots, \mathsf{X}_n$ be a basis for $\mathfrak{g}$ and consider $\exp(x^1 \mathsf{X}_1 + \cdots x^n \mathsf{X}_n) \in \mathscr{G}$. The $(x^1, \ldots, x^n)$ are local coordinates for $\mathscr{G}$ centred at the identity, which has coordinates $(0, \ldots, 0)$. We may now use left (or right) multiplication to give coordinates in a neighbourhood of any other $g \in \mathscr{G}$; for example, $g \exp(y^1 \mathsf{X}_1 + \cdots y^n \mathsf{X}_n)$ give local coordinates for $\mathscr{G}$ near $g$. On overlaps, the change of coordinates between these exponential coordinates is real analytic, which shows that Lie groups are not just smooth but actually real analytic manifolds.

Now let us consider a homogeneous space $\mathsf{M} \cong \mathscr{G}/\mathscr{H}$ with Klein pair $(\mathfrak{g}, \mathfrak{h})$. Recall that the identification of $\mathsf{M}$ with $\mathscr{G}/\mathscr{H}$ implies a choice of origin $o \in \mathsf{M}$ (corresponding to the identity coset) with stabiliser $\mathscr{H}$. Let us choose a vector space complement to $\mathfrak{h}$ in $\mathfrak{g}$ and write $\mathfrak{g} = \mathfrak{h} \oplus \mathfrak{m}$. In the reductive case, we can (and will) choose $\mathfrak{m}$ so that $[\mathfrak{h}, \mathfrak{m}] \subset \mathfrak{m}$, but in general this may not be possible. Choosing a basis $\mathsf{Y}_1, \ldots, \mathsf{Y}_m$ for $\mathfrak{m}$ we obtain local coordinates near the origin on $\mathsf{M}$ by $\exp(x^1 \mathsf{Y}_1 + \cdots + x^m \mathsf{Y}_m) \cdot o$ or, having identified $\mathsf{M}$ with $\mathscr{G}/\mathscr{H}$, by $\exp(x^1 \mathsf{Y}_1 + \cdots + x^m \mathsf{Y}_m)\mathscr{H}$. In general we can only hope for local coordinates. Indeed, a

coset representative is a choice of section of the principal $\mathscr{H}$-bundle $\mathscr{G} \to \mathscr{G}/\mathscr{H}$ and principal bundles admit sections if and only if they are trivial.

In effect, what we are doing is choosing a **coset representative** (here $\exp(x^1 Y_1 + \cdots + x^m Y_m) \in \mathscr{G}$) for each coset in $\mathscr{G}/\mathscr{H}$ near the identity coset. This is a locally defined smooth map $M \to \mathscr{G}$, which is only defined in a neighbourhood of the origin, and we may use it to pull back differential forms on $\mathscr{G}$ to M. Every Lie group $\mathscr{G}$ has a distinguished $\mathfrak{g}$-valued one-form $\vartheta \in \Omega^1(\mathscr{G}, \mathfrak{g})$: the left-invariant Maurer–Cartan one-form. If we identify $\mathfrak{g} = T_e G$ with the tangent space at the identity, then $\vartheta_g : T_g \mathscr{G} \to T_e \mathscr{G}$ is simply the differential of left multiplication by $g^{-1}$. We may use a (local) coset representative $L : M \to \mathscr{G}$ to pull back $\vartheta$ to a local one-form $L^* \vartheta$ on M which, for $\mathscr{G}$ a matrix group, has the simpler expression

$$L^* \vartheta = L^{-1} dL. \tag{4.2}$$

Although this is strictly speaking only valid for $\mathscr{G}$ a matrix group, one does not go wrong by assuming we are in a matrix group for calculations provided that in the end we express the final result in a way that makes sense for a general Lie group. For example, it follows from the above expression that $L^* \vartheta$ is left-invariant, since if we multiply $L(x)$ by a constant group element $\mathfrak{g}$ on the left, it remains invariant:

$$(gL)^{-1} d(gL) = L^{-1} g^{-1} g dL = L^{-1} dL. \tag{4.3}$$

Similarly, differentiating again, we find that

$$d(L^{-1} dL) = dL^{-1} \wedge dL = -L^{-1} dL \wedge L^{-1} dL = -\tfrac{1}{2}[L^{-1} dL, L^{-1} dL], \tag{4.4}$$

which is the Maurer–Cartan structure equation. Notice that in the equation above we wrote the term $L^{-1} dL \wedge L^{-1} dL$ which involves matrix multiplication as a commutator $\frac{1}{2}[L^{-1} dL, L^{-1} dL]$, which makes sense (as the Lie bracket in the Lie algebra $\mathfrak{g}$) for $\mathscr{G}$ any group, not necessarily a matrix group.[9]

Suppose that $\mathscr{G}/\mathscr{H}$ is reductive, so that $\mathfrak{g} = \mathfrak{h} \oplus \mathfrak{m}$ with $[\mathfrak{h}, \mathfrak{m}] \subset \mathfrak{m}$. Then we can split the pull-back of the Maurer–Cartan form into its components along $\mathfrak{h}$ and $\mathfrak{m}$:

$$L^{-1} dL = (L^{-1} dL)_{\mathfrak{h}} + (L^{-1} dL)_{\mathfrak{m}} = \omega + \theta, \tag{4.5}$$

where $\omega$, the component along $\mathfrak{h}$ is a connection one-form and $\theta$, the component along $\mathfrak{m}$, is a soldering form. Indeed, under right multiplication by a local $\mathscr{H}$ transformation $L \mapsto Lh^{-1}$,

$$L^{-1} dL \mapsto (Lh^{-1})^{-1} d(Lh^{-1}) = h(L^{-1} dL)h^{-1} + h dh^{-1} = h\omega h^{-1} - dh h^{-1} + h\theta h^{-1}, \tag{4.6}$$

so that comparing the $\mathfrak{h}$ and $\mathfrak{m}$ components, we arrive at

$$\omega \mapsto h\omega h^{-1} - dh h^{-1} \qquad \text{and} \qquad \theta \mapsto h\theta h^{-1}. \tag{4.7}$$

If $\mathscr{G}/\mathscr{H}$ is not reductive, there is no natural split of the Maurer–Cartan one-form. We can still project to $\mathfrak{g}/\mathfrak{h}$ to obtain a soldering form, but there is no uniquely defined component along $\mathfrak{h}$ and, moreover, no such component can be chosen in such a way that results in a connection.

Let us consider in this light the examples in Section 2: the Galilei and Minkowski spacetimes. Both spacetimes are homogeneous spaces of the translation subgroup: in fact, they are principally homogeneous spaces since every point has trivial stabiliser. However we wish to view them as homogeneous spaces of their kinematical Lie groups: the Galilei and Poincaré groups, respectively. Let us start with Minkowski spacetime, which should be more familiar. We work now in general dimension $d + 1$.

4.1.5. *Minkowski spacetime as a homogeneous space.* The Poincaré algebra is given in equation (2.15). Let $\mathfrak{h}$ be the span of $L_{AB}$ and $\mathfrak{m}$ the span of $P_A$, where $A, B = 0, \ldots, d$. Then it follows that the Klein pair $(\mathfrak{g}, \mathfrak{h})$ is reductive. Here we do have a global coset representative $L(x) = \exp(x^A P_A)$, which gives global coordinates to Minkowski spacetime. We can pull-back the Maurer–Cartan form and we get

$$L^{-1} dL = dx^A P_A. \tag{4.8}$$

We see that here $L^{-1} dL$ takes values in $\mathfrak{m}$. This is very special. In general for a reductive Klein pair $(\mathfrak{g}, \mathfrak{h})$, the pull-back of the Maurer–Cartan one-form takes values in $\mathfrak{g}$: so it has an $\mathfrak{h}$-component and an $\mathfrak{m}$-component. The $\mathfrak{h}$-component is a connection one-form whereas the $\mathfrak{m}$-component is a soldering form (i.e., a coframe or an inverse vielbein). Here we see that the connection one-form is absent. We can explain this as follows. The linear isotropy representation of $\mathfrak{h}$ on $\mathfrak{m}$ admits an invariant symmetric inner product $\eta \in \odot^2 \mathfrak{m}^*$, where $\odot$ denotes the symmetric tensor product, with entries $\eta(P_A, P_B) = \eta_{AB}$ of

---

[9]Whereas Ado's theorem says that any finite-dimensional real Lie algebra is a matrix Lie algebra, the similar result for Lie groups is false. The simplest counterexample to the putative Lie group version of Ado's theorem is the universal cover of $SL(2, \mathbb{R})$. See, for example, Graeme Segal's lectures in [69].

lorentzian signature. We can apply this to $L^{-1}dL$ to obtain a Poincaré-invariant metric on Minkowski spacetime:
$$\eta(L^{-1}dL, L^{-1}dL) = \eta_{AB} dx^A dx^B, \tag{4.9}$$
which is nothing else but the standard Minkowski metric in flat coordinates. Of course, relative to flat coordinates, the connection one-form (relative to the coordinate frame) vanishes, which explains why there is no $\mathfrak{h}$-component in $L^{-1}dL$.

The action of the Poincaré group on Minkowski spacetime relative to these coordinates is easy to work out, since it is induced by left multiplication in the Poincaré group. Translations just shift the coordinates:[10]
$$\exp(a^A P_A)\exp(x^A P_A) = \exp((x^A + a^A)P_A), \tag{4.10}$$
so that $\exp(a^A P_A) \cdot (x^0, \ldots, x^d) = (x^0 + a^0, \ldots, x^d + a^d)$. Lorentz transformations act linearly on the coordinates. If $h \in \mathcal{H}$, then
$$h \exp(x^A P_A) = h \exp(x^A P_A)h^{-1}h = \exp(x^A \operatorname{Ad}_h P_A)h, \tag{4.11}$$
where we have introduced the notation Ad for the restriction to $\mathcal{H}$ of the adjoint representation of $\mathcal{G}$. Since $\mathfrak{m}$ is stable under the action of $\mathcal{H}$, $\operatorname{Ad}_h P_A \in \mathfrak{m}$, so we can write it as $\operatorname{Ad}_h P_A = P_B h^B{}_A$ and hence
$$h \exp(x^A P_A) = \exp(h^B{}_A x^A h P_B)h. \tag{4.12}$$
Acting on the "origin" of Minkowski spacetime or on the identity coset $e\mathcal{H} = \mathcal{H}$, we have that
$$h \exp(x^A P_A)\mathcal{H} = \exp(h^B{}_A x^A h P_B)\mathcal{H}, \tag{4.13}$$
using that $h\mathcal{H} = \mathcal{H}$, since $\mathcal{H}$ is a subgroup. Therefore Lorentz transformations in Minkowski spacetime are linear relative to the exponential coordinates. This is a general fact about reductive homogeneous spaces $\mathcal{G}/\mathcal{H}$: in exponential coordinates, $\mathcal{H}$ acts linearly. It is important to realise that this is a coordinate-dependent statement and, moreover, only applies to the reductive situation. It is the linear isotropy representation (on the tangent space at the origin) which, as the name belies, is always linear, regardless of reductivity.

4.1.6. *Galilei spacetime as a homogeneous space.* Let us now consider Galilei spacetime, which is described by a Klein pair $(\mathfrak{g}, \mathfrak{h})$ where $\mathfrak{g}$ is the Lie algebra spanned by $L_{ab}, B_a, P_a, H$, for $a, b = 1, \ldots, d$, and whose brackets are given by equation (2.7) and $\mathfrak{h}$ is the subalgebra spanned by $L_{ab}, B_a$. We choose the reductive complement $\mathfrak{m}$ to be the span of $P_a, H$. We choose exponential coordinates $(t, x^a)$ via the coset representative
$$L(t, x) = \exp(tH + x^a P_a). \tag{4.14}$$
Here again the pull-back of the Maurer–Cartan one-form has no $\mathfrak{h}$-component:
$$L^{-1}dL = dtH + dx^a P_a. \tag{4.15}$$
The action of the Galilei group is again easy to work out using left multiplication: translations again shift the exponential coordinates $\exp(sH + v^a P_a) \cdot (t, x^a) = (t + s, x^a + v^a)$, whereas rotations and boosts act as follows. Let $R \in \mathcal{G}$ be a rotation; that is, an element of the $SO(d)$ subgroup generated by the $L_{ab}$. Then
$$R \exp(tH + x^a P_a) = R \exp(tH + x^a P_a)R^{-1}R = \exp(tH + x^a \operatorname{Ad}_R P_a)R, \tag{4.16}$$
where we have used that $H$ is a scalar and hence commutes with the rotations. Again $\operatorname{Ad}_R P_a = P_b R^b{}_a$ and hence acting on the identity coset we read off the action of rotations on the exponential coordinates: $R \cdot (t, x^a) = (t, R^b{}_a x^a)$. Now let us consider the boosts. Let $h := \exp(v^a B_a)$. Then, as before,
$$h \exp(tH + x^a P_a) = \exp(\operatorname{Ad}_h(tH + x^a P_a))h. \tag{4.17}$$
We work out the term inside the exponential:
$$\operatorname{Ad}_h(tH + x^a P_a) = \operatorname{Ad}_{\exp(v^b B_b)}(tH + x^a P_a) = \exp(v^b \operatorname{ad}_{B_b})(tH + x^a P_a), \tag{4.18}$$
where $\operatorname{ad}_{B_b} H := [B_b, H] = P_b$ and $\operatorname{ad}_{B_b} P_a = [B_b, P_a] = 0$. Substituting, we find
$$h(tH + x^a P_a)h^{-1} = (tH + (x^a + tv^a)P_a). \tag{4.19}$$
Acting on the identity coset again we see that boosts act on the exponential coordinates by $h \cdot (t, x^a) = (t, x^a + tv^a)$, which are precisely the Galilei boosts we saw in Section 2.2.

To determine the Galilei-invariant tensors in Galilei spacetime $\mathcal{G}/\mathcal{H}$, we need to determine the $\mathcal{H}$-invariant tensors of the linear isotropy representation. Canonically dual to the basis $H, P_a$ for $\mathfrak{m}$ we

---

[10]This is not usually so simple, particularly in exponential coordinates the way we have defined them. In some examples, calculations are simpler in modified exponential coordinates where we take product of exponentials instead of a single exponential.

have a basis $\eta, \pi^a$ for $\mathfrak{m}^*$. We need to work out the linear isotropy representation of $\mathscr{H}$ on both $\mathfrak{m}$ and $\mathfrak{m}^*$ and hence on tensors. The linear isotropy representation is such that the $L_{ab}$ generate rotations and $B_a$ generate boosts. It is easy to determine the tensors invariant under rotations. First of all $H \in \mathfrak{m}$ is invariant, but also its dual $\eta$. A classical theorem of Weyl's [70, Theorem 2.11.A] says that every $SO(d)$ invariant tensor of the $d$-dimensional vector representation can be constructed out of $\delta_{ab}$, its inverse and $\epsilon_{a_1 \dots a_d}$. Concerning the boosts, we have that

$$
\begin{aligned}
B_a \cdot H &= P_a & & & B_a \cdot \eta &= 0 \\
B_a \cdot P_b &= 0 & &\text{and} & B_a \cdot \pi^b &= -\delta_a^b \eta,
\end{aligned}
\tag{4.20}
$$

where we have used that the action of $B_a$ on $\mathfrak{m}$ is induced by the restriction of adjoint representation $B_a \cdot X = \mathrm{ad}_{B_a} X = [B_a, X]$ for $X \in \mathfrak{m}$, whereas that on $\mathfrak{m}^*$ is induced by the restriction of the coadjoint representation $B_a \cdot \alpha = \mathrm{ad}_{B_a}^* \alpha = -\alpha \circ \mathrm{ad}_{B_a}$ for $\alpha \in \mathfrak{m}^*$. We see that $\eta \in \mathfrak{m}^*$ is $\mathscr{H}$-invariant and so is $\delta^{ab} P_a \otimes P_b$. Applying $\eta$ to the pull-back of the Maurer–Cartan one-form we obtain a Galilei-invariant one-form on Galilei spacetime, namely, the clock one-form

$$
\tau := \eta(L^{-1} dL) = \eta(dt H + dx^a P_a) = dt.
\tag{4.21}
$$

The vielbein dual to the soldering form $dt H + dx^a P_a$ is given by $\frac{\partial}{\partial t}$ and $\frac{\partial}{\partial x^a}$. The Galilei-invariant tensor field corresponding to $\delta^{ab} P_a \otimes P_b$ is then $\delta^{ab} \frac{\partial}{\partial x^a} \otimes \frac{\partial}{\partial x^b}$, which is the spatial cometric on Galilei spacetime we saw in Section 2.2.

4.1.7. *Summary.* We may summarise the above discussion as follows:

- Homogeneous spaces of a group $\mathscr{G}$ are described infinitesimally by a Klein pair $(\mathfrak{g}, \mathfrak{h})$ where $\mathfrak{g}$ is the Lie algebra of $\mathscr{G}$ and $\mathfrak{h}$ a Lie subalgebra generating a closed subgroup $\mathscr{H}$ of $\mathscr{G}$. The homogeneous space can be identified with the coset space $\mathscr{G}/\mathscr{H}$ consisting of left $\mathscr{H}$-cosets $g\mathscr{H}$ in $\mathscr{G}$. The action of $\mathscr{G}$ on $\mathscr{G}/\mathscr{H}$ is induced from left multiplication on $\mathscr{G}$.
- As a vector space, $\mathfrak{g} = \mathfrak{h} \oplus \mathfrak{m}$, where if possible we choose $\mathfrak{m}$ in such a way that $[\mathfrak{h}, \mathfrak{m}] \subset \mathfrak{m}$. If this is possible, we say that $(\mathfrak{g}, \mathfrak{h})$ is reductive.
- Every choice of basis $X_1, \dots, X_m$ for $\mathfrak{m}$ gives rise to exponential coordinates near the identity coset of $\mathscr{G}/\mathscr{H}$ corresponding to a (locally defined) coset representative $L : \mathscr{G}/\mathscr{H} \to \mathscr{G}$, where $L(x) = \exp(x^1 X_1 + \cdots x^m X_m)$. The action of $\mathscr{G}$ on the exponential coordinates can be calculated in principle simply by left multiplying in $\mathscr{G}$: $gL(x) = L(g \cdot x)h(g, x)$ for some $h(g, x) \in \mathscr{H}$. In the reductive case, the action of $h \in \mathscr{H}$ on the exponential coordinates is linear.
- We can use the coset representative to pull-back the Maurer–Cartan one-form to $\mathscr{G}/\mathscr{H}$. This results in a locally defined one-form with values in $\mathfrak{g}$. In the reductive case, it decomposes into an $\mathfrak{h}$-connection and a soldering form. In the non-reductive case, the $\mathfrak{h}$-component is *not* a connection, but the projection to $\mathfrak{g}/\mathfrak{h}$ is still a soldering form.
- In the reductive case, the representation of $\mathscr{H}$ on $\mathfrak{m}$ is called the the linear isotropy representation. In the non-reductive case, the linear isotropy representation is carried by the quotient vector space $\mathfrak{g}/\mathfrak{h}$. In practice we work with $\mathfrak{g}/\mathfrak{h}$ by calculating brackets in $\mathfrak{g}$ and then dropping from the RHS anything belonging to $\mathfrak{h}$.
- $\mathscr{G}$-invariant tensor fields on $\mathscr{G}/\mathscr{H}$ are in one-to-one correspondence with $\mathscr{H}$-invariant tensors on $\mathfrak{g}/\mathfrak{h}$ (or on $\mathfrak{m}$ in the reductive situation). If $\mathscr{H}$ is connected, this is the same as $\mathfrak{h}$-invariant tensors, which are typically simple to determine, at least if of small rank.

4.2. **Homogeneous kinematical spacetimes.** A homogeneous kinematical spacetime is a homogeneous space of a kinematical group of the right dimension. Recall from Definition 1 (but now for arbitrary dimension) that a kinematical Lie algebra for $(d + 1)$-dimensional spacetimes consists of a subalgebra $\mathfrak{r} \cong \mathfrak{so}(d)$ with generators $L_{ab}$, two copies of the vector representation with generators $B_a, P_a$ and an additional scalar generator $H$. Suppose that $\mathfrak{g}$ is such a kinematical Lie algebra. A kinematical Klein pair for a $(d + 1)$-dimensional spacetime takes the form $(\mathfrak{g}, \mathfrak{h})$, where $\mathfrak{h} \subset \mathfrak{g}$ is a Lie subalgebra spanned by $L_{ab}$ and $V_a = \alpha B_a + \beta P_a$ for some $\alpha, \beta \in \mathbb{R}$.

The determination of such Klein pairs was done in [40], whose results we summarise in Table 5, where we have excluded some spacetimes which only exist for $d = 1, 2$. We have chosen a basis for $\mathfrak{g}$ in such a way that $\mathfrak{h}$ is always spanned by $L_{ab}$ and (the new) $B_a$. This facilitates comparison of the different homogeneous spacetimes, but also obscures the isomorphisms between some of the Lie algebras. For example, the kinematical Lie algebras for Minkowski (M) and anti de Sitter–Carroll (AdSC) spacetimes are isomorphic to the Poincaré algebra. The explicit isomorphism is the identity on the rotation and time-translation generators, but exchanges boosts and spatial momenta:

$$
\mathbf{L}^{\mathsf{M}} = \mathbf{L}^{\mathsf{AdSC}}, \quad \mathsf{H}^{\mathsf{M}} = \mathsf{H}^{\mathsf{AdSC}}, \quad \mathbf{B}^{\mathsf{M}} = \mathbf{P}^{\mathsf{AdSC}} \quad \text{and} \quad \mathbf{P}^{\mathsf{M}} = -\mathbf{B}^{\mathsf{AdSC}}.
\tag{4.22}
$$

This illustrates the need to specify a geometric realisation before assigning a physical/geometric meaning to the generators of a kinematical Lie algebra, since what is a translation in Minkowski spacetime is a carrollian boost in anti de Sitter–Carroll.

Similarly, the kinematical Lie algebras for hyperbolic space (H), de Sitter spacetime (dS) and the lightcone (LC) are isomorphic (to the Lorentz algebra in one dimension higher). The explicit isomorphisms are again the identity on the rotation and time-translation generators:

$$\mathbf{L}^{\mathsf{dS}} = \mathbf{L}^{\mathsf{H}} = \mathbf{L}^{\mathsf{LC}} \quad \text{and} \quad \mathsf{H}^{\mathsf{dS}} = \mathsf{H}^{\mathsf{H}} = \mathsf{H}^{\mathsf{LC}}, \tag{4.23}$$

but now

$$\mathbf{P}^{\mathsf{dS}} = \mathbf{B}^{\mathsf{H}} \quad \text{and} \quad \mathbf{B}^{\mathsf{dS}} = -\mathbf{P}^{\mathsf{H}}, \tag{4.24}$$

and

$$\mathbf{B}^{\mathsf{LC}} = \tfrac{1}{\sqrt{2}}(\mathbf{B}^{\mathsf{dS}} - \mathbf{P}^{\mathsf{dS}}) = -\tfrac{1}{\sqrt{2}}(\mathbf{B}^{\mathsf{H}} + \mathbf{P}^{\mathsf{H}}) \quad \text{and} \quad \mathbf{P}^{\mathsf{LC}} = \tfrac{1}{\sqrt{2}}(\mathbf{B}^{\mathsf{dS}} + \mathbf{P}^{\mathsf{dS}}) = \tfrac{1}{\sqrt{2}}(\mathbf{B}^{\mathsf{H}} - \mathbf{P}^{\mathsf{H}}). \tag{4.25}$$

TABLE 5. Homogeneous $(\mathsf{d}+1)$-dimensional (spatially isotropic) kinematical spacetimes

| Name | Klein pair | Nonzero Lie brackets in addition to $[\mathbf{L},\mathbf{L}]=\mathbf{L}$, $[\mathbf{L},\mathbf{B}]=\mathbf{B}$, $[\mathbf{L},\mathbf{P}]=\mathbf{P}$ | | | | |
|---|---|---|---|---|---|---|
| Minkowski | $(\mathfrak{iso}(d,1),\mathfrak{so}(d,1))$ | $[\mathsf{H},\mathbf{B}]=-\mathbf{P}$ | | $[\mathbf{B},\mathbf{B}]=\mathbf{L}$ | $[\mathbf{B},\mathbf{P}]=\mathsf{H}$ | |
| de Sitter | $(\mathfrak{so}(d+1,1),\mathfrak{so}(d,1))$ | $[\mathsf{H},\mathbf{B}]=-\mathbf{P}$ | $[\mathsf{H},\mathbf{P}]=-\mathbf{B}$ | $[\mathbf{B},\mathbf{B}]=\mathbf{L}$ | $[\mathbf{B},\mathbf{P}]=\mathsf{H}$ | $[\mathbf{P},\mathbf{P}]=-\mathbf{L}$ |
| anti de Sitter | $(\mathfrak{so}(d,2),\mathfrak{so}(d,1))$ | $[\mathsf{H},\mathbf{B}]=-\mathbf{P}$ | $[\mathsf{H},\mathbf{P}]=\mathbf{B}$ | $[\mathbf{B},\mathbf{B}]=\mathbf{L}$ | $[\mathbf{B},\mathbf{P}]=\mathsf{H}$ | $[\mathbf{P},\mathbf{P}]=\mathbf{L}$ |
| euclidean | $(\mathfrak{iso}(d+1),\mathfrak{so}(d+1))$ | $[\mathsf{H},\mathbf{B}]=\mathbf{P}$ | | $[\mathbf{B},\mathbf{B}]=-\mathbf{L}$ | $[\mathbf{B},\mathbf{P}]=\mathsf{H}$ | |
| sphere | $(\mathfrak{so}(d+2),\mathfrak{so}(d+1))$ | $[\mathsf{H},\mathbf{B}]=\mathbf{P}$ | $[\mathsf{H},\mathbf{P}]=-\mathbf{B}$ | $[\mathbf{B},\mathbf{B}]=-\mathbf{L}$ | $[\mathbf{B},\mathbf{P}]=\mathsf{H}$ | $[\mathbf{P},\mathbf{P}]=-\mathbf{L}$ |
| hyperbolic | $(\mathfrak{so}(d+1,1),\mathfrak{so}(d+1))$ | $[\mathsf{H},\mathbf{B}]=\mathbf{P}$ | $[\mathsf{H},\mathbf{P}]=\mathbf{B}$ | $[\mathbf{B},\mathbf{B}]=-\mathbf{L}$ | $[\mathbf{B},\mathbf{P}]=\mathsf{H}$ | $[\mathbf{P},\mathbf{P}]=\mathbf{L}$ |
| Galilei | $(\mathfrak{g},\mathfrak{iso}(d))$ | $[\mathsf{H},\mathbf{B}]=-\mathbf{P}$ | | | | |
| de Sitter–Galilei | $(\mathfrak{n}^+_{\gamma=-1},\mathfrak{iso}(d))$ | $[\mathsf{H},\mathbf{B}]=-\mathbf{P}$ | $[\mathsf{H},\mathbf{P}]=-\mathbf{B}$ | | | |
| torsional de Sitter–Galilei | $(\mathfrak{n}^+_{\gamma\in(-1,1)},\mathfrak{iso}(d))$ | $[\mathsf{H},\mathbf{B}]=-\mathbf{P}$ | $[\mathsf{H},\mathbf{P}]=\gamma\mathbf{B}+(1+\gamma)\mathbf{P}$ | | | |
| torsional de Sitter–Galilei | $(\mathfrak{n}^0,\mathfrak{iso}(d))$ | $[\mathsf{H},\mathbf{B}]=-\mathbf{P}$ | $[\mathsf{H},\mathbf{P}]=\mathbf{B}+2\mathbf{P}$ | | | |
| anti de Sitter–Galilei | $(\mathfrak{n}^-_{\chi=0},\mathfrak{iso}(d))$ | $[\mathsf{H},\mathbf{B}]=-\mathbf{P}$ | $[\mathsf{H},\mathbf{P}]=\mathbf{B}$ | | | |
| torsional anti de Sitter–Galilei | $(\mathfrak{n}^-_{\chi>0},\mathfrak{iso}(d))$ | $[\mathsf{H},\mathbf{B}]=-\mathbf{P}$ | $[\mathsf{H},\mathbf{P}]=(1+\chi^2)\mathbf{B}+2\chi\mathbf{P}$ | | | |
| Carroll | $(\mathfrak{c},\mathfrak{iso}(d))$ | | | | $[\mathbf{B},\mathbf{P}]=\mathsf{H}$ | |
| de Sitter–Carroll | $(\mathfrak{iso}(d+1),\mathfrak{iso}(d))$ | | $[\mathsf{H},\mathbf{P}]=-\mathbf{B}$ | | $[\mathbf{B},\mathbf{P}]=\mathsf{H}$ | $[\mathbf{P},\mathbf{P}]=-\mathbf{L}$ |
| anti de Sitter–Carroll | $(\mathfrak{iso}(d,1),\mathfrak{iso}(d))$ | | $[\mathsf{H},\mathbf{P}]=\mathbf{B}$ | | $[\mathbf{B},\mathbf{P}]=\mathsf{H}$ | $[\mathbf{P},\mathbf{P}]=\mathbf{L}$ |
| lightcone | $(\mathfrak{so}(d+1,1),\mathfrak{iso}(d))$ | $[\mathsf{H},\mathbf{B}]=\mathbf{B}$ | $[\mathsf{H},\mathbf{P}]=-\mathbf{P}$ | | $[\mathbf{B},\mathbf{P}]=\mathsf{H}+\mathbf{L}$ | |

Table 5 is divided into sections corresponding to the class of geometry the spacetime describes: lorentzian, riemannian, galilean and carrollian. They can be distinguished by the type of invariant tensor fields or, as explained in Section 4.1.2, by the $\mathfrak{h}$-invariant tensors of the linear isotropy representation on $\mathfrak{g}/\mathfrak{h}$. We shall now go through the table in some detail. Further details can be found in [40, 41].

4.2.1. *Homogeneous lorentzian spacetimes.* The first section of the table consists of those homogeneous kinematical spacetimes admitting an invariant lorentzian metric. They admit an $\mathfrak{h}$-invariant lorentzian inner product on $\mathfrak{g}/\mathfrak{h}$. The dimension of the kinematical Lie group for a $(\mathsf{d}+1)$-dimensional homogeneous spacetime is $\frac{1}{2}(\mathsf{d}+1)(\mathsf{d}+2)$, hence if the kinematical Lie group is acting via isometries, the geometry is maximally symmetric. In lorentzian signature they are Minkowski, de Sitter and anti de Sitter spacetimes. Geometrically, there is a one-parameter (the scalar curvature) family of both de Sitter and anti de Sitter spacetimes, but as homogeneous spacetimes they are isomorphic. The parameter is simply the scale of the invariant metric. These spacetimes are symmetric spaces and the stabiliser subalgebra $\mathfrak{h} \cong \mathfrak{so}(\mathsf{d},1)$ in all cases.

4.2.2. *Homogeneous riemannian "spacetimes".* The second section of the table consists of homogeneous spaces of kinematical Lie groups which admit an invariant riemannian metric. They can hardly be considered as spacetimes, so we will not mention them again. The same dimension arguments as for the lorentzian spacetimes imply that these riemannian homogeneous spaces are maximally symmetric, so they are the euclidean and hyperbolic spaces and the round sphere. Again the curvature of the sphere and hyperbolic space is a choice of additional structure on the homogeneous spaces. All round spheres are described by the same Klein pair $(\mathfrak{so}(\mathsf{d}+2),\mathfrak{so}(\mathsf{d}+1))$, for example. These riemannian spaces are symmetric spaces and the stabiliser subalgebra $\mathfrak{h} \cong \mathfrak{so}(\mathsf{d}+1)$ in all cases.

4.2.3. *Homogeneous galilean spacetimes.* The third section of the table consists of homogeneous spaces of kinematical Lie groups admitting an invariant galilean structure: a clock one-form and a spatial cometric. The clock one-form comes from an $\mathfrak{h}$-invariant covector in $(\mathfrak{g}/\mathfrak{h})^*$, the dual of the linear isotropy representation. The spatial cometric comes from an $\mathfrak{h}$-invariant symmetric bivector in $\odot^2(\mathfrak{g}/\mathfrak{h})$, the symmetric square of the linear isotropy representation. Apart from Galilei spacetime, discussed in 4.1.6, which is the non-relativistic limit of Minkowski spacetime, there are two one-parameter families of spacetimes. One family is the de Sitter–Galilei family with parameter $\gamma \in [-1, 1]$. For $\gamma = -1$, it is the non-relativistic limit of de Sitter spacetimes and hence a symmetric space associated to one of the Newton–Hooke algebras. For any $\gamma \in (-1, 1)$, the spacetime is reductive but not symmetric and associated to the kinematical Lie algebra $\mathfrak{n}_\gamma^+$. The notation notwithstanding, the spacetime with $\gamma = 1$ is not associated to $\mathfrak{n}_{\gamma=1}^+$ but instead to the Lie algebra $\mathfrak{n}^0$, which is obtained as a (singular) limit $\lim_{\gamma \to 1} \mathfrak{n}_\gamma^+$. This limit is analogous to a contraction, but it is not a contraction in that the Lie algebra $\mathfrak{n}_\gamma^+$ are not isomorphic for different values of $\gamma \in [-1, 1]$. The canonical invariant connection (see, e.g., [71]) has torsion proportional to $1 + \gamma$ and hence the spacetimes for $\gamma \neq -1$ may be thought of as torsional de Sitter–Galilei spacetimes. The other family is the anti de Sitter–Galilei family with parameter $\chi \geqslant 0$. For $\chi = 0$, it is the non-relativistic limit of anti de Sitter spacetime and hence a symmetric space, associated to the other Newton–Hooke algebra. For any $\chi > 0$, it is a reductive, non-symmetric homogeneous spacetime. Again the canonical invariant connection has torsion (proportional to $\chi$) and these spacetimes are therefore called torsional anti de Sitter–Galilei spacetimes. The limit $\chi \to \infty$ of the torsional anti de Sitter–Galilei spacetimes coincides with the limit $\gamma \to 1$ of the torsional de Sitter–Galilei spacetime. In all cases, the stabiliser subalgebra $\mathfrak{h} \cong \mathfrak{iso}(d)$.

A final remark is that the torsional de Sitter–Galilei spacetime with $\gamma = 0$ is isomorphic to Galilei spacetime as a (weak) Newton–Cartan geometry: both arise as null reductions of Minkowski spacetime, but they are homogeneous spaces of non-isomorphic kinematical Lie groups and hence not isomorphic as homogeneous spacetimes.

4.2.4. *Homogeneous carrollian spacetimes.* The fourth section of the table consists of four homogeneous kinematical spacetimes admitting an invariant carrollian structure: a nowhere-vanishing vector field and a "spatial metric". The vector field comes from an invariant vector in the linear isotropy representation $\mathfrak{g}/\mathfrak{h}$ and the spatial metric comes from an invariant symmetric bilinear form in $\odot^2(\mathfrak{g}/\mathfrak{h})^*$. Three of these carrollian spacetimes are the ultra-relativistic limits of the lorentzian spacetimes in the Table: Carroll spacetime (of Minkowski) and de Sitter–Carroll and anti de Sitter–Carroll spacetimes (of de Sitter and anti de Sitter, respectively). They are symmetric spaces. The fourth carrollian spacetime is the lightcone in Minkowski spacetime one dimension higher. It is the only non-reductive homogeneous spacetime in the table. In all cases, the stabiliser subalgebra $\mathfrak{h} \cong \mathfrak{iso}(d)$, but despite being abstractly isomorphic to the stabiliser subalgebra of the galilean spacetimes, their images under the linear isotropy representation are not conjugate subalgebras of $\mathfrak{gl}(\mathfrak{g}/\mathfrak{h})$, which explains why they have different invariants and hence why the geometries are different: carrollian instead of galilean.

4.2.5. *Homogeneous aristotelian spacetimes.* Although not in the Table, there are also aristotelian spacetimes which are homogeneous spaces of Lie groups of the aristotelian Lie algebras in Table 2. The stabiliser subalgebra is always the rotational subalgebra $\mathfrak{r} \cong \mathfrak{so}(d)$ and hence there is a unique Klein pair for each aristotelian Lie algebra. In the order given in Table 2, they are the static aristotelian spacetime, the torsional static aristotelian spacetime and the product of the round $d$-dimensional sphere or $d$-dimensional hyperbolic space with the real line. All are reductive and all but the torsional static spacetime, whose canonical connection has torsion, are symmetric.

4.3. **Non-lorentzian geometries.** As we saw in Section 4.2, the homogeneous spatially isotropic kinematical spacetimes come in several families depending on their invariant tensors. We shall ignore the riemannian case in what follows, since they do not admit an interpretation as spacetimes (e.g., the boosts are actually rotations). We shall now describe the (Cartan) geometries modelled on the homogeneous spacetimes. It turns out that all of the geometries we consider: lorentzian, galilean, carrollian and aristotelian are examples of G-structures; that is, they are defined by distinguished vielbeins transforming under a subgroup of the general linear group. The prime example is lorentzian geometry, where the distinguished vielbeins transform under local Lorentz transformations on overlaps and can subsequently be interpreted as the (pseudo) orthonormal frames relative to a lorentzian metric.

4.3.1. *Basic notions about G-structures.* Now consider an $n$-dimensional manifold $M$. Let $p \in M$. A **frame at** $p$ is an isomorphism $u : \mathbb{R}^n \to T_p M$ of vector spaces. The images under $u$ of the standard basis $(e_1, \ldots, e_n)$ of $\mathbb{R}^n$ give a basis $(u(e_1), \ldots, u(e_n))$ for the tangent space at $p$. If $u, u'$ are two frames

at $\mathfrak{p}$, then $h := u^{-1} \circ u' \in GL(n, \mathbb{R})$, which we may rewrite as $u' = u \circ h$. This defines a right action of $GL(n, \mathbb{R})$ on the set $F_\mathfrak{p}$ of frames at $\mathfrak{p}$, which is free (if $u \circ h = u$, then $h$ is the identity) and transitive (any two frames are related by some $h \in GL(n, \mathbb{R})$). The disjoint union $F(M) = \bigsqcup_{\mathfrak{p} \in M} F_\mathfrak{p}$ is the total space of the **frame bundle** of $M$: a smooth right principal $GL(n, \mathbb{R})$-bundle, whose (local) sections are called **moving frames** or **vielbeins**.

Let $\mathscr{G} \subset GL(n, \mathbb{R})$ be a Lie subgroup. A $\mathscr{G}$-**structure** on $M$ is a principal $\mathscr{G}$-subbundle $P \subset F(M)$ of the frame bundle: this amounts to restricting to a collection of frames such for any two frames $u, u'$ at $\mathfrak{p}$ in this collection, $u^{-1} \circ u' \in \mathscr{G}$. The existence of a $\mathscr{G}$-structure is not guaranteed: there are topological obstructions. For example, if $(M, g)$ is a Lorentzian manifold, we can always pick pseudo-orthonormal frames and it follows that if $u, u'$ are pseudo-orthonormal frames at $\mathfrak{p}$, $u^{-1} \circ u' \in O(d, 1) \subset GL(d + 1, \mathbb{R})$. The Lorentz group $O(d, 1)$ is not connected: it has four connected components: depending on whether or not the temporal or spatial orientations are preserved. This then leads to topological obstructions (temporal and/or spatial orientability) to further reduce the structure group from $O(d, 1)$ to its connected component $SO(d, 1)_0$ (i.e., the proper orthochronous Lorentz group) or some group in between.

Associated to every $\mathscr{G}$-structure $\pi : P \to M$ there is a **soldering form** $\theta \in \Omega^1(P; \mathbb{R}^n)$, which is a $\mathbb{R}^n$-valued one-form on the total space $P$. Let $u$ be a frame at $\mathfrak{p}$ and suppose that $X_u \in T_u P$ is a tangent vector to $P$ at $u$. Then $\theta_u(X_u) = u^{-1}(\pi_* X_u)$. In other words, $\theta_u(X_u)$ is the coordinate vector of $\pi_* X_u \in T_\mathfrak{p} M$ relative to the frame $u$. Given a vielbein (i.e., a local section $s : U \to P$ on some open subset $U \subset M$) we can use it to pull back $\theta$ to $U$. Since $\theta$ is $\mathbb{R}^n$-valued, so is its pull-back and we may write it as a linear combination of the standard basis of $\mathbb{R}^n$: $s^* \theta = \vartheta^i e_i$ for some one-forms $\vartheta^i$ defined only on $U$. Then $(\vartheta^1, \ldots, \vartheta^n)$ is often called the *inverse vielbein*, but of course it is simply the canonically dual coframe to the vielbein.

The soldering form is the fundamental object which allows to relate the representation theory of $\mathscr{G}$ to the geometry of any manifold with a $\mathscr{G}$-structure. For more details about this in the context of non-lorentzian geometry, please see [72, Section 2].

4.3.2. *Lorentzian geometry.* Let us see how these ideas play out in the familiar case of lorentzian geometry.

Let $(\mathfrak{g} = \mathfrak{h} \oplus \mathfrak{m}, \mathfrak{h})$ be a reductive Klein pair for any one of the homogeneous lorentzian manifolds in Table 5. In all cases, $\mathfrak{h} \cong \mathfrak{so}(d, 1)$ and the infinitesimal linear isotropy representation $\lambda : \mathfrak{h} \to \mathfrak{gl}(\mathfrak{m})$ preserves a lorentzian inner product $\eta$, say, on $\mathfrak{m}$; that is, for all $X \in \mathfrak{h}$ and $Y_1, Y_2 \in \mathfrak{m}$, we have that

$$\eta(\lambda_X Y_1, Y_2) + \eta(Y_1, \lambda_X Y_2) = 0. \tag{4.26}$$

Every choice of basis for $\mathfrak{m}$ defines an isomorphism $\mathfrak{m} \to \mathbb{R}^{d+1}$ which may be used to transport the lorentzian inner product to $\mathbb{R}^{d+1}$. Choosing a pseudo-orthonormal basis for $\mathfrak{m}$ brings the inner product on $\mathbb{R}^{d+1}$ to be the standard one with diagonal matrix with entries $(-1, 1, \ldots, 1)$ and embeds $\mathfrak{h} \subset \mathfrak{gl}(d + 1, \mathbb{R})$ as the standard Lorentz algebra.

Let $M$ be a $(d + 1)$-dimensional manifold with a $G = O(d, 1)$-structure. Then $M$ is covered by open subsets $\{U_\alpha\}$ and each $U_\alpha$ we have an inverse vielbein $\vartheta_\alpha$ taking values in $\mathbb{R}^{d+1}$ and such that on nonempty overlaps $U_{\alpha\beta} := U_\alpha \cap U_\beta$, the inverse vielbeins are related by local $O(d, 1)$-transformations $h_{\alpha\beta} : U_{\alpha\beta} \to O(d, 1)$. Using the $O(d, 1)$-invariant lorentzian inner product $\eta$ on $\mathbb{R}^{d+1}$, we can define on each $U_\alpha$ a local lorentzian metric

$$g_\alpha := \eta(\vartheta_\alpha, \vartheta_\alpha). \tag{4.27}$$

But because $\eta$ is $O(d, 1)$ invariant, these local metrics agree on overlaps and hence they glue to a lorentzian metric $g$ on $M$. This shows that a lorentzian metric on an $(d + 1)$-dimensional manifold $M$ is equivalent to a $\mathscr{G}$-structure on $M$ with $G = O(d, 1)$.

This generalises in the sense that if an $n$-dimensional manifold $M$ admits a $\mathscr{G}$-structure with $\mathscr{G} \subset GL(n, \mathbb{R})$, then every (nonzero) $\mathscr{G}$-invariant tensor of $\mathbb{R}^n$ gives rise to a (nowhere-vanishing) global tensor field on $M$: it is defined locally using the (inverse) vielbeins, but the $\mathscr{G}$-invariance guarantees that these local tensor fields glue on overlaps.

4.3.3. *Newton–Cartan geometry.* Let $(\mathfrak{g} = \mathfrak{h} \oplus \mathfrak{m}, \mathfrak{h})$ be a reductive Klein pair for any one of the homogeneous galilean manifolds in Table 5. In all cases, $\mathfrak{h} \cong \mathfrak{iso}(d)$ and the infinitesimal linear isotropy representation $\lambda : \mathfrak{h} \to \mathfrak{gl}(\mathfrak{m})$ preserves a covector in $\mathfrak{m}^*$ and a symmetric bivector in $\odot^2 \mathfrak{m}$. Choose basis $(P_0, P_1, \ldots, P_d)$ for $\mathfrak{m}$ and canonical dual basis $\pi^0, \pi^1, \ldots, \pi^d$ for $\mathfrak{m}^*$, relative to which the invariant covector is $\pi^0$ and the invariant symmetric bivector is $P_1^2 + \cdots + P_d^2 = \delta^{ab} P_a P_b$. The subgroup $\mathscr{G} \subset GL(d + 1, \mathbb{R})$ which preserves these tensors consists of matrices of the form

$$\begin{pmatrix} 1 & \mathbf{0}^\mathsf{T} \\ \mathbf{v} & A \end{pmatrix} \qquad \text{with} \qquad \mathbf{v} \in \mathbb{R}^d, A \in O(d). \tag{4.28}$$

A **(weak) Newton–Cartan structure** on a $(d+1)$-manifold $M$ is a $\mathscr{G}$-structure with $\mathscr{G} \subset \mathrm{GL}(d+1,\mathbb{R})$ the subgroup given by the matrices in equation (4.28). The $\mathscr{G}$-invariant tensors give rise to global (nowhere-vanishing) tensor fields on $M$: the clock one-form $\tau \in \Omega^1(M)$ defined locally on $U_\alpha$ by $\tau_\alpha := \pi^0(\vartheta_\alpha)$ relative to the inverse vielbein $\vartheta_\alpha$. Similarly the spatial cometric is given locally on $U_\alpha$ by $\lambda_\alpha := \delta^{ab}(E_\alpha)_a(E_\alpha)_b$, where $E_\alpha$ is the vielbein dual to $\vartheta_\alpha$. These symmetric bivectors glue to give a symmetric $(2,0)$-tensor field $\lambda$ on $M$. Equivalently, one could define a (weak) Newton–Cartan structure on $M$ by specifying a nowhere vanishing one-form $\tau \in \Omega^1(M)$ and a corank-1 positive-semidefinite symmetric $(2,0)$-tensor field $\lambda \in \Gamma(\odot^2 TM)$ with the property that $\lambda(\tau,-) = 0$.

A **Newton–Cartan structure** is obtained by enhancing a weak Newton–Cartan structure with an **adapted connection**: an affine connection relative to which $\tau$ and $\lambda$ are parallel. Such connections were studied initially in [73,74]. As explained, e.g., in [72, Section 2], every $\mathscr{G}$-structure has an *intrinsic torsion* which is the part of the torsion tensor of an adapted connection which is independent of the connection.

This is not something one is familiar with from lorentzian geometry, since the Fundamental Theorem of lorentzian (or, more generally, pseudo-riemannian) geometry states that there exists a unique torsion-free adapted (here, metric) connection. So that intrinsic torsion of a lorentzian geometry is always zero.

However for a Newton–Cartan structure this is not the case. As first shown in [74], the intrinsic torsion of a Newton–Cartan connection can be identified with $d\tau \in \Omega^2(M)$, for $\tau$ the clock-one form. Hence the intrinsic torsion need not *a priori* be zero. Also shown in [73,74] is that specifying the torsion does not uniquely determine the adapted connection: there is contorsion, which is measured by an arbitrary two-form.

A study of how the bundle of two-forms decomposes under the action of the structure group reveals that there are three[11] classes of Newton–Cartan structures [72, Theorem 6]:

- **torsionless** (NC): $d\tau = 0$;
- **twistless torsional** (TTNC): $d\tau \wedge \tau = 0$; and
- **torsional** (TNC): $d\tau \wedge \tau \neq 0$.

These classes first appeared in [75] (see Table I in that paper) in the context of Lifshitz holography.

The homogeneous examples in Table 5 are all such that $d\tau = 0$, but there are homogeneous examples of all three kinds [76].

A rich source of (weak) Newton–Cartan structures arise as null reductions of lorentzian manifolds [60,77]. Let $(N, g)$ be a lorentzian manifold with a null nowhere-vanishing Killing vector $\xi$ and suppose that $\xi$ is complete so that it integrates to a one-parameter $\Gamma$ subgroup of isometries of $N$. Let us assume that the action of $\Gamma$ on $N$ is such that the quotient $M := N/\Gamma$ is smooth making the projection $\pi : N \to M$ into a smooth submersion. Then $M$ inherits from $N$ a (weak) Newton–Cartan structure as follows. The Killing one-form $\xi^\flat$ dual to $\xi$ is the pull-back via $\pi$ of a clock one-form $\tau \in \Omega^1(M)$

$$\xi^\flat = \pi^* \tau, \tag{4.29}$$

which is nowhere vanishing since $\xi$ is. We define the ruler $\lambda$ as follows. Clearly it is enough to know what $\lambda(\alpha, \beta)$ is for any two one-forms $\alpha, \beta \in \Omega^1(M)$. Give two such one-forms $\alpha, \beta$, let $X_\alpha, X_\beta \in \mathscr{X}(N)$ be vector fields on $N$ which are metrically dual to the pull-backs $\pi^* \alpha, \pi^* \beta \in \Omega^1(N)$. Then $\lambda(\alpha, \beta)$ is the function on $M$ whose pull-back to $N$ agrees with the inner product $g(X_\alpha, X_\beta)$. It follows that $\lambda(\tau, -) = 0$ and hence that $(M, \tau, \lambda)$ is a (weak) Newton–Cartan structure.

4.3.4. *Carrollian geometry.* Not all carrollian spacetimes in Table 5 are reductive: the lightcone is not. So in order to treat all cases together we will be working with a Klein pair $(\mathfrak{g}, \mathfrak{h})$ and simply define the infinitesimal linear isotropy representation $\lambda : \mathfrak{h} \to \mathfrak{gl}(\mathfrak{g}/\mathfrak{h})$. In all cases, $\mathfrak{h} \cong \mathfrak{iso}(d)$, but this is a different (i.e., non-conjugate) Lie subalgebra of $\mathfrak{gl}(\mathfrak{g}/\mathfrak{h})$ than the one in the galilean examples. This means that $\mathfrak{g}/\mathfrak{h}$ has different $\mathfrak{h}$-invariant tensors in this case. The $\mathfrak{h}$-invariant tensors are now a vector in $\mathfrak{g}/\mathfrak{h}$ and a symmetric bilinear form in $\odot^2(\mathfrak{g}/\mathfrak{h})^*$. We can choose basis $(\overline{P}_0, \overline{P}_1, \ldots, \overline{P}_d)$ for $\mathfrak{g}/\mathfrak{h}$, where $\overline{P}_A = P_A$ mod $\mathfrak{h}$, and canonical dual basis $(\pi^0, \pi^1, \ldots, \pi^d)$ for $(\mathfrak{g}/\mathfrak{h})^*$, relative to which the invariant tensors are $P_0$ and $\delta_{ab}\pi^a\pi^b = (\pi^1)^2 + \cdots + (\pi^d)^2$. The subgroup $\mathscr{G} \subset \mathrm{GL}(d+1, \mathbb{R})$ which preserves these two tensors consists of matrices of the form

$$\begin{pmatrix} 1 & \boldsymbol{v}^\mathsf{T} \\ \boldsymbol{0} & A \end{pmatrix} \qquad \text{with} \qquad \boldsymbol{v} \in \mathbb{R}^d, A \in O(d). \tag{4.30}$$

This group is abstractly isomorphic to the one with matrices (4.28), but of course they are not conjugate in $\mathrm{GL}(d+1, \mathbb{R})$ since they have different invariants. The connected component of the group $\mathscr{G}$ (where $A \in \mathrm{SO}(d)$) also leaves invariant $\pi^0 \wedge \pi^1 \wedge \cdots \wedge \pi^d \in \wedge^{d+1}(\mathfrak{g}/\mathfrak{h})^*$.

---

[11]This is in generic dimension $d + 1$: if $d = 1$ then there are only two classes and if $d = 4$ and assuming that $M$ is orientable, there are five classes. See [72, Appendix B].

A **(weak) carrollian structure** on a $(d+1)$-dimensional manifold $M$ is a $\mathscr{G}$-structure with $\mathscr{G} \subset \mathrm{GL}(d+1, \mathbb{R})$ the subgroup consisting of the matrices in equation (4.30). The $\mathscr{G}$-invariant tensors give rise to a (nowhere-vanishing) vector field $\xi \in \mathscr{X}(M)$ and a positive-semidefinite corank-1 symmetric $(0,2)$-tensor field $h \in \Gamma(\odot^2 T^*M)$ with the property that $h(\xi, -) = 0$. If $M$ is simply connected, then the structure group can be further reduced to the connected component $G_0$ and hence there is also a "volume" form $\mu \in \Omega^{d+1}(M)$. Even if the structure group does not reduce, we still have a locally defined volume form $\mu_\alpha$ on each $U_\alpha$ and they can be chosen so that they may change by a sign on overlaps.

A **carrollian structure** is a weak carrollian structure enhanced by an adapted connection. As in the case of a Newton–Cartan structure, the torsion of the adapted connection does not characterise the connection uniquely: the contorsion here is measured by a section of the subbundle $\odot^2 \operatorname{Ann} \xi \subset \odot^2 T^*M$, where $\operatorname{Ann} \xi \subset T^*M$ is the bundle of one-forms which annihilate the vector field $\xi$. The intrinsic torsion is now given by $\mathscr{L}_\xi h$, the Lie derivative of $h$ along $\xi$ (see [72, Proposition 8]) and studying the decomposition of the bundle $\odot^2 \operatorname{Ann} \xi$ under the action of the structure group results in four[12] classes of carrollian structures [72, Theorem 10]:

- **totally geodesic**: $\mathscr{L}_\xi h = 0$;
- **minimal**: $\mathscr{L}_\xi \mu = 0$, where $\mu$ is the (possibly only locally defined) volume form;
- **totally umbilical**: $\mathscr{L}_\xi h = f h$ for some $f \in C^\infty(M)$; and
- **generic**, if none of the above are satisfied.

The names have been chosen in analogy with the theory of hypersurfaces in riemannian geometry. This is more than an analogy in that, as shown in [63, 78] a natural source of carrollian manifolds are null hypersurfaces in lorentzian manifolds. Indeed if $N \subset M$ is a null hypersurface in a lorentzian manifold $(M, g)$, then $h$ is the pull-back of $g$ to $N$ and $\xi$ is the null vector field (tangent to $N$) whose integral curves are the null geodesic generators of $N$. Then $\mathscr{L}_\xi h$ is the null second fundamental form of the hypersurface and the names above coincide with the classification of hypersurfaces based on their second fundamental form. The minimality condition is equivalently but more commonly rephrased as the vanishing of the trace of the Weingarten map. There is also a null Weingarten map for null hypersurfaces and it is traceless if and only if the carrollian structure is minimal. Classic references on null hypersurfaces are [79, 80] and in the present context [72, 78].

The homogeneous carrollian spacetimes in Table 5 can be realised as null hypersurfaces in the maximally symmetric lorentzian manifolds in Table 5, but in one dimension higher: Carroll spacetime and the lightcone are null hypersurfaces in Minkowski spacetime, whereas de Sitter–Carroll and anti de Sitter–Carroll spacetimes are null hypersurfaces in de Sitter and anti de Sitter spacetimes, respectively.

The symmetric carrollian spacetimes in Table 5 (i.e., all but the lightcone) are totally geodesic, whereas the lightcone is totally umbilical. In fact, being homogeneous, the function $f \in C^\infty(M)$ in the definition of totally umbilical is a constant. We are not aware of homogeneous examples of minimal and/or generic carrollian structures, but they should exist.

4.3.5. *Aristotelian geometry.* Aristotelian geometries are also describable in terms of $\mathscr{G}$-structures, where $\mathscr{G} \subset \mathrm{GL}(d+1, \mathbb{R})$ is the intersection of any two of the groups defining a galilean, carrollian or lorentzian structures. Comparing the matrices in equations (4.28) and (4.30), we see that $\mathscr{G} \cong \mathrm{O}(d)$ consists of matrices of the form

$$\begin{pmatrix} 1 & \mathbf{0}^\mathsf{T} \\ \mathbf{0} & A \end{pmatrix} \qquad \text{with} \qquad A \in \mathrm{O}(d). \tag{4.31}$$

Choosing basis $P_0, P_1, \ldots, P_d$ for $\mathbb{R}^{d+1}$ and canonical dual basis $\pi^0, \pi^1, \ldots, \pi^d$, we see that $P_0$ and $\pi^0$ are invariant and so are $\delta^{ab} P_a P_b = P_1^2 + \cdots + P_d^2$ and $\delta_{ab} \pi^a \pi^b = (\pi^1)^2 + \cdots + (\pi^d)^2$.

A **(weak) aristotelian structure** on a $(d+1)$-dimensional manifold $M$ is a $\mathscr{G}$-structure with $\mathscr{G} \subset \mathrm{GL}(d+1, \mathbb{R})$ the subgroup of matrices of the form given in equation (4.31). The $\mathscr{G}$-invariant tensors described above give rise to the following: a vector field $\xi$, a one-form $\tau$, a symmetric $(0,2)$-tensor field $h$ and a symmetric $(2,0)$-tensor field $\lambda$ in such a way that $(\tau, \lambda)$ and $(\xi, h)$ are simultaneously a (weak) Newton–Cartan and (weak) carrollian structure. The details about the classification of aristotelian $\mathscr{G}$-structures via their intrinsic torsion can be found in [72, Section 5] and a recent discussion of aristotelian geometry in the context of fractons can be found in [81, Section 5].

4.4. **Coadjoint orbits.** In this section we describe the method of coadjoint orbits in order to write down particle actions.

---

[12]This is for $d > 1$: if $d = 1$ there are only two classes, as for the $d = 1$ Newton–Cartan geometries, consistent with the fact that in $1 + 1$ dimensions, there is no real distinction between carrollian and Newton–Cartan structures.

4.4.1. *Adjoint and coadjoint actions.* Let $\mathscr{G}$ be a Lie group and let $\mathfrak{g}$ be its Lie algebra, whose dual vector space is denoted $\mathfrak{g}^*$. We identify $\mathfrak{g}$ with the tangent space $\mathsf{T}_e\mathscr{G}$ to $\mathscr{G}$ at the identity. The identity is fixed under conjugation by any $g \in \mathscr{G}$ and therefore the differential of conjugation by $g$ defines a group homomorphism $\mathrm{Ad} : \mathscr{G} \to \mathrm{GL}(\mathfrak{g})$ known as the **adjoint representation**. For a matrix group $\mathscr{G}$, if $g \in \mathscr{G}$ and $X \in \mathfrak{g}$, the adjoint action is simply matrix conjugation $\mathrm{Ad}_g X = gXg^{-1}$. The adjoint representation on $\mathfrak{g}$ induces the **coadjoint representation** $\mathrm{Ad}^* : \mathscr{G} \to \mathrm{GL}(\mathfrak{g}^*)$: if $g \in \mathscr{G}$ and $\alpha \in \mathfrak{g}^*$, we have that $\mathrm{Ad}_g^* \alpha = \alpha \circ \mathrm{Ad}_{g^{-1}}$. In other words, for all $X \in \mathfrak{g}$, and using dual pairing notation:

$$\langle \mathrm{Ad}_g^* \alpha, X \rangle = \langle \alpha, \mathrm{Ad}_{g^{-1}} X \rangle. \tag{4.32}$$

Infinitesimally, we have the adjoint $\mathrm{ad} : \mathfrak{g} \to \mathfrak{gl}(\mathfrak{g})$ and coadjoint $\mathrm{ad}^* : \mathfrak{g} \to \mathfrak{gl}(\mathfrak{g}^*)$ representations of the Lie algebra, defined by $\mathrm{ad}_X Y = [X, Y]$ and $\mathrm{ad}_X^* \alpha = -\alpha \circ \mathrm{ad}_X$ for all $X, Y \in \mathfrak{g}$ and $\alpha \in \mathfrak{g}^*$. This last condition can be expressed in dual pairing notation as

$$\langle \mathrm{ad}_X^* \alpha, Y \rangle = -\langle \alpha, [X, Y] \rangle. \tag{4.33}$$

4.4.2. *The symplectic structure on a coadjoint orbit.* Let $\mathscr{O}_\alpha$ be the **coadjoint orbit** of $\alpha \in \mathfrak{g}^*$; that is,

$$\mathscr{O}_\alpha = \left\{ \mathrm{Ad}_g^* \alpha \mid g \in \mathscr{G} \right\}. \tag{4.34}$$

A fundamental property of coadjoint orbits is that they admit a $\mathscr{G}$-invariant symplectic structure, given by the Kirillov–Kostant–Souriau symplectic form $\omega_{\mathrm{KKS}}$. There are several ways to describe this symplectic form. Perhaps the simplest description is in terms of the corresponding Poisson brackets. Every $X \in \mathfrak{g}$ defines a linear function $\ell_X$ on $\mathfrak{g}^*$ by $\ell_X(\alpha) = \langle \alpha, X \rangle$ for all $\alpha \in \mathfrak{g}^*$. We may restrict the $\ell_X$ to smooth functions on the coadjoint orbit $\mathscr{O}_\alpha$. Their differentials $\mathrm{d}\ell_X$ span the cotangent space to the orbit at any point in the orbit. Therefore the Poisson bivector $\Pi_{\mathrm{KKS}}$ dual to the symplectic form (and hence the symplectic form itself) is uniquely determined by its value on the $\mathrm{d}\ell_X$. These are given by the Lie algebra itself:

$$\Pi_{\mathrm{KKS}}(\mathrm{d}\ell_X, \mathrm{d}\ell_Y) = \{\ell_X, \ell_Y\}_{\mathrm{KKS}} = \ell_{[X,Y]}, \tag{4.35}$$

and hence the Jacobi identity follows from that of the Lie algebra. The Jacobi identity for the Poisson brackets is equivalent to the closure of the 2-form inverse of the Poisson bivector. The functions $\ell_X$ are hamiltonians for the $\mathscr{G}$-action on $\mathscr{O}_\alpha$: the hamiltonian vector fields $\{\ell_X, -\}$ generate the infinitesimal action of $\mathscr{G}$ on $\mathscr{O}_\alpha$. To show this, let $\zeta_X \in \mathscr{X}(\mathscr{O}_\alpha)$ be the vector fields which generate the $\mathscr{G}$ action: $\zeta_X(\alpha) = \mathrm{ad}_X^* \alpha$. It follows that

$$\zeta_X(\ell_Y)(\alpha) = \langle \alpha, [X, Y] \rangle = \ell_{[X,Y]} \tag{4.36}$$

A different description, in the spirit of Section 4.1.2, is via the holonomy principle for homogeneous spaces. A $\mathscr{G}$-invariant 2-form $\omega \in \Omega^2(\mathscr{O}_\alpha)$ determines and is determined by a $\mathscr{G}_\alpha$-invariant $\omega_\alpha \in \wedge^2 \mathsf{T}_\alpha^* \mathscr{O}_\alpha$, where $\mathscr{G}_\alpha \subset \mathscr{G}$ is the stabiliser of $\alpha$. Every $X \in \mathfrak{g}$ defines a vector field on $\mathscr{O}_\alpha$ and, evaluating it at $\alpha$, gives a tangent vector there. This defines a linear map $\mathfrak{g} \to \mathsf{T}_\alpha \mathscr{O}_\alpha$, sending $X \in \mathfrak{g}$ to $\mathrm{ad}_X^* \alpha$ which, since $\mathscr{O}_\alpha$ is an orbit, is surjective. The kernel of this map is the Lie algebra $\mathfrak{g}_\alpha$ of the stabiliser group $\mathscr{G}_\alpha$ of $\alpha$. This shows that $\mathsf{T}_\alpha \mathscr{O}_\alpha$ is isomorphic to $\mathfrak{g}/\mathfrak{g}_\alpha$. Denoting the quotient map $\mathfrak{g} \to \mathfrak{g}/\mathfrak{g}_\alpha$ by $X \mapsto \overline{X}$, we have that

$$\omega_\alpha(\overline{X}, \overline{Y}) := \langle \alpha, [X, Y] \rangle. \tag{4.37}$$

It is not hard to check that the RHS only depends on $X, Y$ modulo $\mathfrak{g}_\alpha$ and that $\omega_\alpha$ is non-degenerate. The resulting $\mathscr{G}$-invariant 2-form, denoted $\omega_{\mathrm{KKS}}$ is closed: indeed, its pull-back $\pi^* \omega_{\mathrm{KKS}} \in \Omega^2(\mathscr{G})$ to $\mathscr{G}$ under the orbit map $\pi : \mathscr{G} \to \mathscr{O}_\alpha$ sending $g \mapsto \mathrm{Ad}_g^* \alpha$ is not just closed but in fact exact:

$$\pi^* \omega_{\mathrm{KKS}} = -\mathrm{d} \langle \alpha, \vartheta \rangle, \tag{4.38}$$

where $\vartheta$ is the left-invariant Maurer–Cartan one-form on $\mathscr{G}$. This shows that $\omega_{\mathrm{KKS}}$ is a $\mathscr{G}$-invariant symplectic form on $\mathscr{O}_\alpha$.

Finally, and perhaps the most conceptual reason why coadjoint orbits are symplectic manifolds is that they arise by symplectic reduction from the canonical symplectic structure on the cotangent bundle $\mathsf{T}^*\mathscr{G}$, which is the phase space of the Lie group $\mathscr{G}$ thought of as a configuration space. Any diffeomorphism of $\mathscr{G}$ induces a diffeomorphism of $\mathsf{T}^*\mathscr{G}$ which preserves the symplectic form. In particular, the symplectic form on $\mathsf{T}^*\mathscr{G}$ is invariant under the diffeomorphisms induced from left- and right-multiplications in $\mathscr{G}$. The existence of left- and right-invariant vector fields on Lie groups says that $\mathscr{G}$ is parallelisable and hence that $\mathsf{T}^*\mathscr{G}$ is trivial: that is, $\mathsf{T}^*\mathscr{G} \cong \mathscr{G} \times \mathfrak{g}^*$. There are two natural trivialisations: one using left-multiplication and the other using right-multiplication. Let us use left-multiplication to trivialise $\mathsf{T}^*\mathscr{G}$ and hence identify it with $\mathscr{G} \times \mathfrak{g}^*$. The cartesian projection $\mathscr{G} \times \mathfrak{g}^* \to \mathfrak{g}^*$ defines a function $\mu : \mathsf{T}^*\mathscr{G} \to \mathfrak{g}^*$ which is $\mathscr{G}$-equivariant: it intertwines between the action of $\mathscr{G}$ on $\mathsf{T}^*\mathscr{G}$ induced by left-multiplication and the coadjoint action of $\mathscr{G}$ on $\mathfrak{g}^*$. Pick $\alpha \in \mathfrak{g}^*$ and consider $\mu^{-1}(\alpha)$. These are all the points in $\mathscr{G} \times \mathfrak{g}^*$ of the form $(g, \alpha)$ for any $g \in \mathscr{G}$ and hence it is a copy of $\mathscr{G}$. This copy of $\mathscr{G}$ in $\mathsf{T}^*\mathscr{G}$ is preserved by the

stabiliser $\mathscr{G}_\alpha$ of $\alpha$. Quotienting gives the symplectic quotient $\mu^{-1}(\alpha)/\mathscr{G}_\alpha$ which is a symplectic manifold diffeomorphic to $\mathscr{G}/\mathscr{G}_\alpha$ or, equivalently, to the coadjoint orbit $\mathscr{O}_\alpha$. The resulting symplectic form on $\mathscr{O}_\alpha$ is uniquely characterised by the fact that its pull-back to $\mu^{-1}(\alpha)$ agrees with the restriction to $\mu^{-1}(\alpha)$ of the canonical symplectic form on $\mathsf{T}^*\mathscr{G}$ and a calculation shows that this is again $\omega_{\mathrm{KKS}}$.

In summary, coadjoint orbits of a group $\mathscr{G}$ are homogeneous symplectic manifolds of $\mathscr{G}$. There is a partial converse to this result, which roughly speaking says that all homogeneous symplectic manifolds are coadjoint orbits. More precisely, one has the following "folkloric" theorem, proved recently in [82].

**Theorem.** *Let $\mathscr{G}$ be a connected Lie group and $(\mathsf{M}, \omega)$ a simply-connected homogeneous symplectic manifold of $\mathscr{G}$. Then there exists a covering $\pi : (\mathsf{M}, \omega) \to (\mathscr{O}, \omega_{KKS})$, with $\mathscr{O}$ a coadjoint orbit of a one-dimensional central extension of $\mathscr{G}$, such that $\pi^*\omega_{KKS} = \omega$.*

4.4.3. *Elementary classical systems.* Homogeneous symplectic manifolds of a Lie group $\mathscr{G}$ are the **elementary classical systems** with symmetry $\mathscr{G}$, or perhaps more colloquially, the **elementary particles** with symmetry $\mathscr{G}$ in the nomenclature of Souriau [43]. The above theorem implies that they are locally symplectomorphic to coadjoint orbits of $\mathscr{G}$ or possibly a one-dimensional central[13] extension of $\mathscr{G}$. As shown by Souriau [43], whether or not we need to centrally extend the group comes down to the symplectic cohomology of $\mathscr{G}$, which we now describe briefly.

A smooth function $\theta : \mathscr{G} \to \mathfrak{g}^*$ allows us to define an affinisation of the coadjoint representation

$$\mathfrak{g} \cdot \alpha := \mathrm{Ad}^*_\mathfrak{g} \alpha + \theta(\mathfrak{g}). \tag{4.39}$$

This defines an affine action $g_1 \cdot (g_2 \cdot \alpha) = (g_1 g_2) \cdot \alpha$ precisely when $\theta$ obeys the cocycle condition

$$\theta(g_1 g_2) = \mathrm{Ad}^*_{g_1} \theta(g_2) + \theta(g_1). \tag{4.40}$$

Differentiating $\theta$ at the identity gives a linear map $\mathrm{d}_e \theta : \mathfrak{g} \to \mathfrak{g}^*$ and hence a bilinear form $\mathfrak{c}$ on the Lie algebra defined by

$$\mathfrak{c}(X, Y) = \langle (\mathrm{d}_e \theta)(X), Y \rangle. \tag{4.41}$$

We say that $\theta$ is a *symplectic cocycle* if it satisfies the cocycle condition and $\mathfrak{c}(X, Y) = -\mathfrak{c}(Y, X)$. In that case, $\mathfrak{c} \in \wedge^2 \mathfrak{g}^*$ is a Chevalley–Eilenberg cocycle and hence defines a central extension of $\mathfrak{g}$, which is trivial if and only if there exists $\beta \in \mathfrak{g}^*$ such that $\mathfrak{c}(X, Y) = -\langle \beta, [X, Y] \rangle$. A symplectic cocycle defines a class in the *symplectic cohomology* $\mathsf{H}^1_{\mathrm{symp}}(\mathscr{G}, \mathfrak{g}^*)$, which is trivial if and only if there exists $\beta \in \mathfrak{g}^*$ such that $\theta(g) = \mathrm{Ad}^*_g \beta - \beta$. In that case, the affine coadjoint action (4.39) becomes equivalent to the linear coadjoint representation, being simply the conjugation of the linear coadjoint representation by the constant translation $\alpha \mapsto \alpha - \beta$.

As shown by Souriau, every time a Lie group $\mathscr{G}$ acts symplectically on a symplectic manifold $(\mathsf{M}, \omega)$ and assuming that the fundamental vector fields are hamiltonian, we get a class in the symplectic cohomology of $\mathscr{G}$. Indeed, consider the linear map $\varphi : \mathfrak{g} \to \mathsf{C}^\infty(\mathsf{M})$, sending $X \to \varphi_X$, where the hamiltonian vector field $\{-, \varphi_X\}$ generates the infinitesimal action of $\mathscr{G}$. Dual to $\varphi$ we have the[14] *momentum map* $\mu : \mathsf{M} \to \mathfrak{g}^*$, defined by $\langle \mu, X \rangle = \varphi_X$, originally introduced by Souriau. The symplectic cocycle $\theta : G \to \mathfrak{g}^*$ measures the failure of the moment map to be equivariant relative to the coadjoint representation:

$$\theta(g) := \mathrm{Ad}^*_g \mu(p) - \mu(g \cdot p) \tag{4.42}$$

for any $p \in \mathsf{M}$. Surprisingly, perhaps, assuming that $\mathsf{M}$ is connected, this does not depend on $p$. If the symplectic cohomology class of $\theta$ is trivial, so that $\theta(g) = \mathrm{Ad}^*_g \beta - \beta$ for some fixed $\beta \in \mathfrak{g}^*$, we can translate $\mu \mapsto \mu - \beta$ in such a way that the translated $\mu$ is equivariant:

$$\mu(g \cdot p) - \beta = \mathrm{Ad}^*_g(\mu(p) - \beta). \tag{4.43}$$

This modifies the functions $\varphi_X$ by constants $\varphi_X \mapsto \varphi_X - \langle \beta, X \rangle$, which do not change the hamiltonian vector fields.

Suppose that $\mathscr{G}$ acts transitively on $\mathsf{M}$, so that $\mathsf{M}$ is one orbit of $\mathscr{G}$. If the class of $\theta$ (given in (4.42)) in symplectic cohomology vanishes, then the image of the moment map $\mu : \mathsf{M} \to \mathfrak{g}^*$ is a coadjoint orbit of $\mathscr{G}$. One can show that $\mu$ is a covering map and hence $\mathsf{M}$ covers a coadjoint orbit of $\mathscr{G}$. If $\mathscr{G}$ has vanishing symplectic cohomology, all elementary systems with symmetry $\mathscr{G}$ are (up to covering) coadjoint orbits of $\mathscr{G}$. In contrast, if the symplectic cohomology of $\mathscr{G}$ is not zero, then some elementary systems with symmetry $\mathscr{G}$ do not cover coadjoint orbits of $\mathscr{G}$ but of a one-dimensional central extension of $\mathscr{G}$.

For example, the symplectic cohomology of the Poincaré group vanishes, so that Poincaré-invariant classical elementary systems (i.e., particles) are classified by the coadjoint orbits of the Poincaré group.

---

[13]The extension has to be central since the coadjoint orbit of a non-central extension of a Lie group $\mathscr{G}$ does not admit a natural action of $\mathscr{G}$.

[14]*moment* in the original French and also in part of the symplectic geometry literature

In contrast, the Galilei group does have nontrivial symplectic cohomology and hence Galilei-invariant particles are classified by coadjoint orbits of the Bargmann group, the one-dimensional central extension of the Galilei group already discussed (at the Lie algebraic level) in Section 3.3.

4.4.4. *Coadjoint orbits from geodesic motion.* In the context of lorentzian geometry we can understand the emergence of the coadjoint orbits as follows. Suppose that $(M, g)$ is a lorentzian manifold and let $\gamma$ be an affinely parametrised geodesic for the Levi-Civita connection; i.e., a solution of $\frac{D\dot\gamma}{dt} = 0$. If $\xi \in \mathscr{X}(M)$ is a Killing vector field, then the inner product $g(\xi, \dot\gamma)$ is constant along the geodesic. Let $\mathscr{G}$ be the isometry group and let $\mathfrak{g}$ be its Lie algebra. Then to every $X \in \mathfrak{g}$ we associate a Killing vector field $\xi_X$ and therefore every geodesic defines a momentum $\mu$ in $\mathfrak{g}^*$; namely, the linear map $\mu : \mathfrak{g} \to \mathbb{R}$ defined by $\langle \mu, X \rangle = g(\xi_X, \dot\gamma)$. For every $a \in \mathscr{G}$, let $\phi_a : M \to M$ be the corresponding isometry and suppose that $\gamma$ is a geodesic. Then $\phi_a \circ \gamma$ is also a geodesic and its momentum is given by $\mathrm{Ad}_a^* \mu$, where $\mu$ is the momentum of $\gamma$. The collection of momenta corresponding to all geodesics which are related to $\gamma$ by an isometry define the coadjoint orbit of the momentum of $\gamma$.

As an example, consider affinely parametrised geodesics in Minkowski spacetime $(M, \eta)$. In flat coordinates $x^\mu$, they are given by straight lines $x^\mu(\lambda) = a^\mu + \lambda k^\mu$. Therefore $\dot\gamma = \dot{x}^\mu \partial_\mu = k^\mu \partial_\mu$ and the momentum $\mu$ is given by the linear function sending

$$P_\mu \mapsto \eta(\partial_\mu, \dot\gamma) = \eta(\partial_\mu, k^\nu \partial_\nu) = k^\mu \eta_{\mu\nu} = k_\mu \tag{4.44}$$

and

$$L_{\mu\nu} \mapsto \eta(x_\mu \partial_\nu - x_\nu \partial_\mu, \dot\gamma) = \eta(x_\mu \partial_\nu - x_\nu \partial_\mu, k^\rho \partial_\rho) = a_\mu k_\nu - a_\nu k_\mu, \tag{4.45}$$

which are the linear and relativistic angular momenta, respectively, of the particle. The quadratic function $P^2 = \eta^{\mu\nu} P_\mu P_\nu$ on $\mathfrak{g}^*$ which takes the value $k^2$ on the momentum $\mu$ is constant on the coadjoint orbit and corresponds to $-m^2$, where $m$ is the particle mass. Acting with the translations in the Poincaré group on the geodesic, we can set $a^\mu = 0$ and acting with the Lorentz transformations which preserve the origin, we can bring $k^\mu$ to any desired point on the mass-shell $k^2 = -m^2$. For $m \neq 0$, we can take $k^\mu = (m, 0, 0, 0)$, and for $k^2 = 0$ we can take $k^\mu = (1, 0, 0, 1)$, for instance.

As an example, consider the geodesic traced by a massive particle with mass $m$ in the rest frame, whose momentum is $\mu = m\pi^0$ relative to the basis $\pi^\mu, \lambda^{\mu\nu}$ for $\mathfrak{g}^*$ canonically dual to the basis $P_\mu, L_{\mu\nu}$ for $\mathfrak{g}$. Relative to this basis, the infinitesimal coadjoint action is given by

$$\begin{aligned}
\mathrm{ad}^*_{L_{\mu\nu}} \pi^\rho &= \delta^\rho_\nu \pi_\mu - \delta^\rho_\mu \pi_\nu \\
\mathrm{ad}^*_{P_\mu} \pi^\rho &= \lambda^\rho{}_\mu \\
\mathrm{ad}^*_{L_{\mu\nu}} \lambda^{\alpha\beta} &= \delta^\alpha_\nu \lambda_\mu{}^\beta - \delta^\alpha_\mu \lambda_\nu{}^\beta - \delta^\beta_\nu \lambda_\mu{}^\alpha + \delta^\beta_\mu \lambda_\nu{}^\alpha \\
\mathrm{ad}^*_{P_\mu} \lambda^{\alpha\beta} &= 0,
\end{aligned} \tag{4.46}$$

where we have raised and lowered indices with $\eta_{\mu\nu}$. This allows us to determine the stabiliser subalgebra of $\mu = m\pi^0$, which is seen to be spanned by $L_{12}, L_{13}, L_{23}, P_0$: that is, by the infinitesimal generators of rotations and time translations. These generate a subgroup of the Poincaré group isomorphic to $SO(3) \times \mathbb{R}$. The coadjoint orbit $\mathscr{O}$ is a homogeneous space of the Poincaré group with Klein pair $(\mathfrak{iso}(3, 1), \mathfrak{so}(3) \oplus \mathbb{R})$ and can be identified with the cotangent bundle of (one sheet of) the mass-shell hyperboloid. The homogeneous space with Klein pair $(\mathfrak{iso}(3, 1), \mathfrak{so}(3))$ is, in the language of Souriau [43], the *evolution space* $\mathscr{E}$ of the free massive particle. It is a principal bundle over the coadjoint orbit $\mathscr{O}$ with structure group the one-dimensional group generated by time translations (in Minkowski spacetime). If we let $\varpi : \mathscr{E} \to \mathscr{O}$ denote the bundle projection, the pullback $\sigma := \varpi^* \omega_{\mathrm{KKS}}$ of the symplectic structure defines a pre-symplectic structure on $\mathscr{E}$. It is a closed degenerate two-form on $\mathscr{E}$ whose kernel $\ker \sigma$ defines a rank-one integrable distribution whose leaves are the trajectories of the massive particle. In Souriau's language, but going back to Lagrange, the coadjoint orbit is the *space of motions* of the massive particle.

4.4.5. *Particle actions from coadjoint orbits.* The trajectory of a free particle defines a point in the space of motions but a curve in the evolution space. Therefore if we wish to define a variational problem whose extrema are the trajectories of a free particle, the lagrangian should be defined on the evolution space. We shall assume that just like the space of motions, the evolution space is also a homogeneous space of the symmetry group $\mathscr{G}$ under discussion. In the example above, $\mathscr{G}$ is the Poincaré group and for the massive spinless particle, it is indeed the case that both the space of motions and the evolution space are homogeneous spaces of $\mathscr{G}$.

In this more general discussion, we shall assume that the space of motions is a coadjoint orbit $\mathscr{O}_\alpha$ of $\mathscr{G}$ and we shall let $\mathscr{G}_\alpha \subset \mathscr{G}$ denote the stabiliser subgroup of $\alpha$. We shall let $\varpi : \mathscr{E} \to \mathscr{O}_\alpha$ be the projection sending a point $p \in \mathscr{E}$ to the unique trajectory passing through $p$, which is a point in the space of motions.

This projection is $\mathscr{G}$-equivariant, so that $\varpi(g \cdot \mathfrak{p}) = \mathrm{Ad}_g^* \varpi(\mathfrak{p})$. Choosing a point $\mathfrak{o} \in \mathscr{E}$ with $\varpi(\mathfrak{o}) = \alpha$, we have a a commuting triangle

$$
\begin{array}{ccc}
\mathscr{G} & \xrightarrow{\widehat{\pi}} & \mathscr{E} \\
 & \searrow{\scriptstyle \pi} & \downarrow{\scriptstyle \varpi} \\
 & & \mathscr{O}_\alpha,
\end{array}
\tag{4.47}
$$

where $\widehat{\pi} : \mathscr{G} \to \mathscr{E}$ and $\pi : \mathscr{G} \to \mathscr{O}_\alpha$ are the orbit maps: $\widehat{\pi}(g) = g \cdot \mathfrak{o}$ and $\pi(g) = \mathrm{Ad}_g^* \alpha$, respectively. Commutativity of the triangle says that $\pi = \varpi \circ \widehat{\pi}$. We will let $\mathscr{G}_\mathfrak{o} \subset \mathscr{G}$ be the stabiliser subgroup of $\mathfrak{o} \in \mathscr{E}$ and we observe that $\mathscr{G}_\mathfrak{o} \subset \mathscr{G}_\alpha$. Indeed, if $g \in \mathscr{G}_\mathfrak{o}$, then

$$
\alpha = \varpi(\mathfrak{o}) = \varpi(g \cdot \mathfrak{o}) = \mathrm{Ad}_g^* \varpi(\mathfrak{o}) = \mathrm{Ad}_g^* \alpha,
\tag{4.48}
$$

where we have used the equivariance of $\varpi$. Let $\sigma = \varpi^* \omega_{\mathrm{KKS}} \in \Omega^2(\mathscr{E})$ and $\omega = \pi^* \omega_{\mathrm{KKS}} \in \Omega^2(\mathsf{G})$. Then the commutativity of the above triangle implies that

$$
\widehat{\pi}^* \sigma = \widehat{\pi}^* \varpi^* \omega_{\mathrm{KKS}} = (\varpi \circ \widehat{\pi})^* \omega_{\mathrm{KKS}} = \pi^* \omega_{\mathrm{KKS}} = \omega.
\tag{4.49}
$$

Now let $I \subset \mathbb{R}$ be an interval with parameter $\lambda$ and let $\gamma : I \to \mathscr{E}$ be a curve in the evolution space passing through $\mathfrak{o}$. It is a physical trajectory if and only if $\dot{\gamma} \in \ker \sigma$, so that $\varpi(\gamma(\lambda))$ is constant and equal to $\varpi(\mathfrak{o}) = \alpha$. We now set up a variational problem whose extremals are precisely such curves.

Any curve $\gamma : I \to \mathscr{E}$ may be lifted to a curve $\widehat{\gamma} : I \to \mathscr{G}$ in the group so that $\widehat{\gamma}(\lambda) \cdot \mathfrak{o} = \gamma(\lambda)$. This lift is not unique, since we may multiply on the right with any $h : I \to \mathscr{G}_\mathfrak{o}$. Indeed,

$$
(\widehat{\gamma}h)(\lambda) \cdot \mathfrak{o} = (\widehat{\gamma}(\lambda)h(\lambda)) \cdot \mathfrak{o} = \widehat{\gamma}(\lambda) \cdot h(\lambda) \cdot \mathfrak{o} = \widehat{\gamma}(\lambda) \cdot \mathfrak{o} = \gamma(\lambda).
\tag{4.50}
$$

Recall that $\widehat{\pi}^* \sigma = \omega = \pi^* \omega_{\mathrm{KKS}}$ and hence by equation (4.38), it is exact:

$$
\widehat{\pi}^* \sigma = -\mathrm{d} \langle \alpha, \vartheta \rangle.
\tag{4.51}
$$

We may define an action functional for $\gamma : I \to \mathscr{E}$ by lifting the curve to the group $\widehat{\gamma} : I \to \mathscr{G}$ and defining

$$
S[\widehat{\gamma}] := \int_I \langle \alpha, \widehat{\gamma}^* \vartheta \rangle.
\tag{4.52}
$$

At first sight it seems that this depends on the lift $\widehat{\gamma}$, but notice that under a gauge transformation $\widehat{\gamma} \mapsto \widehat{\gamma}h$, the above action transforms as

$$
S[\widehat{\gamma}h] = S[\widehat{\gamma}] + S[h],
\tag{4.53}
$$

where $S[h]$ is a constant. We thus conclude that the variational problem for the action functional (4.52) is independent on the lift and, therefore, it defines an action functional for curves $\gamma : I \to \mathscr{E}$.

Varying the action functional we find, using the Maurer–Cartan structure equation $\mathrm{d}\vartheta = -\frac{1}{2}[\vartheta, \vartheta]$, that

$$
\delta S[\widehat{\gamma}] = -\int_I \left\langle \alpha, \left[ \widehat{\gamma}^{-1}\dot{\widehat{\gamma}}, \widehat{\gamma}^{-1}\delta\widehat{\gamma} \right] \right\rangle \mathrm{d}\lambda = \int_I \left\langle \mathrm{ad}^*_{\widehat{\gamma}^{-1}\dot{\widehat{\gamma}}} \alpha, \widehat{\gamma}^{-1}\delta\widehat{\gamma} \right\rangle \mathrm{d}\lambda,
\tag{4.54}
$$

where we have used equation (4.33). This vanishes for all variations if and only if $\mathrm{ad}^*_{\widehat{\gamma}^{-1}\dot{\widehat{\gamma}}} \alpha = 0$, so that $\widehat{\gamma}^{-1}\dot{\widehat{\gamma}} = \vartheta(\dot{\widehat{\gamma}}) \in \mathfrak{g}_\alpha$. We claim that this is equivalent to $\dot{\widehat{\gamma}} \in \ker \omega$. Indeed,

$$
\begin{aligned}
\iota_{\dot{\widehat{\gamma}}} \omega &= -\iota_{\dot{\widehat{\gamma}}} \mathrm{d} \langle \alpha, \vartheta \rangle \\
&= \tfrac{1}{2} \iota_{\dot{\widehat{\gamma}}} \langle \alpha, [\vartheta, \vartheta] \rangle \\
&= \left\langle \alpha, [\vartheta(\dot{\widehat{\gamma}}), \vartheta] \right\rangle \\
&= -\left\langle \mathrm{ad}^*_{\vartheta(\dot{\widehat{\gamma}})} \alpha, \vartheta \right\rangle,
\end{aligned}
\tag{4.55}
$$

which vanishes if and only if $\vartheta(\dot{\widehat{\gamma}}) \in \mathfrak{g}_\alpha$. Finally, we observe that $\dot{\widehat{\gamma}} \in \ker \omega$ if and only if $\dot{\gamma} \in \ker \sigma$.

In summary, free particle motion with momentum in the coadjoint orbit $\mathscr{O}_\alpha$ defines a curve in the evolution space which is an extremal of the action functional (4.52). In the following section we will see several examples of this construction, but before doing that let us make an important remark.

As observed in Section 4.2, the same kinematical Lie group might have inequivalent homogeneous spacetimes. For example, the Poincaré group has both Minkowski and anti de Sitter–Carroll as homogeneous spaces. Since coadjoint orbits are a property of the group, their interpretation as the space of motions of a particle in a homogeneous space requires additional information. In this example, the Poincaré coadjoint orbit $\mathscr{O}$ of $\alpha = \mathfrak{m}\pi^0$ can be interpreted as the space of motions of a spinless particle of mass $\mathfrak{m}$ in Minkowski spacetime. What is its interpretation in terms of anti de Sitter–Carroll spacetime? The evolution space $\varpi : \mathscr{E} \to \mathscr{O}$ is also common to both Minkowski and anti de Sitter–Carroll spacetimes, but it admits projections to both spacetimes. In terms of their Klein pairs, with $\mathfrak{g}$ standing for the

Poincaré algebra and $\mathfrak{h}_{\mathscr{O}} = \langle \mathsf{L}_{ab}, \mathsf{H} \rangle$, $\mathfrak{h}_{\mathscr{E}} = \langle \mathsf{L}_{ab} \rangle$, $\mathfrak{h}_{\mathsf{M}} = \langle \mathsf{L}_{ab}, \mathsf{B}_a \rangle$ and $\mathfrak{h}_{\mathsf{AdSC}} = \langle \mathsf{L}_{ab}, \mathsf{P}_a \rangle$, we have the following maps:

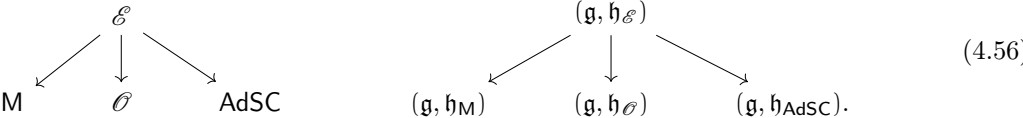

$$(4.56)$$

A point in the coadjoint orbit $\mathscr{O}$ lifts to a curve in the evolution space $\mathscr{E}$ and this projects to a curve in Minkowski spacetime $\mathsf{M}$ or to a curve in anti de Sitter–Carroll spacetime $\mathsf{AdSC}$. The curve in Minkowski spacetime corresponds to the trajectory of a massive spinless particle, since that is how we arrived at this space of motions. We may similarly interpret the curve in anti de Sitter–Carroll spacetime as the trajectory of a carrollian particle. We will see this example in detail in Section 5.1.4.

## 5. Dynamics

In this section we will construct dynamical systems living in some of the homogeneous kinematical spacetimes introduced in Section 4.2. We will focus on the construction of particle actions in various dimensions for some of these spacetimes. We will use the techniques of nonlinear realisations [83, 84] and the coadjoint orbit method [42, 43] described in Section 4.4. Although we are in the context of particle dynamics, the language of nonlinear realisations borrows from its original use in quantum field theory.

In the current context and at its most basic, a nonlinear realisation of a Lie group $\mathscr{G}$ is a smooth transitive action of $\mathscr{G}$ on a manifold $\mathsf{M}$. As we discussed in Section 4.1, once we choose an origin $\mathsf{o} \in \mathsf{M}$, we get a diffeomorphism $\mathsf{M} \cong \mathscr{G}/\mathscr{H}$, with $\mathscr{H}$ the stabiliser of the origin. This diffeomorphism is $\mathscr{G}$-equivariant, intertwining between the $\mathscr{G}$ action on $\mathsf{M}$ and the $\mathscr{G}$ action on $\mathscr{G}/\mathscr{H}$ given simply by left multiplication. Another choice of origin would select a different subgroup $\mathscr{H}$, but any two such subgroups are conjugate in $\mathscr{G}$ and hence the choice is immaterial. Let us choose an origin once and for all and hence a description of $\mathsf{M}$ as the coset space $\mathscr{G}/\mathscr{H}$. As discussed in Section 4.1.3, such a nonlinear realisation of $\mathscr{G}$ is described (up to coverings) infinitesimally by a Klein pair $(\mathfrak{g}, \mathfrak{h})$, where $\mathfrak{g}$ is the Lie algebra of $\mathscr{G}$ and $\mathfrak{h} \subset \mathfrak{g}$ the Lie subalgebra corresponding to the subgroup $\mathscr{H}$. In the language of nonlinear realisations, the subalgebra $\mathfrak{h}$ generates some of the *unbroken* symmetries.

One of the fundamental assumptions in the pioneering papers [83, 84] on nonlinear realisations is that the group $\mathscr{G}$ is compact, connected and semisimple. Since $\mathscr{H}$ is a closed subgroup, it is also compact and hence any finite-dimensional representation of $\mathscr{H}$ is completely reducible into simple (i.e., irreducible) representations. In particular, we can consider the restriction to $\mathscr{H}$ of the adjoint representation of $\mathscr{G}$ on $\mathfrak{g}$. The subalgebra $\mathfrak{h}$ is a subrepresentation and since $\mathscr{H}$ is compact, it has a complementary subrepresentation $\mathfrak{g} = \mathfrak{h} \oplus \mathfrak{m}$, where $\mathfrak{m}$ is isomorphic to the tangent space of the homogeneous space $\mathsf{M}$ at the origin, not just as a vector space but as a representation space of $\mathscr{H}$. (Recall that $\mathscr{H}$ acts on $\mathsf{T}_\mathsf{o}\mathsf{M}$ via the linear isotropy representation.) In other words, the Klein pair $(\mathfrak{g}, \mathfrak{h})$ is reductive. The subspace $\mathfrak{m}$ is said to generate the *broken* symmetries and its elements are often referred as *Goldstone bosons*.

Of course, as we have already seen, kinematical groups are certainly not compact and seldom semisimple, so its Klein pairs $(\mathfrak{g}, \mathfrak{h})$ need not be reductive. However, as we saw in Section 4.2, with one notable exception (the lightcone), the Klein pairs of the kinematical spacetimes are reductive and hence we can talk unambiguously about broken and unbroken generators. In general, the broken generators are equivalence classes in $\mathfrak{g}/\mathfrak{h}$.

Spacetimes are not the only nonlinear realisations of kinematical groups that we will be interested in. In a sense, these describe the vacua. When discussing particle dynamics, the mere existence of the particle in the spacetime breaks the symmetry further. The resulting nonlinear realisation can often be interpreted as the evolution space (in the sense of Souriau [43]) of the particle dynamical system. In the case of elementary particles (again in the sense of Souriau), the evolution space fibres over a coadjoint orbit of the kinematical group. And indeed, one of the approaches to elementary particles in a given homogeneous spacetime $\mathscr{G}/\mathscr{H}$ is to classify coadjoint orbits of $\mathscr{G}$, pass to their evolution spaces and project onto the spacetime. This method was illustrated in Section 4.4.5, resulting in an explicit expression for the particle action functional (4.52).

In practical terms, the method we will follow in this section is the following.

(1) We consider nonlinear realisations of a kinematical (or closely related) group $\mathscr{G}$ on the evolution space of the dynamical system corresponding to a particle propagating on a homogeneous spacetime. Let the evolution space be the coset manifold $\mathscr{G}/\mathscr{H}$ where $\mathscr{H}$ a subgroup of unbroken symmetries.

(2) We then choose a basis of the Lie algebra $\mathfrak{h}$ and extend it to a basis for $\mathfrak{g}$. If the Klein pair $(\mathfrak{g}, \mathfrak{h})$ is reductive, we will choose the basis in such a way that the split $\mathfrak{g} = \mathfrak{h} \oplus \mathfrak{m}$ is preserved by the adjoint action of $\mathscr{H}$. Even if $(\mathfrak{g}, \mathfrak{h})$ is not reductive, we will choose a vector space complement $\mathfrak{m}$.

(3) This choice of basis for $\mathfrak{m}$ gives local coordinates for $\mathscr{G}/\mathscr{H}$ near the origin via (modified) exponential coordinates, as discussed in Section 4.1.4. In effect what this local coordinates do is define a local coset representative $g : \mathscr{G}/\mathscr{H} \to \mathscr{G}$, which in principle is only defined in a neighbourhood of the origin.

(4) We pull back the left-invariant Maurer–Cartan one form on $\mathscr{G}$ via the local coset representative and obtain a $\mathfrak{g}$-valued one-form $\Omega = g^{-1}dg$. Being $\mathfrak{g}$-valued it may be decomposed into a component along $\mathfrak{h}$ and a component along the chosen complement $\mathfrak{m}$: $\Omega = \Omega_{\mathscr{H}} + \Omega_{\mathscr{G}/\mathscr{H}}$.

(5) In order to obtain a particle lagrangian with lowest order in derivatives, we take a linear combination of the $\mathscr{H}$-invariant components of $\Omega$ and pull them back to the interval parametrising the worldline of the particle. In some cases, for example that of the one-dimensional Schwarzian particle discussed in Section 5.4.4, we may consider instead $\mathscr{H}$-invariant quadratic expressions in the components of the Maurer–Cartan form.

We shall have ample opportunity to see how this process works in practice in a number of examples, to which we now turn.

## 5.1. Relativistic Particle lagrangians.
Here, we give a short exposition of the lagrangians of free spinless particles built on the Poincaré group for spacetime dimensions $> 3$ using nonlinear realisations, see e.g. [85]. Free particles can be timelike, lightlike and tachyonic due to the causal structure of Minkowski spacetime. The Poincaré algebra in $d + 1$ spacetime dimensions, denoted $\mathfrak{iso}(d, 1)$, is given in (2.15), where now $A, B = 0, \ldots, d$ and $\eta_{AB}$ is the mostly-plus Minkowski metric. We separate the time and space indices according to $A = (0, a)$, with $a = 1, \ldots, d$. We may also consider lightcone coordinates, where $A = (+, -, i)$ with $i = 1, \ldots, d - 1$.

### 5.1.1. *Massive particle.*
We begin with the construction from nonlinear realisations. For a massive particle in the rest frame, the momentum eigenvalues take the form $p_A = (m, 0, 0, \ldots, 0)$ with $SO(d)$ stabiliser in the Lorentz group generated by rotations[15]. The Klein pair is $(\mathfrak{iso}(d, 1), \mathfrak{so}(d))$, which describes the evolution space of a spinless massive particle in Minkowski spacetime, as described in Section 4.4.4 in the special case of $d = 3$.

The local subgroup $H$ of the nonlinear realisation is thus $SO(d)$ and we write the coset representative for the evolution space as

$$g = \underbrace{e^{x^A P_A}}_{g_0} b, \tag{5.1}$$

where $g_0$ is a coset representative of Minkowski spacetime thought of as the coset space Poincaré/Lorentz and $b = e^{v^a B_a}$ $(a = 1, \ldots, d)$ is a general boost generated by those boost generators $B_a := L_{0a}$ of the Lorentz group which are broken due to the presence of a massive particle in Minkowski spacetime.

The pull-back of the Maurer–Cartan form of the Poincaré group is

$$\Omega = g^{-1}dg = b^{-1}\Omega_0 b + b^{-1}db = \Omega_{(P)}^A P_A + \tfrac{1}{2}\Omega_{(M)}^{AB}L_{AB}, \tag{5.2}$$

where $\Omega_0 = dx^A P_A = g_0^{-1}dg_0$ is the Maurer–Cartan form of Minkowski spacetime.

For a relativistic massive particle in another background, for example AdS, the form of $\Omega_0$ will differ: it would be the pull-back of the Maurer–Cartan form via a local coset representative for AdS.

The explicit form of the Maurer–Cartan forms are obtained by computing the adjoint representation of the boost $b$ on the generators of space-time translation $P_A$:

$$b^{-1}P_A b = \Phi_A{}^B(v^a)P_B, \tag{5.3}$$

and

$$b^{-1}db = \tfrac{1}{2}\eta^{AB}\Phi_A{}^C(v^a)\, d\Phi_B{}^D(v^a)L_{CD}, \tag{5.4}$$

where $\Phi_A{}^B(v^a)$ is the fundamental representation of the Lorentz group, which here depends on the $d$ boost parameters $v^a$. Explicitly,

$$b^{-1}P_0 b = \cosh\|v\|P_0 - \frac{\sinh\|v\|}{\|v\|}v^a P_a$$

$$b^{-1}P_a b = P_a - \frac{1 - \cosh\|v\|}{\|v\|^2}v_a v^b P_b - \frac{\sinh\|v\|}{\|v\|}v_a P_0, \tag{5.5}$$

---

[15]The stabiliser in the Poincaré group contains also the time translations $P_0$.

where $\|\nu\|^2 = \delta_{ab}\nu^a\nu^b$.

We want to construct a lagrangian in terms of the pull-back of the Maurer–Cartan forms subject to two conditions: it should have the lowest possible number of derivatives and it should be invariant under the unbroken $\mathrm{SO}(d)$ subgroup of the Lorentz group. One can therefore choose any component of the Maurer–Cartan form which is invariant under rotations. In this example, we can take the component $\Omega^0_{(P)}$ along $P_0$. Therefore we take lagrangian[16] as the pull-back of $\Omega^0_{(P)}$ to the world-line of the particle: a curve $\gamma(\tau)$ parametrised by $\tau$.

The action of a free massive particle is given by

$$I[t, x^a, \nu^a] = -m \int \gamma^* \Omega^0_{(P)} = -m \int d\tau \, \dot{x}^A \Phi_A{}^0(\nu^a) = -m \int \left( \cosh\|\nu\| \dot{t} - \frac{\sinh\|\nu\|}{\|\nu\|} \nu_a \dot{x}^a \right) d\tau. \qquad (5.6)$$

The vector $\Phi_A{}^0(\nu^a)$ is a timelike unit Lorentz vector and therefore $\eta^{AB}\Phi_A{}^0\Phi_B{}^0 = -1$. The lagrangian depends on the $d+1$ spacetime coordinates $t, x^a$ and the $d$ boost parameters $\nu^a$. The action constructed by nonlinear realisations could be interpreted as a canonical action [88]. In fact the momentum is

$$p_A = \frac{\partial L}{\partial \dot{x}^A} = -m\Phi_A{}^0(\nu^a), \qquad (5.7)$$

so that

$$p_0 = -m\cosh\|\nu\| \qquad \text{and} \qquad p_a = m\frac{\sinh\|\nu\|}{\|\nu\|}\nu_a. \qquad (5.8)$$

and the action becomes

$$I[x^A, \nu^a] = \int d\tau \, p_A(\nu^a)\dot{x}^A. \qquad (5.9)$$

This action is invariant under reparametrisations and the reduced physical space (e.g., by choosing $x^0 = \tau$) has a symplectic structure. For other orbits the structure of the action is the same, the only difference will be the form of the constraint of $\Phi_A{}^0(\nu^a)$. This form of the action is recovered in the coadjoint orbit approach, see for example [89] and below.

Now if we regard $p_A$ as $d+1$ independent degrees of freedom, we can rewrite the action as

$$I[x^A, p_A] = \int d\tau \left( p_A \dot{x}^A - \frac{\gamma}{2}\left( \eta^{AB}p_A p_B + m^2 \right) \right), \qquad (5.10)$$

which is the canonical action of a massive spinless relativistic free particle. Using the equations of motion of $\Phi_A{}^0$ and $\gamma$, we obtain

$$\Phi_A{}^0 = -\frac{\dot{x}_A}{\sqrt{-\dot{x}^2}}. \qquad (5.11)$$

Note that the previous relation can also be obtained from the the vanishing the Maurer–Cartan form $\Omega^a_{(P)}$ associated to the broken translations

$$\Omega^a_{(P)} = dx^A \Phi_A{}^a = 0, \qquad (5.12)$$

which is known as the inverse Higgs mechanism [90].

Substituting back (5.11) in the canonical action and using the equations of motion of $p_A$ and $\gamma$, we obtain the geometrical action

$$I[x^A] = -m \int d\tau \sqrt{-\dot{x}^2}. \qquad (5.13)$$

The quantisation of the mass-shell constraint for a massive particle gives the wave equation

$$\left( \Box - m^2 \right) \Phi(t, \vec{x}) = 0. \qquad (5.14)$$

5.1.2. *Relativistic Massless particle.* Let us now consider a spinless massless particle. In the standard frame the momentum takes the form $p_A = (1, 0, 0, \ldots, 1)$, whose stabiliser in the Lorentz group is isomorphic to the euclidean group $\mathrm{ISO}(d-1)$, with null rotations playing the rôle of euclidean translations. The Klein pair for the evolution space is therefore $(\mathfrak{iso}(d,1), \mathfrak{iso}(d-1))$. It is useful to work in a lightcone frame,

---

[16]The cases of two- and three-dimensional spacetimes are special. In three dimensions we could also add a Wess–Zumino term associated to the $\mathfrak{so}(2)$ rotational component of the Maurer–Cartan form [86–88]; whereas in two dimensions, the absence of spatial rotations implies that we can use any component of the Maurer–Cartan form in the lagrangian.

associated to the lightcone coordinates: $x^+, x^-, x^i$ with $x^\pm = \frac{1}{\sqrt{2}}(x^d \pm x^0)$ and transverse coordinates $x^i$ with $i = 1, \ldots, d-1$. The Poincaré algebra in a lightcone frame has the following nonzero brackets:

$$
\begin{aligned}
[L_{ij}, L_{kl}] &= \delta_{jk} L_{il} - \delta_{ik} L_{jl} - \delta_{jl} L_{ik} + \delta_{il} L_{jk} & [L_{+-}, P_\pm] &= \pm P_\pm \\
[L_{ij}, L_{\pm k}] &= \delta_{jk} L_{\pm i} - \delta_{ik} L_{\pm j} & [L_{ij}, P_k] &= \delta_{jk} P_i - \delta_{ik} P_j \\
[L_{+-}, L_{\pm i}] &= \pm L_{\pm i} & [L_{\pm i}, P_\mp] &= -P_i \\
[L_{+i}, L_{-j}] &= -\delta_{ij} L_{+-} - L_{ij} & [L_{\pm i}, P_j] &= \delta_{ij} P_\pm,
\end{aligned}
\tag{5.15}
$$

where we have used that $\eta_{ij} = \delta_{ij}$ and $\eta_{+-} = 1$. In this case, the Klein pair is $(\mathfrak{iso}(d,1), \mathfrak{iso}(d-1))$ with $\mathfrak{iso}(d-1)$ spanned by $L_{ij}, L_{+i}$ and hence it is not reductive: indeed, $[L_{+i}, L_{-j}]$ always has a rotational component.

The coset space describing the evolution space is now $\mathrm{ISO}(d,1)/\mathrm{ISO}(d-1)$ and we can choose a local coset representative

$$
g = e^{x^A P_A} b = g_0 b,
\tag{5.16}
$$

where now $b = e^{v^i L_{-i}} e^{u L_{+-}}$ with $L_{-i}, L_{+-}$ are the broken boost generators.

In order to compute the Maurer–Cartan form we need the analogue of (5.3) for this new form of $b$. Computing the adjoint representation in this case, the result is a Lorentz matrix $\Phi_A{}^B(v^i, u)$ where now $A, B$ are lightcone indices. The translational component of the Maurer–Cartan form invariant under $\mathrm{ISO}(d-1)$ is $\Omega_{(P)}^+$ and, therefore, the invariant action is

$$
I[x^A, v^i, u] = \int d\tau \, \dot{x}^A \Phi_A{}^+(v^i, u),
\tag{5.17}
$$

where now $\Phi_A{}^+(v^i, u)$ is a Lorentz vector with vanishing norm. The momenta are

$$
p_A = \frac{\partial L}{\partial \dot{x}^A} = \Phi_A{}^+(v^i, u)
\tag{5.18}
$$

and the action becomes

$$
I[x^A, v^i, u] = \int d\tau \, p_A(v^i, u) \dot{x}^A.
\tag{5.19}
$$

In this case, $p_A$ are the components of a null vector, but if we consider $p_A$ as $d+1$ independent variables, we need to introduce a Lagrange multiplier $\gamma$ to implement the constraint $p^2 = 0$ so that the action becomes

$$
I[x^A, p_A] = \int d\tau \left( p_A \dot{x}^A - \tfrac{1}{2} \gamma \eta^{AB} p_A p_B \right),
\tag{5.20}
$$

which is the canonical action of a massless spinless relativistic particle.

5.1.3. *The coadjoint orbit method for relativistic particles.* We note that the component $\Omega_{(P)}^0$ of the Maurer–Cartan form that we used to construct the action for the massive spinless particle is nothing but the pairing of the Maurer–Cartan form with the momentum vector. Indeed, as was done inn Section 4.4.4 in four dimensions, canonically dual to the basis $L_{AB}, P_A$ of the Poincaré algebra $\mathfrak{g}$, we have the basis $\lambda^{AB}, \pi^A$ for the dual $\mathfrak{g}^*$. Then the momentum for a spinless massive particle in the restframe is $p_A \pi^A = m \pi^0$, where $p_A = (m, 0, \ldots, 0)$, and hence

$$
\Omega_{(P)}^0 = \left\langle \pi^0, \Omega_{(P)} \right\rangle,
\tag{5.21}
$$

which, using equation (5.2), can be rewritten as

$$
\Omega_{(P)}^0 = \left\langle \pi^0, b^{-1} \left( dx^A P_A \right) b \right\rangle = \left\langle \pi^0, \mathrm{Ad}_{b^{-1}} \left( dx^A P_A \right) \right\rangle = \left\langle \mathrm{Ad}_b^* \pi^0, dx^A P_A \right\rangle,
\tag{5.22}
$$

where we have used that the action of the Lorentz group element $b^{-1}$ on the vector representation is the adjoint action of $b^{-1}$ as an element of the Poincaré group, which a semi-direct product of the Lorentz group and the vector representation. Similarly, $\mathrm{Ad}_b^*$ is the co-adjoint action on the dual space.

Writing $x^A P_A = \mathbb{X}$ we therefore have the equivalent form of the lagrangian (5.11), see for example [89],

$$
L = \langle \pi, \dot{\mathbb{X}} \rangle,
\tag{5.23}
$$

where $\pi$ is an arbitrary element of the orbit of $m \pi^0$. This orbit can be parametrised with the boost parameters $v^a$ that then appear algebraically in the lagrangian. The momenta can be thought as elements of the dual of the Lie algebra. The derivative in $\dot{\mathbb{X}}$ denotes the derivative with respect to the parameter of the world-line and so we have explicitly carried out the pull-back.

For other orbits like the massless case, we take as element of the dual Lie algebra $\pi^0 + \pi^d$ or for the tachyonic case, the element $\pi^d$. The action of the Lorentz group on the space of momenta take the form (5.23) is universal in all cases.

5.1.4. *Comparing Minkowski and* AdSC *particles.* As discussed at the end of Section 4.4.5, coadjoint orbits are intrinsic to the group $\mathscr{G}$ and the same kinematical group might give rise to different kinematical spacetimes: e.g., Minkowski and anti de Sitter–Carroll (AdSC) are both homogeneous spacetimes of the Poincaré group.

In Section 5.1.1 we derived the lagrangian for a massive spinless particle in Minkowski spacetime. Consider a curve

$$\gamma(\tau) = e^{tH} e^{x^a P_a} e^{v^a B_a} \tag{5.24}$$

in the evolution space, where $t, x^a, v^a$ are functions of $\tau$. Then the lagrangian corresponding to the coadjoint orbit of $m\pi^0 \in \mathfrak{g}^*$ is given in equation (5.6) by

$$L = -m \left( \cosh \|v\| \dot{t} - \frac{\sinh \|v\|}{\|v\|} v_a \dot{x}^a \right). \tag{5.25}$$

We would like to interpret this as a particle action in AdSC. As shown in equation (4.22), what is a translation in Minkowski is a carrollian boost in anti de Sitter–Carroll. This suggests considering the same curve but written in different coordinates adapted to AdSC:

$$\gamma(\tau) = e^{uH} e^{y^a B_a} e^{w^a P_a} \tag{5.26}$$

for some functions $u, y^a, w^a$ of $\tau$. We now interpret $u, y^a$ as local coordinates on AdSC and $w^a$ as parametrising the carrollian boosts which are broken due to the presence of a particle in AdSC. The explicit change of coordinates from $(t, x^a, v^a)$ to $(u, y^a, w^a)$ is given by

$$
\begin{aligned}
u &= t - \frac{\tanh \|v\|}{\|v\|} x^a v_a \\
y^a &= v^a \\
w^a &= x^a + \left( \frac{1 - \cosh \|v\|}{\cosh \|v\|} \right) \frac{x_b v^b}{\|v\|^2} v^a
\end{aligned}
\qquad \text{with inverse} \qquad
\begin{aligned}
t &= u + \frac{\sinh \|y\|}{\|y\|} w_a y^a \\
x^a &= w^a - \frac{1 - \cosh \|y\|}{\|y\|^2} w_b y^b y^a \\
v^a &= y^a.
\end{aligned}
\tag{5.27}
$$

We can then perform the change of variables in the lagrangian to arrive at the following lagrangian for a particle in AdSC:

$$L = -m \left( \cosh \|y\| \dot{u} + \frac{\sinh \|y\|}{\|y\|} w_a \dot{y}^a + \left( 1 - \frac{\sinh \|y\|}{\|y\|} \right) \frac{w_b y^b y_a \dot{y}^a}{\|y\|^2} \right). \tag{5.28}$$

The canonical momenta are

$$
\begin{aligned}
p_0 &= \frac{\partial L}{\partial \dot{u}} = -m \cosh \|y\| \\
p_a &= \underbrace{\frac{w_b y^b}{\|y\|^2} y_a}_{p_a^{\parallel}} + \underbrace{\frac{\sinh \|y\|}{\|y\|} \left( \delta_{ab} - \frac{y_a y_b}{\|y\|^2} \right) w^b}_{p_a^{\perp}}.
\end{aligned}
\tag{5.29}
$$

We see that the "spatial" momentum $p_a$ breaks up into a longitudinal component $p_a^{\parallel}$ along $y^a$ and a transverse component $p_a^{\perp}$. The Euler–Lagrange equation for $u$ says that $\|y\|$ is constant, whereas the Euler–Lagrange equation for $w^a$ says that if $\|y\| \neq 0$, then $y^a$ is constant. Hence a massive particle in AdSC does not move.

5.2. **Non-relativistic particle lagrangians.** In this section we will consider non-relativistic particle lagrangians. We will start by considering the centrally extended Newton–Hooke algebra $\widetilde{\mathfrak{n}}^-$. The Newton–Hooke algebra $\mathfrak{n}^-$ was defined in Section 3.1 and corresponds to $\chi = 0$ in the family of kinematical Lie algebras $\mathfrak{n}_\chi^-$ in Table 1. Its central extension is listed in Table 3. We will introduce an additional parameter in the Lie brackets to allow us to take a limit to the Bargmann Lie algebra which is the universal central extension of the Galilei algebra $\mathfrak{g}$.

The centrally extended Newton–Hooke Lie algebra $\widetilde{\mathfrak{n}}^-$ is spanned by $L_{ab}, B_a, P_a, H, Z$ where $L_{ab}$ span the $\mathfrak{so}(d)$ rotational subalgebra. The Lie brackets are the generic kinematical Lie brackets of equations (3.1), (3.2) and (3.3) together with the following nonzero brackets:

$$[H, B_a] = P_a, \qquad [H, P_a] = \tfrac{1}{R^2} B_a \qquad \text{and} \qquad [B_a, P_b] = \delta_{ab} Z, \tag{5.30}$$

with $Z$ central. Notice that for any nonzero real number $R$, these Lie algebras are all isomorphic, but if we take the limit $R \to \infty$ we obtain the Bargmann algebra in Section 3.1 after $H \mapsto -H$.

5.2.1. *Massive particle.* In this section we will see that a massive spinless particle with symmetry algebra $\widetilde{\mathfrak{n}}^-$ is equivalent to a $d$-dimensional harmonic oscillator. In order to see that, we will consider the coset space with Klein pair $(\widetilde{\mathfrak{n}}^-, \mathfrak{so}(d))$.

We choose a local coset representative of the form

$$\mathfrak{g} = \underbrace{e^{x^0 H} e^{x^a P_a} e^{uZ}}_{\mathfrak{g}_0} \underbrace{e^{v^a B_a}}_{\mathfrak{b}}. \tag{5.31}$$

Here $\mathfrak{g}_0$ is the coset representative of the generalised non-relativistic spacetime and $\mathfrak{b}$ is a Galilei boost parametrised by $v^a$.

The role of the coordinate $u$ of the central charge $Z$ in non-relativistic theories to construct a Wess–Zumino term was first discussed in [91]. The coordinates $x^0, x^a, u$ suggest an interpretation of these coordinates as relativistic coordinates in a space of one higher dimension, here $d + 2$.

We now calculate the pull-back of the Maurer–Cartan form along the coset representative:

$$\Omega = \mathfrak{g}^{-1} d\mathfrak{g} = \mathfrak{b}^{-1} \Omega_0 \mathfrak{b} + \mathfrak{b}^{-1} d\mathfrak{b}, \tag{5.32}$$

where $\Omega_0 = \mathfrak{g}_0^{-1} d\mathfrak{g}_0$. These are easy to calculate and one finds

$$\mathfrak{b}^{-1} d\mathfrak{b} = dv^a B_a \tag{5.33}$$

and

$$\Omega_0 = dx^0 H + dx^a P_a + \left( du - \frac{x^2}{2R^2} dx^0 \right) Z - \frac{x^a}{R} dx^0 B_a. \tag{5.34}$$

We then calculate $\mathfrak{b}^{-1} \Omega_0 \mathfrak{b}$ to arrive at the final expression

$$\Omega = \left( du - \frac{x^2}{2R^2} dx^0 - \tfrac{1}{2} v^2 dx^0 - v_a dx^a \right) Z + dx^0 H + (dv^a - \tfrac{1}{R} x^a dx^0) B_a + (dx^a + v^a dx^0) P_a. \tag{5.35}$$

The $\mathfrak{so}(d)$-invariants in the adjoint representation are $H$ and $Z$ and hence the $\mathfrak{so}(d)$-invariant lagrangian is built out of the $H$ and $Z$ components of $\Omega$. The $H$-component is an exact form, hence it does not contribute to the Euler–Lagrange equations. We will therefore concentrate on the $Z$-component. Pulling it back to the interval parametrising the worldline of the particle, we arrive at the following lagrangian

$$L = \dot{u} - \tfrac{1}{2} \left( \frac{x^2}{R^2} + v^2 \right) \dot{x}^0 - v_a \dot{x}^a. \tag{5.36}$$

The first term is again a total derivative, so it does not contribute to the Euler–Lagrange equations. Its role is to make the lagrangian invariant, since without it the lagrangian is only quasi-invariant. In other words, it is a Wess–Zumino term.

Solving for $v^a$ via its equation $\frac{\partial L}{\partial v^a} = 0$, we find

$$v^a = -\frac{\dot{x}^a}{\dot{x}^0} \tag{5.37}$$

and re-introducing this into the lagrangian, we obtain

$$L = \frac{\dot{x}^2}{2\dot{x}^0} - \frac{x^2}{2R^2} \dot{x}^0. \tag{5.38}$$

If we choose the gauge $\dot{x}^0 = 1$, so we use $x^0$ as the parameter along the worldline, we see that $L$ is indeed the lagrangian for a $d$-dimensional harmonic oscillator with characteristic frequency $\frac{1}{R}$. Taking the limit $R \to \infty$ in the lagrangian, we arrive at the lagrangian for a non-relativistic spinless particle of unit mass:

$$L = \frac{\dot{x}^2}{2\dot{x}^0}. \tag{5.39}$$

Had we considered instead the central extension of the other Newton–Hooke algebra $\mathfrak{n}^+ = \mathfrak{n}^+_{\gamma=-1}$ in Table 1, we would have obtained the inverted harmonic oscillator [92].

5.2.2. *Massless Galilei Particle.* Is there a massless particle associated to the (unextended) Galilei algebra? The answer is yes[17]. The model was first introduced by Souriau [43]. A massless relativistic particle follows a direction on the lightcone. In the non-relativistic case, since the speed of light is infinite, the particle follows a spatial longitudinal direction, say $x^d$. In this case, the unbroken group is generated by $L_{ij}, B_i, P_d$; that is, the infinitesimal rotations $L_{ij}$ on the hyperplane spanned by $x^1, x^2, \ldots, x^{d-1}$, the infinitesimal galilean boosts $B_i$ in directions perpendicular to $x^d$ and the infinitesimal longitudinal translations along $x^d$. The evolution space has Klein pair $(\mathfrak{g}, \mathfrak{iso}(d-1))$, where $\mathfrak{g}$ is the Galilei algebra, as in Table 1, and the $\mathfrak{iso}(d-1)$ subalgebra is spanned by $L_{ij}, B_i$, for $i, j = 1, \ldots, d-1$.

---

[17]We acknowledge discussions with Axel Kleinschmidt on this point.

A local coset representative is

$$g = \underbrace{e^{tH + x^i P_i + x^d P_d}}_{g_0} \underbrace{e^{\theta^i R_i} e^{\nu B_d}}_{b}, \tag{5.40}$$

with $i = 1, \ldots, d-1$, where $R_i := L_{id}$ are the broken rotations and $B_d$ is the broken longitudinal boost.

The pull-back of the Maurer–Cartan form is given as usual by

$$\Omega = \mathrm{Ad}_{b^{-1}} \Omega_0 + b^{-1} db, \tag{5.41}$$

where

$$\Omega_0 = g_0^{-1} dg_0 = dt H + dx^i P_i + dx^d P_d. \tag{5.42}$$

The $\mathrm{ISO}(d-1)$-invariant subspace of $\mathfrak{g}$ is spanned by $P_d$ and $B_d$, hence we need to extract those components of $\Omega$ in order to write down the lagrangian. We notice that

$$b^{-1} db = e^{-\nu B_d} e^{-\theta^i R_i} d(e^{\theta^i R_i} e^{\nu B_d}) = e^{-\nu B_d} \left( e^{-\theta^i R_i} de^{\theta^i R_i} \right) e^{\nu B_d} + d\nu B_d. \tag{5.43}$$

Since $[R_i, R_j] = -L_{ij}$, the expression in parenthesis lives in the span of $L_{ij}, R_i$ and hence the first of the above terms lives in the span of $L_{ij}, R_i, B_i$. Therefore the only term in $b^{-1} db$ which contributes to the lagrangian is $d\nu B_d$.

We calculate $\mathrm{Ad}_{b^{-1}} \Omega_0$ paying particular attention to the $P_d$ component:

$$\begin{aligned}
\mathrm{Ad}_{b^{-1}} \Omega_0 &= dt H - \nu dt P_d + \exp(\mathrm{ad}_{-\theta \cdot R}) dx^i P_i + \exp(\mathrm{ad}_{-\theta \cdot R}) dx^d P_d \\
&= dt H + \left( dx^i + \frac{\cos \|\theta\| - 1}{\|\theta\|^2} \theta^i \theta_j dx^j - \frac{\sin \|\theta\|}{\|\theta\|} \theta^i dx^d \right) P_i \\
&\quad + \left( \cos \|\theta\| dx^d + \frac{\sin \|\theta\|}{\|\theta\|} \theta_i dx^i - \nu dt \right) P_d,
\end{aligned} \tag{5.44}$$

where $\|\theta\|^2 = \delta_{ij} \theta^i \theta^j$. In summary,

$$\Omega = d\nu B_d + \left( \cos \|\theta\| dx^d + \frac{\sin \|\theta\|}{\|\theta\|} \theta_i dx^i - \nu dt \right) P_d + \cdots \tag{5.45}$$

omitting terms which are not $\mathrm{ISO}(d-1)$-invariant. The $B_d$ component is exact, so that it does not contribute to the Euler–Lagrange equations. Therefore we concentrate on the $P_d$ component and introducing the "colour" $k$ [43], we may write the lagrangian as the pull-back to the interval of the $P_d$-component:

$$L = k \left( \cos \|\theta\| \dot{x}^d + \frac{\sin \|\theta\|}{\|\theta\|} \theta_i \dot{x}^i - \nu \dot{t} \right). \tag{5.46}$$

We calculate the spatial canonical momentum $\vec{p} = (p_1, p_2, \ldots, p_d)$ to be

$$p_d = \frac{\partial L}{\partial \dot{x}^d} = k \cos \|\theta\| \qquad \text{and} \qquad p_i = \frac{\partial L}{\partial \dot{x}^i} = k \frac{\sin \|\theta\|}{\|\theta\|} \theta_i, \tag{5.47}$$

from where we see that $\vec{p} \cdot \vec{p} = k^2$. Introducing the associated unconstrained momentum, we implement the above constraint via a Lagrange multiplier $e$ to arrive at the lagrangian

$$L = p_d \dot{x}^d + p_i \dot{x}^i - \nu \dot{t} + \tfrac{1}{2} e \left( \vec{p} \cdot \vec{p} - k^2 \right). \tag{5.48}$$

Notice that the $\nu$ equation of motion is $\frac{\partial L}{\partial \nu} = -k \dot{t} = 0$, so that propagation is instantaneous.

The quantisation of the mass-shell constraint gives the Helmholtz equation

$$(\nabla^2 + k^2) \Phi(t, \vec{x}) = 0, \tag{5.49}$$

which agrees with the field equation of the galilean magnetic Klein–Gordon field (see equation (7.4)). The approach to quantum field theory in terms of particle variables is know as the *world approach* to field theory and it was first considered by Feynman [93, 94], see also for example [95, 96].

## 5.3. Carroll particle lagrangians.
In this section we will construct the action of a massive (timelike) particle in Carroll spacetime. The case of a massless Carroll particle can be obtained from from the massive one by taking the mass to be zero. We will also construct the lagrangian of a tachyonic particle. The presence of these three kinds of particles (timelike, lightlike and tachyonic) is due to the causal structure of the Carroll geometry. This is analogous to what happens in lorentzian geometry, but in contrast with the galilean case, where the notion of mass is not related to the causal structure.

The Carroll algebra is denoted $\mathfrak{c}$ and given in Table 1. Besides the Lie brackets in equations (3.1), (3.2) and (3.3), which are shared by all kinematical Lie algebras, the only nonzero bracket in $\mathfrak{c}$ is

$$[B_a, P_b] = \delta_{ab} H. \tag{5.50}$$

In contrast to the Galilei algebra, the Carroll algebra (for $d \geqslant 3$) does not allow nontrivial central extensions, although $H$ is a central element.

5.3.1. *Massive Carroll particle.* We construct the timelike massive Carroll particle lagrangian [97] [63] using the method of nonlinear realisations. The Klein pair for the evolution space is $(\mathfrak{c}, \mathfrak{so}(d))$, where $\mathfrak{so}(d)$ is the span of the rotations $L_{ab}$. A coset representative for the corresponding coset space is given by

$$g = \underbrace{e^{tH + x^a P_a}}_{g_0} \underbrace{e^{\nu^a B_a}}_{b}, \tag{5.51}$$

where $t, x^a$ are the Goldstone bosons associated to spacetime translations and $\nu^a$ are the Goldstone bosons associated to the broken boosts.

The Maurer-Cartan form $\Omega$ is given by

$$\Omega = g^{-1} dg = \mathrm{Ad}_{b^{-1}} \Omega_0 + b^{-1} db, \tag{5.52}$$

where

$$\Omega_0 = g_0^{-1} dg_0 = dtH + dx^a P_a \qquad \text{and} \qquad b^{-1} db = d\nu^a B_a. \tag{5.53}$$

It follows that

$$\mathrm{Ad}_{b^{-1}} \Omega_0 = dtH + \exp(-\mathrm{ad}_{\nu^a B_a}) dx^b P_b = dtH + dx^b (P_b - \nu_b H) = (dt - \nu_b dx^b)H + dx^b P_b. \tag{5.54}$$

The lagrangian is the pull-back to the interval of the rotationally invariant component of $\Omega$, which is the component along $H$:

$$L = M(-\dot{t} + \nu_a \dot{x}^a) \tag{5.55}$$

where we have introduced a mass $M$. Notice that the ordinary massive particle does not move: the momentum of the Carroll particle $p_a = M\nu_a$ and there is no relation between the momentum and the velocity of the particle. The canonical lagrangian is obtained by introducing a Lagrange multiplier $e$:

$$L_{\mathrm{can}} = -E\dot{t} + p_a \dot{x}^a - \tfrac{1}{2} e \left( E^2 - M^2 \right). \tag{5.56}$$

Note that in this form we have introduced also negative energies that are allowed in the Carroll case. Since $H$ is a Casimir, its eigenvalues can take any real value: positive negative or zero. Physically, a timelike or lightlike Carroll particle does not move.

The quantisation of the mass-shell constraint for a Carroll massive particle gives the wave equation [97]

$$\left( \frac{\partial^2}{\partial t^2} + M^2 \right) \Phi(t, \vec{x}) = 0; \tag{5.57}$$

that is, the equation of motion of the carrollian electric Klein-Gordon field theory, see (7.18).

5.3.2. *Tachyonic Carroll Particle.* Here we construct the tachyon Carroll particle lagrangian [98]. A relativistic tachyon has a spacelike momentum, but in the ultra-relativistic limit, the lightcone collapses to the timeline and hence any momentum having a nonzero component along a spacelike direction is tachyonic. For example, we may take the momentum purely along the $d$-direction: $\alpha = M\pi^d \in \mathfrak{c}^*$ in the dual of the Carroll algebra. The resulting coadjoint orbit has Klein pair $(\mathfrak{c}, \mathfrak{h}_\alpha)$, where $\mathfrak{h}_\alpha$ is spanned by $L_{ij}, B_i, B_d, P_d, P_0$, where now $i, j = 1, \ldots, d-1$. The subalgebra $\mathfrak{h}_\alpha$ is isomorphic to the direct sum of the $\mathfrak{iso}(d-1)$ algebra generated by $L_{ij}, B_i$ and the Heisenberg algebra generated by $B_d, P_d, P_0$. The evolution space is obtained by breaking the translation symmetry in the $d$-direction. Therefore the Klein pair for the evolution space is $(\mathfrak{c}, \mathfrak{h})$, where $\mathfrak{h}$ is spanned by $L_{ij}, B_i, B_d, P_0$ and isomorphic now to $\mathfrak{iso}(d-1) \oplus \mathbb{R}^2$. This Klein pair is not reductive, but we may choose a complement $\mathfrak{m}$ spanned by $R_i := L_{id}, P_i, P_d$. It is not reductive because $[B_d, P_d] = P_0 \notin \mathfrak{m}$. Nevertheless the image $\bar{P}_d$ of $P_d$ in $\mathfrak{c}/\mathfrak{h}$ is an invariant of the linear isotropy representation of $\mathfrak{h}$ on $\mathfrak{c}/\mathfrak{h}$.

Let us choose a coset representative

$$g = \underbrace{e^{x^d P_d} e^{x^i P_i}}_{g_0} \underbrace{e^{\theta^i R_i}}_{b} \tag{5.58}$$

and pull back the left-invariant Maurer–Cartan form. This will take values in the Carroll algebra, but we project to $\mathfrak{c}/\mathfrak{h}$ and keep the invariant components, which here is only the component along $P_d$. Equivalently, we calculate the dual pairing $\langle \alpha, g^{-1} dg \rangle$. A calculation shows that

$$g^{-1} dg = \left( \cos \|\theta\| dx^d + \frac{\sin \|\theta\|}{\|\theta\|} \theta_i dx^i \right) P_d + \cdots \tag{5.59}$$

where $\|\theta\|^2 = \delta_{ij} \theta^i \theta^j$, hence

$$\langle \alpha, g^{-1} dg \rangle = M \langle \pi^d, g^{-1} dg \rangle = M \left( \cos \|\theta\| dx^d + \frac{\sin \|\theta\|}{\|\theta\|} \theta_i dx^i \right). \tag{5.60}$$

The lagrangian is now obtained by pulling back this component to the interval parametrising the worldline of the particle:

$$L = M \left( \cos \|\theta\| \dot{x}^d + \frac{\sin \|\theta\|}{\|\theta\|} \theta_i \dot{x}^i \right). \tag{5.61}$$

The canonical momenta are given by

$$p_d := \frac{\partial L}{\partial \dot{x}^d} = M \cos \|\theta\| \qquad \text{and} \qquad p_i := \frac{\partial L}{\partial \dot{x}^i} = M \frac{\sin \|\theta\|}{\|\theta\|} \theta_i, \tag{5.62}$$

from where it follows that they are constrained:

$$\vec{p} \cdot \vec{p} := p_d^2 + \sum_i p_i^2 = M^2. \tag{5.63}$$

We may implement this constraint via a Lagrange multiplier to arrive at the lagrangian

$$L = p_d \dot{x}^d + p_i \dot{x}^i + \tfrac{1}{2} \lambda (\vec{p} \cdot \vec{p} - M^2). \tag{5.64}$$

Since $\dot{x}^0$ does not appear in the action, its momentum $p_0$ is also zero. We may implement that constraint with a second Lagrange multiplier and write the canonical lagrangian as

$$L = p_A \dot{x}^A + \tfrac{1}{2} \lambda (\vec{p} \cdot \vec{p} - M^2) + \mu p_0, \tag{5.65}$$

Notice that the mass-shell constraint $\vec{p} \cdot \vec{p} - M^2 = 0$ coincides with the one of the massless galilean particle and also that the energy ($p_0$) of a tachyon particle is zero. The associated wave equation reduces to a Helmholtz equation:

$$(\vec{\nabla}^2 + M^2)\Phi(t, \vec{x}) = 0 \qquad \text{and} \qquad \frac{\partial}{\partial t}\Phi(t, \vec{x}) = 0, \tag{5.66}$$

which is related to the equations of motion of magnetic Carroll field theory as in equation (7.15). The relation among Galilean and Carroll particles has been studied in [99] based on the duality among Galilei and Carroll algebras [33], at the level of the associated wave equations (5.49) and (5.57).

5.4. **One- and two-dimensional particle dynamics with** $\mathrm{SL}(2, \mathbb{R})$ **symmetry.** In this section we will give several examples of one- and two-dimensional particle dynamics with $\mathrm{SL}(2, \mathbb{R})$ symmetry. There are three two-dimensional homogeneous spaces of $\mathrm{SL}(2, \mathbb{R})$: corresponding to the hyperbolic plane, the lightcone and (anti) de Sitter spacetime. In addition, $\mathrm{SL}(2, \mathbb{R})$ acts on the real projective line $\mathbb{R}\mathrm{P}^1$ (also known as one-dimensional conformal space) via projective transformations. Among the particle dynamics discussed here, we will recover the conformal mechanics of [44], see, e.g., [100–103] and the Schwarzian particle action of [45–47].

As already discussed in Section 4.2, there are three spatially isotropic homogeneous spaces associated to the Lorentz group $\mathrm{SO}(d, 1)$: namely, hyperbolic space $\mathsf{H}_d$, de Sitter spacetime $\mathsf{dS}_d$ and the future lightcone $\mathbb{L}_d$. The picture is the familiar foliation of Minkowski spacetime into orbits of the Lorentz group. Whereas de Sitter spacetime is a maximally symmetric lorentzian manifold and hyperbolic space is a maximally symmetric riemannian manifold, the future lightcone is what we could term a maximally symmetric carrollian manifold. Each of these three spaces is described infinitesimally by a Klein pair $(\mathfrak{g}, \mathfrak{h})$ where $\mathfrak{g} = \mathfrak{so}(d, 1)$ and

$$\mathfrak{h} \cong \begin{cases} \mathfrak{so}(d) & (\mathsf{H}_d) \\ \mathfrak{so}(d-1, 1) & (\mathsf{dS}_d) \\ \mathfrak{iso}(d-1) & (\mathbb{L}_d) \end{cases} \tag{5.67}$$

In this section we will concentrate in the case $d = 2$ and we will use the isomorphism $\mathfrak{so}(2, 1) \cong \mathfrak{sl}(2, \mathbb{R})$. Each of the above Klein pairs can thus be realised geometrically as coset spaces $\mathrm{SL}(2, \mathbb{R})/\mathscr{H}$ for some one-dimensional connected closed Lie subgroup $\mathrm{H} \subset \mathrm{SL}(2, \mathbb{R})$. Up to conjugation in $\mathrm{SL}(2, \mathbb{R})$, there are three connected closed one-dimensional Lie subgroups $\mathrm{H} \subset \mathrm{SL}(2, \mathbb{R})$:

$$(\text{elliptic}) \qquad \mathrm{H} = \left\{ \begin{pmatrix} \cos \theta & -\sin \theta \\ \sin \theta & \cos \theta \end{pmatrix} \, \middle| \, \theta \in \mathbb{R}/2\pi\mathbb{Z} \right\} \tag{5.68}$$

$$(\text{hyperbolic}) \qquad \mathrm{H} = \left\{ \begin{pmatrix} \cosh \tau & \sinh \tau \\ \sinh \tau & \cosh \tau \end{pmatrix} \, \middle| \, \tau \in \mathbb{R} \right\} \tag{5.69}$$

$$(\text{parabolic}) \qquad \mathrm{H} = \left\{ \begin{pmatrix} 1 & \zeta \\ 0 & 1 \end{pmatrix} \, \middle| \, \zeta \in \mathbb{R} \right\}. \tag{5.70}$$

They can be distinguished by the trace of the non-identity elements: $< 2$ in the elliptic case, $> 2$ in the hyperbolic case and $= 2$ in the parabolic case. They can also be distinguished by the causal nature of

the vectors they leave invariant in the three-dimensional vector representation of $\mathfrak{so}(2,1)$: timelike in the elliptic case, spacelike in the hyperbolic case and lightlike in the parabolic case.

Since the vector representation of $\mathrm{SL}(2,\mathbb{R})$ is isomorphic to the coadjoint representation, these homogeneous spaces can also be interpreted as coadjoint orbits and hence, according to Souriau, as the space of motions of elementary systems. The evolution spaces can in all cases be interpreted as the Lie group $\mathrm{SL}(2,\mathbb{R})$ itself. It is then a matter of interpretation how to project the trajectories on the evolution space into particle trajectories in the spacetime.

The Lie algebras of these Lie subgroups are given by

$$\text{(elliptic)} \qquad \mathfrak{h} \; = \left\{ \begin{pmatrix} 0 & -z \\ z & 0 \end{pmatrix} \bigg| \; z \in \mathbb{R} \right\} \tag{5.71}$$

$$\text{(hyperbolic)} \qquad \mathfrak{h} \; = \left\{ \begin{pmatrix} 0 & z \\ z & 0 \end{pmatrix} \bigg| \; z \in \mathbb{R} \right\} \tag{5.72}$$

$$\text{(parabolic)} \qquad \mathfrak{h} \; = \left\{ \begin{pmatrix} 0 & z \\ 0 & 0 \end{pmatrix} \bigg| \; z \in \mathbb{R} \right\}. \tag{5.73}$$

We will write $\mathfrak{g} = \mathfrak{h} \oplus \mathfrak{m}$ in each case with

$$\text{(elliptic)} \qquad \mathfrak{m} \; = \left\{ \begin{pmatrix} x & y \\ y & -x \end{pmatrix} \bigg| \; x, y \in \mathbb{R} \right\} \tag{5.74}$$

$$\text{(hyperbolic)} \qquad \mathfrak{m} \; = \left\{ \begin{pmatrix} x & -y \\ y & x \end{pmatrix} \bigg| \; x, y \in \mathbb{R} \right\} \tag{5.75}$$

$$\text{(parabolic)} \qquad \mathfrak{m} \; = \left\{ \begin{pmatrix} x & 0 \\ y & -x \end{pmatrix} \bigg| \; x, y \in \mathbb{R} \right\}. \tag{5.76}$$

In the elliptic and hyperbolic cases, the split $\mathfrak{g} = \mathfrak{h} \oplus \mathfrak{m}$ is reductive, so that $[\mathfrak{h}, \mathfrak{m}] \subset \mathfrak{m}$ in the obvious notation, whereas in the parabolic case no reductive split exists. Let us write $\mathfrak{m} = \{xP_1 + yP_2 \mid x, y \in \mathbb{R}\}$ and $\mathfrak{h} = \{zB \mid z \in \mathbb{R}\}$ in all cases, which defines the matrices $B, P_1, P_2$:

$$\text{(elliptic)} \qquad B \; = \begin{pmatrix} 0 & -1 \\ 1 & 0 \end{pmatrix} \qquad P_1 = \begin{pmatrix} 1 & 0 \\ 0 & -1 \end{pmatrix} \qquad P_2 = \begin{pmatrix} 0 & 1 \\ 1 & 0 \end{pmatrix} \tag{5.77}$$

$$\text{(hyperbolic)} \qquad B \; = \begin{pmatrix} 0 & 1 \\ 1 & 0 \end{pmatrix} \qquad P_1 = \begin{pmatrix} 1 & 0 \\ 0 & -1 \end{pmatrix} \qquad P_2 = \begin{pmatrix} 0 & -1 \\ 1 & 0 \end{pmatrix} \tag{5.78}$$

$$\text{(parabolic)} \qquad B \; = \begin{pmatrix} 0 & 1 \\ 0 & 0 \end{pmatrix} \qquad P_1 = \begin{pmatrix} 1 & 0 \\ 0 & -1 \end{pmatrix} \qquad P_2 = \begin{pmatrix} 0 & 0 \\ 1 & 0 \end{pmatrix}. \tag{5.79}$$

In the elliptic case, there is a positive-definite inner product on $\mathfrak{m}$ which is $\mathscr{H}$-invariant: $\langle P_a, P_b \rangle = \delta_{ab}$, whereas in the hyperbolic case there is an $\mathscr{H}$-invariant lorentzian inner product on $\mathfrak{m}$ given by $\langle P_a, P_b \rangle = \eta_{ab}$, with $\eta$ diagonal with $\eta_{11} = -\eta_{22} = 1$. In the parabolic case, $P_1$ is $\mathscr{H}$-invariant (in $\mathfrak{g}/\mathfrak{h}$) and so is the degenerate bilinear form $\mathfrak{b}$ whose only nonzero entry is $\mathfrak{b}(P_2, P_2)$.

We shall describe $\mathrm{SL}(2,\mathbb{R})$-invariant particle dynamics on each of the coset manifolds $\mathrm{SL}(2,\mathbb{R})/\mathscr{H}$, where $\mathscr{H}$ is either an elliptic, hyperbolic or parabolic subgroup. To do so we will parametrise a neighbourhood of the identity of $\mathrm{SL}(2,\mathbb{R})$ via $g : \mathbb{R}^3 \to \mathrm{SL}(2,\mathbb{R})$ where

$$g(x, y, z) = \underbrace{e^{yP_2} e^{xP_1}}_{g_0} \underbrace{e^{zB}}_{\mathfrak{b}}, \tag{5.80}$$

where $g_0$ is a coset representative for the spacetime and $\mathfrak{b}$ corresponds to the extra generator that is broken by the presence of the particle. Notice that $B, P_1, P_2$ are defined differently in each of the three cases, as show above, consistent with this interpretation.

The left-invariant Maurer–Cartan one-form on $\mathrm{SL}(2,\mathbb{R})$ pulls back to $g^{-1}dg \in \Omega^1(\mathbb{R}^3, \mathfrak{g})$. Choosing $\alpha \in \mathfrak{g}^*$, we have that $L_\alpha := \langle \alpha, g^{-1}dg \rangle \in \Omega^1(\mathbb{R}^3)$, where we have used $\langle -, - \rangle$ to denote the dual pairing between $\mathfrak{g}$ and $\mathfrak{g}^*$. Let $I := [a, b] \subset \mathbb{R}$ and let $\gamma : I \to \mathbb{R}^3$ be a regular curve. We may pull back $L_\alpha$ via $\gamma$ to produce a one-form $\gamma^* L_\alpha \in \Omega^1(I)$ which we may integrate to arrive at the following action functional:

$$S_\alpha = \int_I \gamma^* L_\alpha. \tag{5.81}$$

We will see that after partially solving the Euler–Lagrange equations, $S_\alpha$ will induce an action for particle dynamics in $\mathrm{SL}(2,\mathbb{R})/\mathscr{H}$.

5.4.1. *Particle dynamics on the hyperbolic plane.* Despite the name, the hyperbolic plane $\mathsf{H}_2$ is the quotient of $\mathrm{SL}(2,\mathbb{R})$ by an elliptic subgroup. Let us write $g^{-1}dg = \theta^1 P_1 + \theta^2 P_2 + \theta^3 B$, where

$$\begin{aligned}
\theta^1 &= \cosh(2x)\cos(2z)dy - \sin(2z)dx \\
\theta^2 &= \cos(2z)dx + \cosh(2x)\sin(2z)dy \\
\theta^3 &= dz + \sinh(2x)dy.
\end{aligned} \tag{5.82}$$

The invariant metric on $\mathrm{SL}(2,\mathbb{R})/\mathscr{H}$ is given (up to homothety) by

$$ds^2 = (\theta^1)^2 + (\theta^2)^2 = dx^2 + \cosh(2x)^2 dy^2. \tag{5.83}$$

The action is given by

$$S_\alpha = \int_a^b \left( \alpha_1(\cosh(2x)\cos(2z)\dot{y} - \sin(2z)\dot{x}) + \alpha_2(\cos(2z)\dot{x} + \cosh(2x)\sin(2z)\dot{y}) + \alpha_3(\dot{z} + \sinh(2x)\dot{y}) \right) dt. \tag{5.84}$$

The Euler–Lagrange equation for $z$ is simply $\frac{\partial L}{\partial z} = 0$, which is equivalent to

$$\alpha_1(\cosh(2x)\sin(2z)\dot{y} + \cos(2z)\dot{x}) = \alpha_2(\cosh(2x)\cos(2z)\dot{y} - \sin(2z)\dot{x}), \tag{5.85}$$

from where we may solve (implicitly) for $z$ as follows:

$$\tan(2z) = \frac{\alpha_2 \cosh(2x)\dot{y} - \alpha_1 \dot{x}}{\alpha_1 \cosh(2x)\dot{y} + \alpha_2 \dot{x}}. \tag{5.86}$$

Reinserting into the action (and dropping total derivatives), we arrive at

$$S'_\alpha = \int_a^b \left( \sqrt{\alpha_1^2 + \alpha_2^2}\sqrt{\dot{x}^2 + \cosh(2x)^2\dot{y}^2} + \alpha_3 \sinh(2x)\dot{y} \right) dt. \tag{5.87}$$

We recognise the first term as the line element in $\mathsf{H}_2$ with hyperbolic metric

$$ds^2 = (\alpha_1^2 + \alpha_2^2)(dx^2 + \cosh(2x)^2 dy^2), \tag{5.88}$$

whereas the second term is the coupling to a Maxwell field

$$A = \alpha_3 \sinh(2x)dy, \tag{5.89}$$

whose fieldstrength

$$F = dA = 2\alpha_3 \cosh(2x)dx \wedge dy = \frac{2\alpha_3}{\alpha_1^2 + \alpha_2^2}\mathrm{dvol}, \tag{5.90}$$

where $\mathrm{dvol}$ is the hyperbolic area form of the metric in equation (5.88).

5.4.2. *Particle dynamics on de Sitter spacetime.* This case is very similar *mutatis mutandis* to the previous case, although now we quotient by a hyperbolic subgroup. Again we write $g^{-1}dg = \theta^1 P_1 + \theta^2 P_2 + \theta^3 B$, where

$$\begin{aligned}
\theta^1 &= \cosh(2x)\cosh(2z)dy + \sinh(2z)dx \\
\theta^2 &= \cosh(2z)dx + \cosh(2x)\sinh(2z)dy \\
\theta^3 &= dz - \sinh(2x)dy.
\end{aligned} \tag{5.91}$$

The invariant metric on $\mathrm{SL}(2,\mathbb{R})/\mathscr{H}$ is now given (up to homothety) by

$$ds^2 = (\theta^1)^2 - (\theta^2)^2 = -dx^2 + \cosh(2x)^2 dy^2. \tag{5.92}$$

As we see from the metric, $x$ is a time coordinate and $y$ is a space coordinate. This metric could also be re-interpreted as $\mathsf{AdS}_2$, by reinterpreting $x$ as space and $y$ as time.

The action is now given by

$$S_\alpha = \int_a^b \left( \alpha_1(\cosh(2x)\cosh(2z)\dot{y} + \sinh(2z)\dot{x}) + \alpha_2(\cosh(2z)\dot{x} + \cosh(2x)\sinh(2z)\dot{y}) + \alpha_3(\dot{z} - \sinh(2x)\dot{y}) \right) dt. \tag{5.93}$$

The Euler–Lagrange equation for $z$ is again simply $\frac{\partial L}{\partial z} = 0$, which translates into

$$\alpha_1(\cosh(2x)\sinh(2z)\dot{y} + \cosh(2z)\dot{x}) + \alpha_2(\cosh(2x)\cosh(2z)\dot{y} + \sinh(2z)\dot{x}), \tag{5.94}$$

and which allows us to solve for $z$ implicitly:

$$\tanh(2z) = \frac{-(\alpha_2 \cosh(2x)\dot{y} + \alpha_1 \dot{x})}{\alpha_1 \cosh(2x)\dot{y} + \alpha_2 \dot{x}}. \tag{5.95}$$

Reinserting into the action (and dropping total derivatives), we arrive at (see, also, [104, eq.(5.10)])

$$S'_\alpha = \int_a^b \left( \sqrt{\alpha_1^2 - \alpha_2^2} \sqrt{-\dot{x}^2 + \cosh(2x)^2 \dot{y}^2} - \alpha_3 \sinh(2x)\dot{y} \right) dt. \tag{5.96}$$

We recognise the first term as the line element in $\mathsf{dS}_2$ with metric

$$ds^2 = (\alpha_1^2 - \alpha_2^2)(-dx^2 + \cosh(2x)^2 dy^2), \tag{5.97}$$

whereas the second term is the coupling to a Maxwell field

$$A = -\alpha_3 \sinh(2x)dy, \tag{5.98}$$

whose fieldstrength

$$F = dA = -2\alpha_3 \cosh(2x)dx \wedge dy = \frac{2\alpha_3}{\alpha_2^2 - \alpha_1^2} \mathrm{dvol}, \tag{5.99}$$

where dvol is now the area form of the de Sitter metric in equation (5.97).

5.4.3. *Particle dynamics on the lightcone.* Finally, we discuss the parabolic case. Again we write $g^{-1}dg = \theta^1 P_1 + \theta^2 P_2 + \theta^3 B$, where now

$$\begin{aligned} \theta^1 &= e^{-2x}dy \\ \theta^2 &= dx - ze^{-2x}dy \\ \theta^3 &= dz - 2zdx + z^2 e^{-2x}dy. \end{aligned} \tag{5.100}$$

There is no $\mathrm{SL}(2,\mathbb{R})$-invariant metric here, but only a carrollian structure $(\kappa, \eta)$, where the carrollian vector field is $\kappa = z\partial_x + e^{2x}\partial_y$ and the carrollian degenerate metric is given by $\eta = (dx - ze^{-2x}dy)^2$.

The action is now given by

$$S_\alpha = \int_a^b \left( \alpha_1 e^{-2x}\dot{y} + \alpha_2(\dot{x} - ze^{-2x}\dot{y}) + \alpha_3(\dot{z} - 2z\dot{x} + z^2 e^{-2x}\dot{y}) \right) dt. \tag{5.101}$$

The Euler–Lagrange equation for $z$ is again simply $\frac{\partial L}{\partial z} = 0$, which is easily solved for $z$:

$$z = \frac{\alpha_2}{2\alpha_3} + e^{2x}\frac{\dot{x}}{\dot{y}}. \tag{5.102}$$

Reinserting into the action (and dropping total derivatives), we arrive at

$$S'_\alpha = \int_a^b \left( \left( \alpha_1 - \frac{\alpha_2^2}{4\alpha_3} \right) e^{-2x}\dot{y}^2 - \alpha_3 e^{2x}\dot{x}^2 \right) \frac{dt}{\dot{y}}. \tag{5.103}$$

Choosing the "static gauge" where $\dot{y} = 1$ and changing variables to $u = e^x$, we arrive at the following action

$$S''_\alpha = \int_a^b \left( \left( \alpha_1 - \frac{\alpha_2^2}{4\alpha_3} \right) u^{-2} - \alpha_3 \dot{u}^2 \right) dt, \tag{5.104}$$

which we recognise as a version of the one-dimensional conformal mechanics of [44].

5.4.4. *One-dimensional Schwarzian particle.* Here we will rederive the $\mathrm{SL}(2,\mathbb{R})$-invariant Schwarzian action of [45–47] using the method of nonlinear realisations and the inverse Higgs mechanism applied to $\mathrm{SL}(2,\mathbb{R})$. An alternative derivation using nonlinear realisations for $\mathrm{SL}(2,\mathbb{R}) \times \mathbb{R}^+$ can be found in [105].

Let $\mathbb{RP}^1$ denote the real projective line: the space of straight lines through the origin in the plane $\mathbb{R}^2$. The real projective line is diffeomorphic to the circle. Given a diffeomorphism $\varphi: \mathbb{RP}^1 \to \mathbb{RP}^1$, we define its *Schwarzian derivative* (or simply its *Schwarzian*) by the formula

$$\mathrm{Sch}(\varphi) := \frac{\varphi'''}{\varphi'} - \frac{3}{2}\left( \frac{\varphi''}{\varphi'} \right)^2, \tag{5.105}$$

where the primes represent derivatives with respect to the local coordinate on $\mathbb{RP}^1$. The Schwarzian defines a quadratic differential on $\mathbb{RP}^1$ or, in physical terms, a quasiprimary field with weight 2 under diffeomorphisms of the circle and it plays an important rôle in projective geometry (see, e.g., [106]). One of its most important properties is its invariance under $\mathrm{PSL}(2,\mathbb{R})$ Möbius transformations:

$$\varphi \mapsto \frac{a\varphi + b}{c\varphi + d} \qquad \text{with} \qquad ad - bc = 1. \tag{5.106}$$

We will re-use the basis for $\mathfrak{sl}(2, \mathbb{R})$ in Section 5.4.3, but with a change of notation to reflect that $SL(2, \mathbb{R})$ is the one-dimensional group of conformal transformations. Therefore we introduce the basis $H, K, D$ for $\mathfrak{sl}(2, \mathbb{R})$ where

$$K = \begin{pmatrix} 0 & 1 \\ 0 & 0 \end{pmatrix}, \qquad D = \begin{pmatrix} 1 & 0 \\ 0 & -1 \end{pmatrix} \qquad \text{and} \qquad H = \begin{pmatrix} 0 & 0 \\ 1 & 0 \end{pmatrix}. \tag{5.107}$$

Here $H$ generates translations, $K$ generates special conformal transformations and $D$ generates dilatations, which are all the one-dimensional conformal transformations. The ad-invariant inner product on $\mathfrak{sl}(2, \mathbb{R})$, which is a multiple of the Killing form, can be normalised to $\langle D, D \rangle = 2$ and $\langle H, K \rangle = 1$ in this basis.

We will choose a local chart $(\rho, y, u)$ for $SL(2, \mathbb{R})$ different than the one we introduced in Section 5.4.3 to derive the lagrangian for one-dimensional conformal mechanics. We shall parametrise group elements near the identity by

$$g = \underbrace{e^{\rho H}}_{g_0} e^{yK} e^{uD}, \tag{5.108}$$

where $g_0$ is a local coset representative for the one-dimensional conformal space thought of as the coset space $SL(2, \mathbb{R})/\mathscr{H}$, with $\mathscr{H}$ the two-dimensional non-abelian Lie group generated by $K$ and $D$. Explicitly, the above parametrisation is

$$g = \begin{pmatrix} e^u & e^{-u}y \\ e^u \rho & e^{-u}(1 + \rho y) \end{pmatrix}. \tag{5.109}$$

The Maurer–Cartan form is given by

$$\begin{aligned} \Omega = g^{-1}dg &= \Omega_H H + \Omega_D D + \Omega_K K \\ &= e^{2u}d\rho H + (du - yd\rho)D + e^{-2u}(dy - y^2 d\rho)K. \end{aligned} \tag{5.110}$$

In contrast to what we did in Section 5.4.3, the lagrangian here will not be linear in the components of the Maurer–Cartan form, but rather quadratic, resulting from applying the inverse Higgs mechanism to the lagrangian for geodesic motion on $SL(2, \mathbb{R})$ relative to the bi-invariant metric

$$\langle \Omega, \Omega \rangle = 2\Omega_D^2 + 2\Omega_H \Omega_K = 2\left(du^2 - 2ydud\rho + d\rho dy\right). \tag{5.111}$$

The geodesic lagrangian is obtained by pulling back the metric to the interval parametrising the world-line of the particle:

$$L = \tfrac{1}{2}\left\langle g^{-1}g', g^{-1}g' \right\rangle = u'^2 + (y' - 2yu')\rho', \tag{5.112}$$

where we are using primes to denote differentiation with respect to the parameter of the world-line of the particle. This lagrangian is invariant under both left and right multiplication by $SL(2, \mathbb{R})$. For example, under infinitesimal left multiplication with parameter $\alpha H + \beta D + \gamma K$, we have

$$\delta u = \beta + \gamma \rho, \qquad \delta \rho = \alpha - 2\beta\rho + \gamma\rho^2 \qquad \text{and} \qquad \delta y = \gamma + 2y\beta + 2\rho y\gamma. \tag{5.113}$$

We recognise in (5.113) the transformation of the Goldstone field $\rho$ under an infinitesimal Möbius transformation.

We can reduce the number of Goldstone fields in the action by imposing some conditions on the Maurer–Cartan form, a procedure also known as the inverse Higgs mechanism [90]. In the present context, the conditions are familiar from Drinfel'd–Sokolov reduction [107, 108] and are given by

$$\Omega_H = 1 \qquad \text{and} \qquad \Omega_D = 0. \tag{5.114}$$

This breaks the symmetry of the lagrangian under right multiplication, leaving only the global symmetry described infinitesimally in equation (5.113).

We can solve the constraints (5.114) explicitly for $u, y$ in terms of $\rho$:

$$y = \frac{u'}{\rho'} \qquad \text{and} \qquad u = \tfrac{1}{2}\log\left(\frac{1}{\rho'}\right). \tag{5.115}$$

It follows that

$$u' = -\tfrac{1}{2}\frac{\rho''}{\rho'} \qquad \text{and} \qquad y' = \frac{\rho''^2}{\rho'^3} - \frac{1}{2}\frac{\rho'''}{\rho'^2}. \tag{5.116}$$

Substituting this in the lagrangian (5.112), we obtain

$$L = -\frac{1}{2}\left(\frac{\rho'''}{\rho'} - \frac{3}{2}\left(\frac{\rho''}{\rho'}\right)^2\right) = -\tfrac{1}{2}\operatorname{Sch}(\rho). \tag{5.117}$$

In summary, it is possible to obtain the Schwarzian action using the inverse Higgs mechanism. It is also possible to obtain the Schwarzian action by integrating out the gauge transformations of the particle model with variables $x^\mu, \lambda$ and lagrangian $L = \tfrac{1}{2}\dot{x}^2 - \tfrac{1}{2}\lambda x^2$, as in [109, 110].

**5.5. Non-relativistic limit of relativistic particle actions.** In this section we obtain some of the non-lorenztian particle dynamics studied in the previous sections as non-relativistic limits of relativistic particles.

5.5.1. *Non-relativistic limit of the* $\mathsf{AdS}_{d+1}$ *particle action.* We consider the action of a massive particle propagating in $\mathsf{AdS}_{d+1}$, with metric

$$ds^2 = -\cosh^2\frac{r}{R}(dX^0)^2 + \left(\frac{\sinh\frac{r}{R}}{\frac{r}{R}}\right)^2(dX^a)^2 - \left(\left(\frac{\sinh\frac{r}{R}}{\frac{r}{R}}\right)^2 - 1\right)(dr)^2, \tag{5.118}$$

where $r = \sqrt{X_a X^a}$. The particle lagrangian is that describing geodesic motion on this geometry:

$$L = -m\sqrt{\cosh^2\frac{r}{R}(\dot{X}^0)^2 - \left(\frac{\sinh\frac{r}{R}}{\frac{r}{R}}\right)^2(\dot{X}^a)^2 + \left(\left(\frac{\sinh\frac{r}{R}}{\frac{r}{R}}\right)^2 - 1\right)(\dot{r})^2}. \tag{5.119}$$

In order to take the non-relativistic limit, we introduce an invertible change of variables with a dimensionless parameter $\omega$:

$$X^0 = \omega x^0, \qquad m = \omega M \qquad \text{and} \qquad R = \omega\tilde{R}. \tag{5.120}$$

After this change of variable, the lagrangian becomes

$$L = -M\omega^2\dot{x}^0 + \frac{M(\dot{x}^a)^2}{2\dot{x}^0} - \dot{x}^0\frac{Mr^2}{2R^2} + O(\omega^{-2}). \tag{5.121}$$

The omitted terms $O(\omega^{-2})$ will not contribute in the limit $\omega \to \infty$, but this limit is problematic due to the presence of a quadratically divergent term. This term may be cancelled if we introduce at the relativistic level a coupling to a constant electromagnetic field [13, 31] $A$ (with $F = dA = 0$) in such a way we preserve the same physical degrees of freedom:

$$L_{\text{em}} = A_\mu e^\mu, \quad A_\mu = (M\omega, \vec{0}), \tag{5.122}$$

where $e^\mu$ are the components of the vielbein of the metric (5.118).

Doing so and taking the limit $\omega \to \infty$, the lagrangian becomes

$$L = \frac{M(\dot{x}^a)^2}{2\dot{x}^0} - \dot{x}^0\frac{Mr^2}{2R^2}, \tag{5.123}$$

which takes the expected form of the reparametrization invariant $\mathfrak{n}^-$-particle lagrangian in equation (5.38).

5.5.2. *Massless galilean particle and the non-relativistic limit of a tachyon.* Now we will consider the non-relativistic limit of a tachyon. We start with the relativistic canonical action of a tachyon of mass m

$$S = \int d\tau \left(p_A\dot{x}^A - \frac{e}{2}\left(p^2 - m^2c^2\right)\right). \tag{5.124}$$

The non-relativistic limit is defined by taking $c \to \infty$ in

$$x^0 = ct, \qquad \text{and} \quad p_0 = -\frac{E}{c} \tag{5.125}$$

while keeping the *colour* $k = mc$ finite. The action becomes [111]

$$S = \int d\tau \left(-E\dot{t} + \vec{p}\cdot\dot{\vec{x}} - \frac{e}{2}\left(\vec{p}^2 - k^2\right)\right). \tag{5.126}$$

If we eliminate the momenta we have

$$S = \int d\tau \left(-E\dot{t} + \frac{k}{2}\sqrt{\dot{\vec{x}}^2}\right). \tag{5.127}$$

In this form the action can be interpreted a relativistic tachyonic particle with an instantaneous interaction [33]. The field theory associated to this particle model is the galilean magnetic Klein–Gordon field theory as in equation (7.4).

The non-relativistic limit of the one-dimensional conformal mechanics and of the Schwarzian particle has been studied in [112, 113].

**5.6. Carrollian limits of particle actions.** In this section we obtain some of the non-lorenztian particle dynamics studied in the previous sections as carrollian limits of relativistic particles.

5.6.1. *Carrollian limit of a massive particle in* $\mathsf{AdS}_{d+1}$ *background.* We consider the canonical action of a massive particle in $\mathrm{AdS}_{d+1}$ background

$$S = \int d\tau \left( p_\mu \dot{x}^\mu - \frac{e}{2} \left( g^{\mu\nu} p_\mu p_\nu - m^2 \right) \right), \tag{5.128}$$

where $g^{\mu\nu}$ is the inverse metric of (5.118).

The carrollian limit is defined by taking $\omega \to \infty$ in

$$x^0 = \frac{t}{\omega}, \qquad p_0 = -\omega E,, \qquad \text{and} \qquad m = M\omega, \tag{5.129}$$

and keeping $R$ fixed. It is understood that, before taking the limit, we rescale the Einbein variable like

$$e \to \frac{-e}{\omega^2}. \tag{5.130}$$

The carrollian action is given by

$$S_C = \int d\tau \left( -E\dot{t} + \dot{\vec{x}} \cdot \vec{p} - \frac{e}{2} \left( \frac{E^2}{\cosh^2 \frac{r}{R}} - M^2 \right) \right). \tag{5.131}$$

A particle in AdS Carroll does not move. The Carroll particle in flat space time is obtained by sending $R \to \infty$ and it can be written as

$$S = \int d\tau (-M\sqrt{\dot{t}^2} + M\vec{p}\dot{\vec{x}}), \tag{5.132}$$

which can be interpreted[18] as a timelike relativistic particle which is at rest in a given point in space: $\dot{\vec{x}} = 0$ [33]. The field theory associated to this particle model is the Carroll electric Klein–Gordon field theory as in equation (7.18) [97].

The massless Carroll particle is obtained by putting $M = 0$.

5.6.2. *Carrollian limit of relativistic tachyon.* We consider the action of a relativistic tachyon in configuration space the action is given by

$$S = -mc \int d\tau \sqrt{(\dot{\vec{x}})^2 - (\dot{x}^0)^2}. \tag{5.133}$$

In order to take the carrollian limit, it is useful to introduce the carrollian time and mass $M$ given by

$$s = C, \qquad x^0 = Cct \qquad \text{and} \qquad mc = MC. \tag{5.134}$$

Substituting in the action, we obtain

$$S = MC \int d\tau \sqrt{\dot{\vec{x}}^2 - \frac{\dot{s}^2}{C^2}}. \tag{5.135}$$

The Carroll limit in these variables is given by taking

$$s \to \infty \qquad \text{and} \qquad MC \to \tilde{M}. \tag{5.136}$$

The Carroll action of a tachyon is given by [98]

$$L = -\tilde{M}\sqrt{(\dot{\vec{X}})^2}. \tag{5.137}$$

The canonical action is given by

$$S = \int d\tau \left( -E\dot{t} + \vec{p} \cdot \dot{\vec{x}} - \frac{e}{2} \left( \vec{p}^2 - \tilde{M}^2 \right) - \mu E \right). \tag{5.138}$$

The quantisation of the constraints gives the Helmholtz equation

$$(\nabla^2 + M^2)\Phi(t, \vec{x}) = 0 \qquad \text{and} \qquad \frac{\partial}{\partial t}\Phi(t, \vec{x}) = 0. \tag{5.139}$$

Note that the Helmholtz equation as also appearing the quantisation of massless galilean particles. The relation among Galilei and Carroll particles in its $v/c$ coorrections is analysed in [99].

This ends our discussion of the dynamics of non-lorentzian particles.

---

[18]We acknowledge discussions with Roberto Casalbuoni on this point.

## 6. Gravity

This section contains three subsections. In the first two subsections we will describe gravity from a kinematical point of view by a gauging procedure that uses the Lie algebra of symmetries that underlies the theory as a starting point. We call it a gauging procedure because there are additional steps involved as compared to gauging a Lie algebra of internal symmetries leading to Yang-Mills which makes the relation between the final result and the original Lie algebra less direct (see, e.g., [114]). In the first subsection we explain this gauging procedure for the relativistic case while in the second subsection we will focus on three non-Lorentzian algebras: the Bargmann algebra underlying Newton-Cartan gravity, the Galilei algebra and the Carroll algebra. In the third subsection we will describe Newton-Cartan gravity from a dynamical point of view by defining a suitable non-relativistic limit of the Einstein equations of motion. Next, we will discuss the non-Lorentzian gravity theories underlying the Galilei and Carroll algebras, called Galilei gravity and Carroll gravity, respectively.

6.1. **Gauging the Poincaré algebra.** We first consider the relativistic case. Our starting point is the Poincaré algebra

$$[\mathsf{P}_A, \mathsf{P}_B] = 0, \qquad (6.1a)$$
$$[\mathsf{M}_{AB}, \mathsf{P}_C] = 2\eta_{C[B}\mathsf{P}_{A]}, \qquad (6.1b)$$
$$[\mathsf{M}_{AB}, \mathsf{M}_{CD}] = 4\eta_{[A[C}\mathsf{M}_{D]B]}, \qquad (6.1c)$$

where $\mathsf{P}_A$ and $\mathsf{M}_{AB}$ are the generators of spacetime translations and Lorentz transformations, respectively. The capital indices run over $A = 0, ..., d$ and we have chosen the Minkowski metric to have mostly plus signature. In this subsection we will apply a gauging procedure to this Poincaré algebra keeping the relativistic symmetries intact.

As a first step in the gauging procedure, we associate to the translation and Lorentz rotation generators the independent gauge fields $\mathsf{E}_\mu{}^A$ and $\Omega_\mu{}^{AB}$ which we call the Vierbein and Lorentz spin-connection, respectively:

$$A_\mu^I T_I = \mathsf{E}_\mu{}^A \mathsf{P}_A + \frac{1}{2}\Omega_\mu{}^{AB}\mathsf{M}_{AB}. \qquad (6.2)$$

These gauge fields transform as covariant vectors under a general coordinate transformation with parameter $\xi^\mu$ while their P-transformations corresponding to the translation generators $\mathsf{P}_A$, with parameters $\eta^A$, and their Lorentz transformation rules corresponding to the Lorentz generators $\mathsf{M}_{AB}$, with parameters $\Lambda^{AB}$, follow from the structure constants $f^I{}_{JK}$ of the Poincaré algebra:

$$\delta A_\mu^I = \xi^\lambda \partial_\lambda A_\mu^I + \partial_\mu \xi^\lambda A_\lambda^I + \partial_\mu \Lambda^I - f^I{}_{JK}\Lambda^J A^K \qquad (6.3)$$

or

$$\delta \mathsf{E}_\mu{}^A = \xi^\lambda \partial_\lambda \mathsf{E}_\mu{}^A + \partial_\mu \xi^\lambda \mathsf{E}_\lambda{}^A + \partial_\mu \eta^A - \Omega_\mu{}^A{}_B \eta^B + \Lambda^A{}_B \mathsf{E}_\mu{}^B, \qquad (6.4a)$$
$$\delta \Omega_\mu{}^{AB} = \xi^\lambda \partial_\lambda \Omega_\mu{}^{AB} + \partial_\mu \xi^\lambda \Omega_\lambda{}^{AB} + \partial_\mu \Lambda^{AB} + 2\Lambda^{C[A}\Omega_\mu{}^{B]}{}_C. \qquad (6.4b)$$

We now wish to argue that in the context of general relativity, the general coordinate transformations, Lorentz rotations and P-transformations do not define three independent symmetries of the Einstein-Hilbert action. To write down such an Einstein-Hilbert action we first define the curvature tensors associated to each gauge field as follows:

$$R_{\mu\nu}{}^I(T) = 2\partial_{[\mu}A_{\nu]}^I + \frac{1}{2}f^I{}_{JK}A_\mu^J A_\nu^K \qquad (6.5)$$

or

$$R_{\mu\nu}{}^A(P) = 2\partial_{[\mu}\mathsf{E}_{\nu]}{}^A - 2\Omega_{[\mu}{}^A{}_B \mathsf{E}_{\nu]}{}^B \qquad (6.6)$$
$$R_{\mu\nu}{}^{AB}(M) = 2\partial_{[\mu}\Omega_{\nu]}{}^{AB} + 2\Omega_{[\mu}{}^{AC}\Omega_{\nu]}{}^B{}_C. \qquad (6.7)$$

The Ricci tensor and Ricci scalar are defined by

$$R_\mu{}^A(M) = -\mathsf{E}^\nu{}_C R_{\mu\nu}{}^{CA}(M), \qquad R(M) = \mathsf{E}^\mu{}_A R_\mu{}^A(M), \qquad (6.8)$$

where we have used the inverse Vierbein field $\mathsf{E}^\mu{}_A$ defined by

$$\mathsf{E}^\mu{}_A \mathsf{E}_\mu{}^B = \delta_A{}^B, \qquad \mathsf{E}^\mu{}_A \mathsf{E}_\nu{}^A = \delta_\nu{}^\mu. \qquad (6.9)$$

We now consider the Einstein-Hilbert action (without cosmological constant)

$$S_{EH} = \frac{1}{16\pi G_N}\int d^{d+1}x\, \mathsf{E}R(M), \qquad (6.10)$$

where $E$ is the determinant of the Vierbein field $E_\mu{}^A$ and $G_N$ is Newton's constant. By construction this action is invariant under general coordinate transformations and Lorentz rotations. However, except for $d = 2$, it is not manifestly invariant under the P-transformations given in equation (6.4a) of the Poincaré algebra. This can for instance be seen by writing the Einstein-Hilbert action (6.10) in the equivalent form [19]

$$S_{EH} = \frac{1}{16\pi G_N} \int d^{d+1}x \, \epsilon^{\mu_0 \cdots \mu_d} \epsilon_{A_0 \cdots A_d} E_{\mu_0}{}^{A_0} \cdots E_{\mu_{d-2}}{}^{A_{d-2}} R_{\mu_{d-1} \mu_d}{}^{A_{d-1} A_d}(M) \,. \tag{6.12}$$

The special thing about $d = 2$ is that the Einstein-Hilbert action as given in (6.12) reduces to the Chern-Simons form

$$S_{EH} = \frac{1}{16\pi G_N} \int d^3x \, \epsilon^{\mu\nu\rho} \epsilon_{ABC} E_\mu{}^A R_{\nu\rho}{}^{BC}(M) \,, \tag{6.13}$$

which is manifestly invariant under all the gauge symmetries of the three-dimensional Poincaré algebra. In $d = 3$, one could consider, besides the term

$$\epsilon^{\mu\nu\rho\sigma} \epsilon_{ABCD} E_\mu{}^A E_\nu{}^B R_{\rho\sigma}{}^{CD}(M) \tag{6.14}$$

given in (6.12) the so-called Holst term [115]

$$\alpha \epsilon^{\mu\nu\rho\sigma} E_\mu{}^A E_\nu{}^B R_{\rho\sigma}{}^{AB}(M) \,, \tag{6.15}$$

where $\alpha$ is a real parameter. The two terms together give rise to the usual Einstein equations. This can be seen by first noting that varying the action with respect to the spin-connection gives the same equation of motion as without the Holst term. This follows from the following identity:

$$X^{AB} + \alpha \epsilon^{ABCD} X_{CD} = 0 \qquad \rightarrow \qquad X^{AB} = 0 \,, \tag{6.16}$$

where $X^{AB}$ is a three-form given by

$$X_{\mu\nu\rho}^{AB} = R_{[\mu\nu}{}^{[A}(P) E_{\rho]}{}^{B]} \,. \tag{6.17}$$

The field equation $X^{AB} = 0$ implies $R_{\mu\nu}{}^A(P) = 0$ which is a curvature constraint that can be used to solve for the spin-connection as will be explained below, see the solution given in equation (6.29). Next, varying the action with respect to the Vierbein, the Holst term does not contribute to the equations of motion for a dependent spin-connection due to the Riemann tensor identity $R_{[ABC]D}(M) = 0$.

Although, for $d > 2$, the Einstein-Hilbert action is not invariant under P-transformations, it does transform into terms that vanish upon using the equation of motion of the spin connection field which is given by

$$R_{\mu\nu}{}^A(P) = 0 \,. \tag{6.18}$$

Such a variation can always be cancelled by adding terms to the P-transformation rule of the spin connection. After a long calculation one finds the result given in equation (6.26). Note that, except for the first term, all terms in the transformation rule (6.26) are proportional to the Ricci tensor and Ricci scalar and therefore vanish upon using the equations of motion corresponding to the inverse Vierbein field $E^\mu{}_A$, i.e. the Einstein equations:

$$R_\mu{}^A(M) - 2E_\mu{}^A R(M) = 0 \,. \tag{6.19}$$

One thus ends up with a set of P-transformations that do not straightforwardly follow from the Poincaré algebra. Instead of doing the long calculation mentioned above to obtain the transformation rule (6.26), there is an easier way to derive the P-transformations of the spin-connection field by making use of the fact that these P-transformations are not new but, instead, related to the general coordinate transformations and the Lorentz transformations of the Poincaré algebra. To show this relation we need to make use of a special symmetry which in the literature is called a 'trivial' or 'equation of motion' symmetry (see, e.g., [116]. These symmetries, which are easier to derive than the P-transformation of the spin-connection field, are called 'trivial' because they have the distinguishing feature that all terms in the transformation rules vanish upon using the equations of motion. They therefore correspond to vanishing Noether charges. A most simple example of a trivial symmetry is provided by the following action describing two real Klein-Gordon scalars $A$ and $B$:

$$S = \int d^{d+1}x \, \frac{1}{2} \big( A \square A + B \square B \big) \,. \tag{6.20}$$

This action is invariant under the trivial symmetries with parameter $\lambda$

$$\delta A = \lambda \square B \,, \qquad\qquad \delta B = -\lambda \square A \,. \tag{6.21}$$

---

[19]The equivalence between the expressions (6.10) and (6.12) can be seen by writing out the definition of the determinant $E$ in equation (6.10):

$$E = \frac{1}{(d+1)!} \epsilon^{\mu_0 \cdots \mu_d} \epsilon_{A_0 \cdots A_d} E_{\mu_0}{}^{A_0} \cdots E_{\mu_d}{}^{A_d} \,. \tag{6.11}$$

We can see this by writing

$$\delta S = \lambda \frac{\delta S}{\delta \phi^i} \Omega^{ij} \frac{\delta S}{\delta \phi^j} = 0 \tag{6.22}$$

for $\phi^i = (A, B)$ and using the fact that $\Omega^{ij} = \epsilon^{ij}$ is anti-symmetric.

Similarly, the Einstein-Hilbert action is invariant under the following trivial symmetries with parameters $\sigma^A$:

$$\delta E_\mu{}^A = R_{\mu\nu}{}^A(P)\sigma^\nu, \tag{6.23a}$$

$$\delta \Omega_\mu{}^{AB} = -R_\mu{}^{[A}(M)\sigma^{B]} - \frac{1}{2}E_\mu{}^{[A}R_C{}^{B]}(M)\sigma^C + \frac{3}{4}E_\mu{}^{[A}R(M)\sigma^{B]}, \tag{6.23b}$$

with $\sigma^\nu \equiv \sigma^B E^\nu{}_B$. Like in the example of the two scalar fields the Vierbein field transforms to the equation of the spin-connection field while the spin-connection field transforms to the equation of motion of the Vierbein field leading to a zero variation of the action as follows:

$$\delta S \sim \begin{pmatrix} \frac{\delta S}{\delta E_\mu{}^A} & \frac{\delta S}{\delta \Omega_\rho{}^{CD}} \end{pmatrix} \begin{pmatrix} 0 & E^\mu{}_C E^\rho{}_A \\ -E^\mu{}_C E^\rho{}_A & 0 \end{pmatrix} \begin{pmatrix} \frac{\delta S}{\delta E_\mu{}^A} \\ \frac{\delta S}{\delta \Omega_\rho{}^{CD}} \end{pmatrix} \sigma_D = 0. \tag{6.24}$$

Using these trivial symmetries, we can write the P-transformation given in equation (6.4a) of the Vierbein field as the sum of a special general coordinate transformation, Lorentz transformation and trivial symmetry transformation with parameters given by

$$\xi^\mu = \eta^\mu, \qquad \Lambda^{AB} = \eta^\lambda \Omega_\lambda{}^{AB}, \qquad \sigma^A = \eta^A, \tag{6.25}$$

with $\eta^\mu \equiv \eta^B E^\mu{}_B$. Since the same decomposition rule must apply to the spin-connection field, it follows that the P-transformation of this spin-connection field is given by

$$\delta_\eta \Omega_\mu{}^{AB} = \eta^\lambda R_{\lambda\mu}{}^{AB}(M) + R_\mu{}^{[A}(M)\eta^{B]} + E_\mu{}^{[A}R_C{}^{B]}(M)\eta^C + E_\mu{}^{[A}R(M)\eta^{B]}, \tag{6.26}$$

which is the same expression that one obtains by requiring that the Einstein-Hilbert term is invariant under P-transformations.

Summarizing, the P-transformations given in eqs. (6.4a), (6.4b), (6.26) and the general coordinate transformations given in the same equations equation (6.4a) and (6.4b) do not define two independent symmetries of the first-order Einstein-Hilbert action (6.10). In fact, if they would be independent symmetries, the theory would have no propagating degrees of freedom left. Both symmetries have their advantages. On the one hand, the general coordinate transformations have a nice geometrical interpretation, but, on the other hand, the P-transformations are more directly related to the underlying Poincaré algebra.

When taking the non-relativistic limit of general relativity in subsection 6.3 , we prefer to work with the second-order formulation of general relativity. The reason for this is that for matter-coupled gravity theories, such as supergravity, it is more convenient to work in such a second-order formulation. In that case, it is understood that the equation of motion (6.18) has been used to solve for the spin-connection field $\Omega$ in terms of the Vierbeine $E_\mu{}^A$ and their inverses $E^\mu{}_A$. To solve this constraint it is convenient to introduce the notation

$$E_{\mu\nu}{}^A \equiv \partial_{[\mu}E_{\nu]}{}^A \tag{6.27}$$

and to write the constraint (6.18) in terms of flat indices as

$$2E_{ABC} - \Omega_{ACB} + \Omega_{BCA} = 0. \tag{6.28}$$

Following the solution of the Christoffel symbol in general relativity, we add three times this equation with the flat indices cyclic interchanged and multiply one of the three equations with a minus sign. Adding up the three resulting equations leads to the solution

$$\Omega_{ABC} = 2E_{A[BC]} + E_{BCA} \quad \text{or} \quad \Omega_\mu{}^{AB} = -2E_\mu{}^{[AB]} + E^{AB}{}_\mu. \tag{6.29}$$

The independent fields are then given by the Vierbein fields $E_\mu{}^A$ only. They transform under general coordinate transformations and local lorentz rotations as follows:

$$\delta E_\mu{}^A = \xi^\lambda \partial_\lambda E_\mu{}^A + \partial_\mu \xi^\lambda E_\lambda{}^A + \Lambda^A{}_B E_\mu{}^B. \tag{6.30}$$

The general coordinate transformations are not affected by the NR limit we consider in subsection 6.3, they are the same before and after taking the limit.

6.2. **Gauging non-Lorentzian algebras.** We next consider the non-Lorentzian case. There are several non-Lorentzian algebras we could consider. As specific examples we will consider the Galilei algebra, its central extension called the Bargmann algebra and the so-called Carroll algebra.

**The Galilei algebra.** Before discussing the Bargmann algebra that underlies the symmetries of NC gravity, we will first, as a warming up exercise, shortly discuss the special case of the Bargmann algebra with *zero* central extension, i.e. the Galilei algebra. In the next Chapter, we will show how the symmetries corresponding to the Galilei algebra arise if one takes the so-called Galilei limit of a real Klein-Gordon scalar field. Here, we will show how the Galilei algebra can be obtained as a particular contraction of the Poincaré algebra and how the structure constants of this Galilei algebra fix the transformation rules of the gauge fields under the Galilei symmetries.

To show how the Galilei algebra is obtained by a contraction of the Poincaré algebra, we first decompose the relativistic flat Lorentz index $A$ into $A = \{0, a\}$ with $a = (1, \dots, d)$, and redefine, using a contraction parameter $\omega$, the Poincaré generators according to

$$P_0 = \omega^{-1} H, \tag{6.31}$$

$$J_{0a} = \omega G_a, \tag{6.32}$$

where $H$ and $G_a$ are the generators of time translations and boosts, respectively. The generators $P_a$ of space translations and $J_{ab}$ of spatial rotations are not redefined. Next, taking the limit $\omega \to \infty$ we obtain the following Galilei algebra:

$$[J_{ab}, P_c] = 2\delta_{c[a}P_{b]}, \qquad\qquad [J_{ab}, G_c] = 2\delta_{c[a}G_{b]},$$

$$[J_{ab}, J_{cd}] = 4\delta_{[a[d}J_{c]b]}, \qquad\qquad [H, G_a] = P_a. \tag{6.33}$$

Following [117], we next associate to each generator of the Galilei algebra a gauge field as follows:

$$A_\mu^I T_I = \tau_\mu H + e_\mu{}^a P_a + \frac{1}{2}\omega_\mu{}^{ab}J_{ab} + \omega_\mu{}^a G_a. \tag{6.34}$$

Using the general formula (6.3), the gauge fields transform as 1-forms under general coordinate transformations while under spatial rotations with parameters $\lambda^a{}_b$ and Galilean boosts with parameters $\lambda^a$ they transform as follows:

$$\delta\tau_\mu = 0, \tag{6.35}$$

$$\delta e_\mu{}^a = \lambda^a \tau_\mu + \lambda^{ab} e_\mu{}^b, \tag{6.36}$$

$$\delta\omega_\mu{}^{ab} = (D_\mu \lambda)^{ab}, \tag{6.37}$$

$$\delta\omega_\mu{}^a = (D_\mu \lambda)^a + \lambda^a{}_b \omega_\mu{}^b. \tag{6.38}$$

Here $D_\mu$ is the covariant derivative with respect to spatial rotations, e.g., $(D_\mu \lambda)^a = \partial_\mu \lambda^a - \omega_\mu{}^{ab}\lambda^b$.

**The Bargmann algebra.** Our starting point is now the centrally extended Galilei algebra which is called the Bargmann algebra. The reason that we need to add one more generator to the Galilei algebra, which has the same number of generators as the Poincaré algebra, is that in the relativistic case energy is equivalent to mass but in the non-relativistic case energy and mass are two separately conserved quantities. The corresponding Noether symmetries lead to two generators in the Bargmann algebra: the time translation generator corresponding to the conservation of energy and the central charge or mass generator corresponding to the conservation of mass.

The Bargmann algebra can be obtained by performing a special Wigner-Inönü contraction of the direct product of the Poincaré algebra given in equation (6.1) with a U(1) algebra with generator $Z$. As a first step we make the following invertable redefinition of the relativistic generators

$$P_0 = \frac{1}{2\omega}H + \omega Z, \qquad M_{ab} = J_{ab}, \qquad M_{a0} = \omega G_a,$$

$$P_a = P_a, \qquad\qquad Z = \frac{1}{2\omega}H - \omega Z, \tag{6.39}$$

where $\omega$ is a (dimensionless) contraction parameter and where we have decomposed the flat space-time index $A$ into a time-like 0-index and spatial $a$-indices as $A = (0, a)$. Note the off-diagonal nature of the redefinitions. Would we have restricted to rescaling each generator separately, we would only be able to obtain the Galilei algebra times U(1). Using the redefinition (6.39), we find that the redefined generators,

after taking the limit that $\omega$ goes to infinity, generate the following Bargmann algebra:

$$[P_a, J_{bc}] = 2\,\delta_{a[b}\,P_{c]}\,, \qquad\qquad [J_{ab}, J_{cd}] = 4\,\delta_{[a[c}\,J_{d]b]}\,,$$
$$[G_a, J_{bc}] = 2\,\delta_{a[b}\,G_{c]}\,, \qquad\qquad\qquad [H, G_a] = P_a\,, \qquad\qquad [P_a, G_b] = \delta_{ab}\,Z\,, \tag{6.40}$$

where the generator $Z$ has taken the role of the central charge generator.

We next associate to each generator of the Bargmann algebra a gauge field as follows:

$$A_\mu^I T_I = \tau_\mu H + e_\mu{}^a P_a + \frac{1}{2}\omega_\mu{}^{ab} J_{ab} + \omega_\mu{}^a G_a + m_\mu Z\,. \tag{6.41}$$

Using the general formula (6.3), the gauge fields transform as 1-forms under general coordinate transformations while under spatial rotations with parameters $\lambda^a{}_b$, Galilean boosts with parameters $\lambda^a$ and central charge transformations with parameter $\sigma$ they transform as follows:

$$\delta\tau_\mu = 0\,,$$
$$\delta e_\mu{}^a = \lambda^a{}_b\,e_\mu{}^b + \lambda^a\tau_\mu\,,$$
$$\delta\omega_\mu{}^{ab} = \partial_\mu\lambda^{ab} + 2\,\lambda^{[a}{}_c\,\omega_\mu{}^{cb]}\,, \tag{6.42}$$
$$\delta\omega_\mu{}^a = \partial_\mu\lambda^a + \lambda^a{}_b\,\omega_\mu{}^b - \omega_\mu{}^a{}_c\,\lambda^c\,,$$
$$\delta m_\mu = \partial_\mu\sigma + \lambda_a\,e_\mu{}^a\,.$$

Using the general formula (6.5) the curvatures corresponding to these gauge fields that transform covariantly under these symmetries are given by

$$R_{\mu\nu}(H) \quad = \quad 2\partial_{[\mu}\tau_{\nu]}\,, \tag{6.43}$$

$$R_{\mu\nu}{}^a(P) \quad = \quad 2\partial_{[\mu}e_{\nu]}{}^a - 2\omega_{[\mu}{}^{ab}e_{\nu]b} - 2\omega_{[\mu}{}^a\tau_{\nu]}\,, \tag{6.44}$$

$$R_{\mu\nu}{}^{ab}(J) \quad = \quad 2\partial_{[\mu}\omega_{\nu]}{}^{ab} + 2\omega_{[\mu}{}^{ac}\omega_{\nu]}{}^b{}_c\,, \tag{6.45}$$

$$R_{\mu\nu}{}^a(G) \quad = \quad 2\partial_{[\mu}\omega_{\nu]}{}^a - 2\omega_{[\mu}{}^a{}_b\omega_{\nu]}{}^b\,, \tag{6.46}$$

$$R_{\mu\nu}(Z) \quad = \quad 2\partial_{[\mu}m_{\nu]} - 2\omega_{[\mu}{}^a e_{\nu]a}\,. \tag{6.47}$$

Note that the curvature $R_{\mu\nu}(H)$ corresponding to $\tau_\mu$ does not contain any of the other gauge fields. It therefore can describe an *intrinsic torsion* [72]. Imposing a constraint on this curvature leads to a purely geometric constraint.[20] This is quite different from the conventional curvature constraints, to be discussed below, that will be used to solve some gauge fields in terms of the others. Instead of $R_{\mu\nu}(H)$ we will sometimes use a notation in terms of the torsion tensor

$$T_{\mu\nu} = \partial_{[\mu}\tau_{\nu]}\,. \tag{6.48}$$

One may distinguish between the following three different cases:[21]

$$T_{\mu\nu} = 0 \quad : \quad \text{zero torsion}\,, \tag{6.49}$$

$$T_{ab} = 0 \quad : \quad \text{twistless torsional}\,, \tag{6.50}$$

$$T_{\mu\nu} \neq 0 \quad : \quad \text{general torsion}\,. \tag{6.51}$$

We have used here the projective inverse NR Vierbeine $\tau^\mu$ and $e^\mu{}_a$ defined by

$$\tau_\mu\tau^\mu = 1\,, \qquad \tau_\mu e^\mu{}_a = \tau^\mu e_\mu{}^a = 0\,, \qquad e_\mu{}^a e^\nu{}_a + \tau_\mu\tau^\nu = \delta_\mu{}^\nu\,. \tag{6.52}$$

to convert curved indices into flat indices. For instance,

$$T_{ab} = e^\mu{}_a e^\nu{}_b T_{\mu\nu}\,. \tag{6.53}$$

The zero torsion case defines a Newtonian spacetime with a co-dimension 1 foliation or, equivalently, a preferred time direction $t$ given by $\tau_\mu = \partial_\mu t$. Any observer traveling along a curve $\mathcal{C}$ from a time slice $\Sigma_{t_A}$ at $t = t_A$ to a time slice $\Sigma_{t_B}$ at $t = t_B$ will measure a time difference $\Delta T$ given by

$$\Delta T = \int_{t_A}^{t_B} dx^\mu \tau_\mu = t_B - t_A \tag{6.54}$$

independent of the curve $\mathcal{C}$. The twistless torsional case leads to a spacetime with a hypersurface orthogonality condition of the clock fucction $\tau_\mu$. Such spacetimes are encountered in Lifschitz holography [75].

---

[20]The following discussion on the intrinsic torsion also applies when we gauge the Galilei algebra.

[21]One cannot impose $T_{0a} = 0$ since such a constraint is not invariant under Galilean boost transformations.

Using the projective inverses of the timelike and spatial Vierbein fields $\tau_\mu$ and $e_\mu{}^a$, the so-called 'conventional constraint' equations [22]

$$R_{\mu\nu}{}^a(P) = R_{\mu\nu}(Z) = 0 \tag{6.55}$$

provide precisely sufficient equations to solve the spin-connection fields for spatial rotations and Galilean boosts in terms of the other independent gauge fields. For the zero torsion case these gauge fields are solved, by doing a similar calculation as in the relativistic case (see after equation (6.28)), as follows:

$$\omega_\mu{}^{ab}(\tau, e, m) = -2e_\mu{}^{ab} + e_{\mu c}e^{abc} - \tau_\mu m^{ab}, \tag{6.56a}$$

$$\omega_\mu{}^a(\tau, e, m) = e_{\mu 0}{}^a - e_{\mu c}e_0{}^{ac} + m_\mu{}^a - \tau_\mu m^{a0}. \tag{6.56b}$$

Here we have defined

$$e_{\mu\nu}{}^a = \partial_{[\mu}e_{\nu]}{}^a, \qquad m_{\mu\nu} = \partial_{[\mu}m_{\nu]}. \tag{6.57}$$

Furthermore, we have again used the inverse NR Vierbeine to convert curved indices into flat indices. For instance

$$e_{\mu 0}{}^a = \tau^\nu e_{\mu\nu}{}^a. \tag{6.58}$$

We note that the transformation of the dependent spin-connection fields is identical to the transformations of the independent spin-connection fields as given in eq (6.42), i.e.

$$\delta\omega_\mu{}^{ab}(\tau, e, m) = \delta\omega_\mu{}^{ab}, \qquad\qquad \delta\omega_\mu{}^a(\tau, e, m) = \delta\omega_\mu{}^a. \tag{6.59}$$

This is due to the fact that the curvatures in the conventional constraint equations (6.55) do not transform to any of the other curvatures under spatial rotations, Galilean boosts and central charge transformations. From now on we will assume that the spin-connections are dependent fields but we will not indicate their dependence anymore. Finally, we note that, by solving the conventional constraints (6.55), we work by definition in a second-order formulation.

Sofar, we did not yet discuss the $P_A = (P_0, P_a)$-transformations with parameters $(\eta, \eta^a)$ of the gauge fields. According to the Bargmann algebra they are given by

$$\delta\tau_\mu = \partial_\mu\eta, \tag{6.60}$$

$$\delta e_\mu{}^a = \partial_\mu\eta^a - \omega_\mu{}^{ab}\eta_b, \tag{6.61}$$

$$\delta m_\mu = -\omega_{\mu a}\eta^a. \tag{6.62}$$

To show how these P-transformations are related to general coordinate transformations, we consider the following general identity valid for any Lie algebra with structure constants $f^I{}_{JK}$:

$$0 = \delta_{gct}(\xi^\lambda)A_\mu{}^I + \xi^\lambda R_{\mu\lambda}{}^I(T) - \sum_{\{J\}}\delta(\xi^\lambda A_\lambda{}^J)A_\mu{}^I, \tag{6.63}$$

where the index I labels the gauge fields $A_\mu{}^I$ and corresponding curvatures $R_{\mu\nu}{}^I(T)$ of the gauge algebra. The sum in the last term is over all gauge fields. To see how this identity works, let us set, for instance, $I = a$ for the $P_a$-transformations and consider the parameters

$$\xi^\lambda = \tau^\lambda\eta + e_a{}^\lambda\eta^a \quad \text{or} \quad \eta = \xi^\lambda\tau_\lambda, \ \eta^a = \xi^\lambda e_\lambda{}^a. \tag{6.64}$$

We can then bring the contribution of $e_\mu{}^a$ to the sum in the last term of (6.63) to the left-hand side of the equation to obtain the following relation between a $P_a$-transformation with parameter $\eta^a$ and a general coordinate transformation with parameter $\xi^\lambda = e_a{}^\lambda\eta^a$:

$$\delta_P(\eta^b)e_\mu{}^a = \delta_{gct}(\xi^\lambda)e_\mu{}^a + \xi^\lambda R_{\mu\lambda}{}^a(P) - \delta_M(\xi^\lambda\omega_\lambda{}^{ab})e_\mu{}^a. \tag{6.65}$$

The same kind of identity holds for each gauge field that transform under a P-transformation, i.e., in our case $\tau_\mu, e_\mu{}^a$ and $m_\mu$, see equation (6.60): one can relate the P-transformation of these gauge fields to a general coordinate transformation plus other symmetries of the Bargmann algebra by setting the curvature of these gauge fields to zero. Following this rule we precisely obtain the zero torsion constraint (6.49) and the two conventional constraints (6.55). Remarkably, these constraints allow us to solve for the remaining gauge fields, i.e. the two spin-connection fields, and hence, as dependent gauge fields, they automatically have a P-transformation that is related to a general coordinate transformation since this was already proven for all the independent gauge fields.

We note that for non-zero torsion, the conventional constraints (6.55) do not transform to each other anymore under all the symmetries of the theory. To achieve this, one needs to add to these conventional constraints additional (independent) torsion tensors with the correct transformation properties. This leads to a notion of *torsional* NC geometry that is discussed in [119].

---

[22]For the use of conventional constraints in gravity and supergravity, see, e.g., [116, 118].

**The Carroll algebra.** Carroll symmetries emerge if one considers an *ultra-relativistic* limit of general relativity which is the opposite of taking a NR limit. At first sight this seems a strange thing to do. However, Carroll symmetries have shown up in several recent investigations in different connections such as strong coupling limits of gravity [120, 121], flat space holography [122], black hole horizons [12], de Sitter cosmology and dark matter [98] and even fractons [81, 123]. Here, for completeness, we shortly discuss the gauging of the Carrol algebra and point out some differences with the Galilei algebra.

To define the contraction of the Poincaré algebra that gives rise to the Carroll algebra, we decompose the A-index into $A = \{0, a\}$ with $a = (1, \ldots, d)$, and redefine the Poincaré generators according to

$$P_0 = \omega H, \tag{6.66}$$

$$J_{0a} = \omega G_a, \tag{6.67}$$

where $H$ and $G_a$ are the generators of time translations and boosts, respectively. The generators $P_a$ of space translations and $J_{ab}$ of spatial rotations are not redefined. Next, taking the limit $\omega \to \infty$ we obtain the following Carroll algebra:

$$[J_{ab}, P_c] = 2\delta_{c[a}P_{b]}, \qquad [J_{ab}, G_c] = 2\delta_{c[a}G_{b]},$$

$$[J_{ab}, J_{cd}] = 4\delta_{[a[d}J_{c]b]}, \qquad [P_a, G_b] = \delta_{ab}H. \tag{6.68}$$

We next associate to each generator of the Carroll algebra a gauge field as follows:

$$A_\mu^I T_I = \tau_\mu H + e_\mu{}^a P_a + \frac{1}{2}\omega_\mu{}^{ab}J_{ab} + \omega_\mu{}^a G_a. \tag{6.69}$$

Using the general formula (6.3), the gauge fields transform as 1-forms under general coordinate transformations while under spatial rotations with parameters $\lambda^a{}_b$ and Carroll boosts with parameters $\lambda^a$ they transform as follows:

$$\delta\tau_\mu = e_\mu{}^a \lambda_a,$$

$$\delta e_\mu{}^a = \lambda^a{}_b e_\mu{}^b,$$

$$\delta\omega_\mu{}^{ab} = (D_\mu \lambda)^{ab}, \tag{6.70}$$

$$\delta\omega_\mu{}^a = (D_\mu \lambda)^a + \lambda^a{}_b \omega_\mu{}^b,$$

where $D_\mu$ is the covariant derivative with respect to spatial rotations. Note that, in contrast to the Galilei algebra, $\tau_\mu$ transforms under a boost transformation while $e_\mu{}^a$ is invariant. Another important difference with the Galilei algebra is that the Carroll algebra does not allow for a central extension.

Unlike the Galilei or Bargmann algebra, all Carroll curvatures contain a spin-connection field. A priori such curvatures are part of conventional constraints, needed to solve for the spin-connection fields, and, therefore cannot describe an intrinsic torsion like the tensor $T_{\mu\nu}$ in the Galilei and Bargmann case. However, given the $R_{\mu\nu}{}^a(P)$ curvature

$$R_{\mu\nu}{}^a(P) = e_{\mu\nu}{}^a - \omega_{[\mu}{}^{ab}e_{\nu]b}, \qquad e_{\mu\nu}{}^a = \partial_\mu e_\nu{}^a - \partial_\nu e_\mu{}^a, \tag{6.71}$$

it turns out that the following boost-invariant projection $K^{ab} = K^{ba}$ does not contain any spin-connection field:

$$K^{ab} = \tau^\mu e^{\nu(a}R_{\mu\nu}{}^{b)}(P) = \tau^\mu e^{\nu(a}e_{\mu\nu}{}^{b)}(P). \tag{6.72}$$

Using that $\tau^\mu e_\mu{}^a = 0$ one can show that $K^{ab}$ is nothing else as than the spatial components of the extrinsic curvature:

$$K_{ab} = e_a{}^\mu e_b{}^\nu K_{\mu\nu}, \qquad K_{\mu\nu} = \tau^\lambda \partial_\lambda h_{\mu\nu} + \partial_\mu \tau^\lambda h_{\lambda\nu} + \partial_\nu \tau^\lambda h_{\lambda\mu} \tag{6.73}$$

with $h_{\mu\nu} = e_\mu{}^a e_\nu{}^b \delta_{ab}$.

6.3. **Taking limits.** The aim of this section is to define the limits of general relativity that correspond to the non-Lorentzian algebras we defined in the previous section as Wigner-Inönü contractions of the Poincaré algebra. Our main target is the Bargmann algebra but, for completeness, we will also shortly discuss the limits corresponding to the Galilei and Carroll algebra leading to Galilei and Carroll gravity, respectively.

Generically, to define a limit in all three cases, we will perform the following two steps:

- we make an invertible field redefinition writing all relativistic fields in terms of the would-be fields of the limiting theory and a dimensionless contraction parameter $\omega$. The invertibility implies that the number of fields before and after taking the limit remains the same. The would-be limiting fields only become the true limiting fields after taking the NR limit in the second step. Before this step we are just rewriting the general relativity theory.

- Either in the action or equations of motion we take the limit that $\omega$ goes to $\infty$. We do not allow divergent terms in the action. A noteworthy feature of several of the limits that we will be taking is that they are based upon a cancellation of the leading divergence by different contributions. The limiting action is given by all terms of order $\omega^0$. Taking the limit of the equations of motion the resulting equations of motion are given by the terms of leading order in $\omega$. Independent of this we will also take the limit of the transformation rules.

One should distinguish between taking limits from making expansions. In an expansion each field is expanded as an infinite sum of terms of increasing powers of $\omega^{-1}$. The leading terms in such an expansion do not necessarily correspond a redefinition defining a limit. For instance, in a post-Newtonian expansion of general relativity one does not introduce the additional field $M_\mu$. Instead, $m_\mu$ occurs as the sub-leading term in an expansion of $E_\mu{}^0$. Some results about limits can, however, be read off from making an expansion. For instance, the first leading term in the expansion in $\omega$ of a relativistic Lagrangian is always invariant under the corresponding non-Lorentzian symmetry [48, 49].

**Galilei gravity.** We first consider the case of Galilei gravity. Using a first-order formulation, an invariant action for Galilei gravity can be obtained by taking a specific NR limit of the Einstein-Hilbert action. To define this limit, we redefine the gauge fields and symmetry parameters with a dimensionless parameter parameter $\omega$ as follows [49]:

$$E_\mu^0 = \omega\tau_\mu, \qquad \Omega_\mu^{0a} = \omega^{-1}\omega_\mu^a, \tag{6.74}$$

$$E_\mu^a = e_\mu^a, \qquad \Omega_\mu^{ab} = \omega_\mu^{ab}, \tag{6.75}$$

$$\Lambda^{0a} = \omega^{-1}\lambda^a, \quad \Lambda^{ab} = \lambda^{ab}. \tag{6.76}$$

Substituting the above field redefinitions into the Einstein-Hilbert action, redefining Newton's constant $G_N = \omega G_G$ and taking the $\omega \to \infty$ limit we end up with the following Galilei action

$$S_{\text{Gal}} = -\frac{1}{2\kappa} \int e R_{\mu\nu}{}^{ab}(J) e_a^\mu e_b^\nu, \tag{6.77}$$

where $\kappa = 8\pi G_G$ and $e = \det(\tau_\mu, e_\mu{}^a)$ is the non-relativistic determinant. The projective inverses $\tau^\mu$ and $e^\mu{}_a$ transform under the Galilei boosts and spatial rotations as follows:

$$\delta\tau^\mu = -\lambda^a e_a^\mu, \qquad\qquad \delta e_a^\mu = \lambda^{ab} e_b^\mu. \tag{6.78}$$

One may verify that the Galilei action (6.77) is not only Galilei invariant but it also has an emergent local scaling symmetry given by

$$\tau_\mu \to \lambda(x)^{-(d-2)}\tau_\mu, \qquad e_\mu^a \to \lambda(x)e_\mu^a, \tag{6.79}$$

where $\lambda(x)$ is an arbitrary function. This emergent local scaling symmetry implies that there is a so-called 'missing' equation of motion that does not follow from the variation of the Galilei action (6.77). This missing equation of motion can be obtained by taking the limit of the relativistic equations of motion. We will encounter a similar situation when discussing NC gravity below.

For any $d > 2$ the equations of motion that follow from the variation of the Galilei action (6.77) lead to the following constraint on the geometry

$$T_{ab} = e_a^\mu e_b^\nu T_{\mu\nu} = 0. \tag{6.80}$$

This constraint means that this geometry has twistless torsion.

For $d > 2$ the equations of motion can be used to solve for the spatial rotation spin connection $\omega_\mu{}^{ab}$ as

$$\omega_\mu^{ab} = \tau_\mu A^{ab} + e_{\mu c}\left(e^{\rho[a}e^{b]\nu}\partial_\rho e_\nu^c + e^{\rho[a}e^{c]\nu}\partial_\rho e_\nu^b - e^{\rho[b}e^{c]\nu}\partial_\rho e_\nu^a\right) + \frac{4}{D-3}e_\mu^{[a}\tau^{b]0}, \tag{6.81}$$

except for $A^{ab}$ which is an undetermined anti-symmetric tensor component of $\omega_\mu{}^{ab}$. In the second order formulation the constraint (6.80) arises from the variation with respect to $A^{ab}$. Hence, we can interpret $A^{ab}$ as a Lagrange multiplier. Indeed, in the case $d > 2$, plugging expression (6.81) into the action (6.77) to obtain it in a second order formulation leads to

$$S_{\text{Gal}} = -\frac{1}{2\kappa} \int e\left(R_{\mu\nu}{}^{ab}(J)e_a^\mu e_b^\nu\big|_{A^{ab}=0} + A^{ab}T_{ab}\right). \tag{6.82}$$

This makes manifest the fact that the variation with respect to $A^{ab}$ of the second order action in equation (6.82) reproduces the constraint (6.80).

The case $d = 2$ is special. In that case we may write $\omega_\mu{}^{ab} = \epsilon^{ab}\omega_\mu$ and it turns out that this $\omega_\mu$ cannot be determined from the field equations, i.e. there is no second-order formulation. Also, in contrast

to the $d > 2$ case, the equations of motion imply a stronger geometrical constraint, namely the zero torsion constraint

$$T_{\mu\nu} = 0 \,. \tag{6.83}$$

Using the identity $e\epsilon^{ab}e_a^\mu e_b^\nu = 2\epsilon^{\mu\nu\rho}\tau_\rho$, which is valid for $d = 2$, the Galilean action (6.77) can be rewritten as

$$S_{\text{Gal 3D}} = -\frac{1}{2\kappa} \int \epsilon^{\mu\nu\rho} \tau_\mu \partial_\nu \omega_\rho \,. \tag{6.84}$$

This form of the action makes manifest that its variation with respect to $\omega_\mu$ precisely reproduces the zero torsion constraint (6.83). We note that the Galilei algebra in $d = 2$ only allows for a degenerate invariant bilinear form. The above action corresponds to the Chern-Simons action for the Galilei algebra with this degenerate bilinear form. The degeneracy of the form explains why not all fields occur in the action.

**NC gravity**  We will now derive the equations of motion describing pure Newton-Cartan (NC) gravity in $d + 1$ dimensions by taking a specific non-relativistic (NR) limit of general relativity.

In the second-order formulation that we are using here, we need to express the relativistic Vierbein field $E_\mu{}^A$ into the would-be non-relativistic fields of NC gravity, that we described in the previous subsection, in an invertible way using a contraction parameter. Inspired by the standard Wigner-Inönu contraction of the Poincaré algebra we first write

$$E_\mu{}^0 = \omega\tau_\mu \,, \qquad\qquad E_\mu{}^a = e_\mu{}^a \,, \tag{6.85}$$

where we have decomposed $A = (0, a), \omega$ is a dimensionless parameter, $\tau_\mu$ is the clock function and $e_\mu{}^a$ are the rulers. It is clear that this limit cannot give rise to NC gravity because in the NR case energy is not the same as mass and hence we need two gauge fields, one for energy and one for mass, that in the previous subsection we called $\tau_\mu$ and $m_\mu$, respectively. Indeed, the NR limit defined by equation (6.85) gives rise to the Galilei gravity we discussed above. The additional mass operator gives rise to a central extension of the Galilei algebra called the Bargmann algebra. We saw in the previous subsection that, in order to obtain this Bargmann algebra from the Wigner-Inönü contraction of a relativistic algebra, we must extend the Poncaré algebra with an additional U(1) generator. In terms of gauge fields this implies that we should extend general relativity with an additional gauge field $M_\mu$ before taking the limit.[23] In order not to extend general relativity with extra degrees of freedom we impose by hand that the field equation of $M_\mu$ is given by the following zero flux condition[24]

$$[M]_{\mu\nu} = \partial_\mu M_\nu - \partial_\nu M_\mu = 0 \,. \tag{6.86}$$

Note that, without extending general relativity any further, this field equation does not follow from a relativistic action and therefore the specific limit we are considering can only be taken at the level of the equations of motion, i.e. the Einstein equations.

Given the extended general relativity theory, we consider the following redefinitions [124]:

$$E_\mu{}^0 = \omega\tau_\mu + \frac{\alpha+1}{\omega}m_\mu \,, \qquad E_\mu{}^a = e_\mu{}^a \,, \qquad M_\mu = \omega\tau_\mu + \frac{\alpha}{\omega}m_\mu \,, \tag{6.87}$$

where $\alpha$ is a real parameter related to the following field redefinition:

$$\tau_\mu \to \tau_\mu + \frac{\alpha}{\omega^2}m_\mu \,. \tag{6.88}$$

From now on, we will take $\alpha = 0$:

$$E_\mu{}^0 = \omega\tau_\mu + \frac{1}{\omega}m_\mu \,, \qquad E_\mu{}^a = e_\mu{}^a \,, \qquad M_\mu = \omega\tau_\mu \,. \tag{6.89}$$

Note that the relativistic inverse Vielbeine are redefined as follows

$$E_0^\mu = \frac{1}{\omega}\tau^\mu + \cdots \,, \qquad\qquad E^\mu{}_a = e^\mu{}_a + \cdots \,, \tag{6.90}$$

where the NR inverse Vielbeine $\tau^\mu$ and $e^\mu{}_a$ were defined in equation (6.52). We have only given here the leading order redefinitions. The lower order dotted terms in (6.90) do not contribute to the final answer when taking the NR limit.

As a simple example of how the limit works we consider the following Lagrangian describing a relativistic particle of mass $M$:

$$S = -M \int d\tau \sqrt{-E_\mu{}^A \dot{X}^\mu E_\nu{}^B \dot{X}^\nu \eta_{AB}} - M_\mu \dot{X}^\mu \,. \tag{6.91}$$

---

[23]We note that in a Post-Newtonian approximation of general relativity there is no need to add this extra gauge field $M_\mu$ since the lowest order terms in such an approximation do not need to constitute an invertible field redefinition.

[24]Here and in the following we will indicate the equation of motion of a field with square brackets.

The last term represents a coupling of the gauge field $M_\mu$ to the particle via a Wess-Zumino term. Substituting the field redefinitions (6.89) into this action and redefining the mass $M$ with $M = \omega\mathfrak{m}$, we obtain, after taking the limit $\omega \to \infty$ and expanding the square root, the following Lagrangian describing the coupling of a non-relativistic particle of mass $\mathfrak{m}$ with embedding coordinates $X^\mu(\tau)$ to a NC background [125]:

$$S = \frac{\mathfrak{m}}{2} \int d\tau \left\{ \frac{e_\mu{}^a \dot{X}^\mu e_\nu{}^b \dot{X}^\nu \delta_{ab}}{\tau_\rho \dot{X}^\rho} - 2\mathfrak{m}_\mu \dot{X}^\mu \right\}. \tag{6.92}$$

One can show that this action, due to the second term, is invariant under Galilean boost transformations. For a flat spacetime with $\tau_\mu = \delta_{\mu,0}$, $e_\mu{}^a = \delta_\mu{}^a = \delta_\mu{}^i$ and $\mathfrak{m}_\mu = \partial_\mu c$ the action reads

$$S_{\text{flat spacetime}} = \frac{\mathfrak{m}}{2} \int d\tau \left\{ \frac{\dot{X}^i \dot{X}^j \delta_{ij}}{\dot{t}} - 2\dot{c} \right\}. \tag{6.93}$$

which describes the coupling of a massive particle to a Newton potential $\Phi$, which is in accordance with equation (5.38) for $R \to \infty$.

We now consider the relativistic Einstein equations

$$\delta S = -\frac{1}{8\pi G_N} \int d^{d+1}x \, E \, \delta E_\mu{}^A E^{\mu B} [G]_{AB}, \tag{6.94}$$

with

$$[G]_{AB} = R_{AB}(\Omega) - \frac{1}{2}\eta_{AB} R(\Omega) = 0. \tag{6.95}$$

This field equation is symmetric in the $A$ and $B$ since the Ricci tensor is symmetric [25]

$$R_{AB}(\Omega) = R_{BA}(\Omega). \tag{6.96}$$

Performing the field redefinitions (6.89) and (6.90) we find

$$\Omega_\mu{}^{ab} = \omega^2 \overset{(2)}{\omega}_\mu{}^{ab} + \overset{(0)}{\omega}_\mu{}^{ab} + \cdots, \tag{6.97a}$$

$$\Omega_\mu{}^{0a} = \omega \overset{(1)}{\omega}_\mu{}^{a} + \omega^{-1}\overset{(-1)}{\omega}_\mu{}^{a} + \cdots, \tag{6.97b}$$

where the $\omega'$s denote expansion coefficients of the relativistic spin-connection fields $\Omega$. The special expansion coefficients $\overset{(0)}{\omega}_\mu{}^{ab}$ and $\overset{(-1)}{\omega}_\mu{}^{a}$ will serve as spin-connection fields in the non-relativistic case and will be denoted by

$$\overset{(2)}{\omega}_\mu{}^{ab} = \omega_\mu{}^{ab}, \qquad\qquad \overset{(-1)}{\omega}_\mu{}^{a} = \omega_\mu{}^{a}. \tag{6.98}$$

We find that the different expansion coefficients are given by

$$\overset{(2)}{\omega}_\mu{}^{ab} = -\tau_\mu T^{ab}, \tag{6.99a}$$

$$\omega_\mu{}^{ab} = -2e_\mu{}^{ab} + e_{\mu c}e^{abc} - \tau_\mu \mathfrak{m}^{ab}, \tag{6.99b}$$

$$\overset{(1)}{\omega}_\mu{}^{a} = e_{\mu b}T^{ba} - 2\tau_\mu T^{a0}, \tag{6.99c}$$

$$\omega_\mu{}^{a} = e_{\mu 0}{}^{a} - e_{\mu c}e_0{}^{ac} + \mathfrak{m}_\mu{}^{a} - \tau_\mu \mathfrak{m}^{a0}. \tag{6.99d}$$

Here we have defined

$$e_{\mu\nu}{}^a = \partial_{[\mu}e_{\nu]}{}^a, \qquad \mathfrak{m}_{\mu\nu} = \partial_{[\mu}\mathfrak{m}_{\nu]}. \tag{6.100}$$

Like before, we have used the inverse NR Vielbeine to convert curved indices into flat indices.

We now substitute the expansions (6.97) of the relativistic spin-connection fields into the Einstein equations (6.95) and the expansion (6.89) of the relativistic vector field $M_\mu$ into the additional equation of motion (6.86). The leading terms in the expressions for the relativistic spin-connection fields are all proportional to the torsion tensor $T_{\mu\nu}$. Upon inserting these terms into the Einstein equations will lead to leading order and sub-leading order terms that are also proportional to $T_{\mu\nu}$. On the other hand, looking at the leading order term of the additional equation of motion (6.86) we already conclude that the torsion is zero: $T_{\mu\nu} = 0$. Substituting this zero torsion constraint into the expanded Einstein equations, we find that the leading order terms of these equations are not anymore given by the terms proportional to the vanishing torsion but instead by terms that involve the NR fields $\omega_\mu{}^{ab}$ and $\omega_\mu{}^{a}$. We thus find that the

---

[25]This follows from inserting into the Bianchi identity for the curvature corresponding to the spacetime translation generator the (conventional) constraint that this curvature is zero.

different components of the relativistic Enstein tensor $[G]_{AB}$ give rise to the following NC equations of motion:

$$[G]_{00} : \quad R_{0a}{}^a(G) = 0 \,, \tag{6.101}$$

$$[G]_{0a} : \quad R_{0ca}{}^c(J) = 0 \,, \tag{6.102}$$

$$[G]_{ab} : \quad R_{acb}{}^c(J) = 0 \,, \tag{6.103}$$

where the curvatures for the Galilean boosts and spatial rotations have been defined in the previous subsection. To derive these equations of motion, we have made use of the identity

$$R_{ab}{}^b(G) = R_{0b}{}^b{}_a(J) \,, \tag{6.104}$$

which follows from taking the NR limit of the relativistic identity (6.96).

In a flat Newtonian spacetime we have

$$\tau_\mu = \delta_\mu{}^0 \,, \qquad e_\mu{}^a = \delta_\mu{}^a \,, \qquad m_\mu = \tau_\mu \, \Phi \,, \tag{6.105}$$

where $\Phi$ is the (time-independent) Newton potential. The only non-trivial spin-connection field for this special case is given by $\omega_0{}^a = \partial^a \Phi$ and the only non-trivial equation of motion reads

$$R_{0a}{}^a(G) = \partial_a \omega_0{}^a = \partial_a \partial^a \Phi = 0 \,, \tag{6.106}$$

thus recovering the well-known sourceless Laplace's equation for the Newton potential. The restrictions (6.105) can be seen as gauge-fixing conditions for the diffeomorphisms restricting to frames with constant acceleration only. The NC equations (6.101)-(6.103) can then be viewed as the extension of the Laplace equation (6.106) to arbitrary frames.

**Carroll gravity** Finally, we consider the case of Carroll gravity. [26] We will derive an invariant action for Carroll gravity by taking the ultra-relativistic limit of the Einstein-Hilbert action. To define this limit, we redefine the gauge fields and symmetry parameters with a dimensionless parameter $\omega$ as follows [49]: [27]

$$E_\mu^0 = \omega^{-1}\tau_\mu \,, \quad \Omega_\mu^{0a} = \omega^{-1}\omega_\mu^a \,, \tag{6.107}$$

$$E_\mu^a = e_\mu^a \,, \quad \Omega_\mu^{ab} = \omega_\mu^{ab} \,, \tag{6.108}$$

$$\Lambda^{0a} = \omega^{-1}\lambda^a \,, \quad \Lambda^{ab} = \lambda^{ab} \,. \tag{6.109}$$

Substituting the field redefinitions (6.107) and (6.108) into the Einstein-Hilbert action, redefining Newton's constant as $G_N = \omega^{-1}G_C$ and taking the $\omega \to \infty$ limit, we end up with the following Carroll action[28]

$$S_{\text{Car}} = -\frac{1}{16\pi G_C} \int e \left( 2\tau^\mu e_a^\nu R(G)_{\mu\nu}{}^a + e_a^\mu e_b^\nu R(J)_{\mu\nu}{}^{ab} \right) \,. \tag{6.110}$$

Here $e = \det(\tau_\mu, e_\mu{}^a)$. The projective inverses $\tau^\mu$ and $e^\mu{}_a$ transform under boosts and spatial rotations as follows:

$$\delta\tau^\mu = 0 \,, \qquad \delta e_a^\mu = -\lambda^a \tau^\mu + \lambda^{ab} e_b^\mu \,. \tag{6.111}$$

The field equations corresponding to the first-order Carroll action (6.110) can be used to solve for the spin connections

$$\omega_\mu{}^a = \tau_\mu \tau^\nu e^{\rho a} \partial_{[\nu}\tau_{\rho]} + e^{\nu a}\partial_{[\mu}\tau_{\nu]} + S^{ab}e_\mu^b \,, \tag{6.112}$$

$$\omega_\mu{}^{ab} = -2e^{\rho[a}\partial_{[\mu}e_{\rho]}^{b]} + e_{\mu c}e^{\rho a}e^{\nu b}\partial_{[\rho}e_{\nu]}^c \,, \tag{6.113}$$

except for a symmetric component $S^{ab} = S^{(ab)} = e^{\mu(a}\omega_\mu^{b)}$ of the boost spin connection $\omega_\mu{}^a$ which remains undetermined.

Plugging the dependent expressions for the spin connections (6.112) and (6.113) into the Carroll action (6.110) we obtain

$$S_{\text{Car}} = -\frac{1}{16\pi G_C} \int e \left( 2\tau^\mu e_a^\nu R(G)^a{}_{\mu\nu}|_{S^{ab}=0} + e_a^\mu e_b^\nu R(J)^{ab}{}_{\mu\nu} + 2K_{ab}S^{ab} - 2\delta^{ab}\delta_{cd}K_{ab}S^{cd} \right) \,. \tag{6.114}$$

From this expression of the action it follows that the equation of motion for $S^{ab}$ implies that $K_{ab} = 0$. In other words, we conclude that $S^{ab}$ is actually a Lagrange multiplier that enforces the intrinsic torsion

---

[26]For other recent work on Carroll gravity, see [51–54].

[27]For a different approach to Carroll gravity, see [78].

[28]This limit shows similarities with the strong coupling limit considered in [120], [121]. Note that both limits lead to a theory with a Carroll-invariant vacuum solution. This suggests that, although looking different at first sight, the result of the two limits might be the same up to field redefinitions.

constraint $K_{ab} = 0$ with $K_{ab}$ defined in equation (6.73). This corresponds to the totally geodesic Carroll structure mentioned in section 4.3.4.

Finally, we note that in $d = 2$ the Carroll algebra can be equipped with a non-degenerate, invariant bilinear form and as a consequence it is possible to write down a Chern-Simons action for the Carroll algebra. This Chern-Simons action is precisely the same as the action given above for $d = 2$.

## 7. Field Theories

In this section we will discuss the non-Lorentzian (NL) field theories for a complex and real massive spin-0 particle, a massive spin-1/2 particle and a massless spin-1 particle. [29] There are two approaches here. Either one takes the NL limit of the relativistic field theory in a flat Minkowski background and after taking the limit one couples the theory to NL gravity or one first couples the model under consideration to general relativity and next takes the NL limit of the matter coupled to gravity system using the NL limits we derived in the previous section. We will opt for this second option. In particular, for spin-0, we will discuss the Galilei, Bargmann and Carroll limits while for spin-1/2 and spin-1 we will only discuss the Bargmann limit.

7.1. **Real Massive Spin-0.** We first discuss the Galilei and Carroll limits of a real massive scalar field. This leads to the following four cases:

**spin-0 Galilei.** We consider the following Lagrangian for a real scalar field:

$$E^{-1} \mathcal{L}_{\rm rel} = +\frac{1}{2} E^{\mu 0} E^{\nu 0} \partial_\mu \Phi \partial_\nu \Phi - \frac{1}{2} E^{\mu a} E^{\nu b} \delta_{ab} \partial_\mu \Phi \partial_\nu \Phi - \frac{1}{2} \epsilon M^2 \Phi^2 \,, \tag{7.1}$$

where $\epsilon = +1$ and $\epsilon = -1$ corresponds to a massive particle and a tachyon, respectively. Performing the Galilei redefinitions (6.74) and (6.75), we obtain

$$e^{-1} \mathcal{L}_{\rm rel} = +\frac{1}{2\omega^2} \tau^\mu \tau^\nu \partial_\mu \Phi \partial_\nu \Phi - \frac{1}{2} e^{\mu a} e^{\nu b} \delta_{ab} \partial_\mu \Phi \partial_\nu \Phi - \frac{1}{2} \epsilon M^2 \Phi^2 \,, \tag{7.2}$$

where $e = \det(\tau_\mu, e_\mu{}^a)$ and where we have ignored an overall power of $\omega$. [30] There are now two ways to proceed. First, by choosing $\epsilon = -1$ and taking the limit $\omega \to \infty$, we obtain the following 'magnetic' Galilei Lagrangian:

$$e^{-1} \mathcal{L}_{\text{magnetic Galilei}} = -\frac{1}{2} e^{\mu a} e^{\nu b} \delta_{ab} \partial_\mu \Phi \partial_\nu \Phi + \frac{1}{2} M^2 \Phi^2 \,. \tag{7.3}$$

The flat spacetime Lagrangian and the corresponding transformation rules are given by

$$\mathcal{L}_{\text{magnetic Galilei}}(\textsf{flat spacetime}) = -\frac{1}{2} (\partial_i \Phi)^2 + \frac{1}{2} M^2 \Phi^2 \tag{7.4}$$

and

$$\delta \Phi = \left( \zeta \, \partial_t + \xi^i \partial_i - \lambda^i \, t \, \partial_i - x^j \lambda^i{}_j \, \partial_i \right) \Phi \,. \tag{7.5}$$

This limit was considered in the context of taking the limit of a tachyonic particle Lagrangian [111] where it leads to the massless Galilei particle of Souriau with 'colour' M [43].

A second option is to first redefine $\Phi = \omega \phi, M = \omega^{-1} m$ and obtain the following Lagrangian:

$$e^{-1} \mathcal{L}_{\rm rel} = +\frac{1}{2} \tau^\mu \tau^\nu \partial_\mu \phi \partial_\nu \phi - \frac{1}{2} \omega^2 e^{\mu a} e^{\nu b} \delta_{ab} \partial_\mu \phi \partial_\nu \phi - \frac{1}{2} \epsilon m^2 \phi^2 \,, \tag{7.6}$$

To deal with the quadratic divergence in the second term, we use a result of [31] and rewrite the Lagrangian, introducing auxiliary fields $\chi^a$, as follows: [31]

$$e^{-1} \mathcal{L}_{\rm rel} = +\frac{1}{2} \tau^\mu \tau^\nu \partial_\mu \phi \partial_\nu \phi + \frac{1}{2\omega^2} \chi^a \chi_a + \chi^a e^\mu{}_a \partial_\mu \phi - \frac{1}{2} \epsilon m^2 \phi^2 \,. \tag{7.7}$$

Next, choosing $\epsilon = +1$ and taking the limit $\omega \to \infty$ the auxiliary fields $\chi^a$ become Lagrange multipliers and we obtain the following Lagrangian:

$$e^{-1} \mathcal{L}_{\text{electric Galilei}} = +\frac{1}{2} \tau^\mu \tau^\nu \partial_\mu \phi \partial_\nu \phi + \chi^a e^\mu{}_a \partial_\mu \phi - \frac{1}{2} m^2 \phi^2 \tag{7.8}$$

---

[29]There is a huge literature on field theories with Galilean and Carrollian symmetries, see e.g. [126, 127] for some early references.

[30]Such an overall power can be cancelled by a further redefinition of the fields.

[31]The general expression is that for each $X$, the quadratic divergence $\omega^2 X^2$ can be rewritten, introducing an auxiliary field $\chi$, as $-\frac{1}{\omega^2} \chi^2 - 2\chi X$.

The flat spacetime Lagrangian and the corresponding transformation rules are given by

$$\mathcal{L}_{\text{elctric Galilei}}(\texttt{flat spacetime}) = +\frac{1}{2}(\partial_t \phi)^2 + \chi^i \partial_i \phi - \frac{1}{2} m^2 \phi^2 \tag{7.9}$$

and

$$\delta\phi = \left( \zeta\,\partial_t + \xi^i \partial_i - \lambda^i\, t\, \partial_i - x^j \lambda^i{}_j\, \partial_i \right)\phi, \qquad \delta\chi^i = \left( \zeta\,\partial_t + \xi^j \partial_j - \lambda^j\, t\, \partial_j - x^j \lambda^k{}_j\, \partial_k \right)\chi^i + \lambda^i(\partial_t \phi). \tag{7.10}$$

**spin-0 Carroll.** This case was recently considered in [98] in connection with dark matter and inflation and in [51] using field theory in an Hamiltonian formulation. The two types of Carroll limits considered here have also been considered in the context of $p$-brane sigma models using a Lagrangian formulation [128]. The fact that there are two types of Carroll limits also follows from the duality between the Galilei and Carrol symmetries considered in [33].

We consider the same Lagrangian for a real scalar field as in the Galilei case:

$$E^{-1}\mathcal{L}_{\text{rel}} = +\frac{1}{2}E^{\mu 0}E^{\nu 0}\partial_\mu\Phi\partial_\nu\Phi - \frac{1}{2}E^{\mu a}E^{\nu b}\delta_{ab}\partial_\mu\Phi\partial_\nu\Phi - \frac{1}{2}\epsilon M^2\Phi^2. \tag{7.11}$$

but now perform the Carroll redefinitions (6.107) and (6.108). In this way we obtain

$$e^{-1}\mathcal{L}_{\text{rel}} = +\frac{1}{2}\omega^2\tau^\mu\tau^\nu\partial_\mu\Phi\partial_\nu\Phi - \frac{1}{2}e^{\mu a}e^{\nu b}\delta_{ab}\partial_\mu\Phi\partial_\nu\Phi - \frac{1}{2}\epsilon M^2\Phi^2, \tag{7.12}$$

where $e = \det(\tau_\mu, e_\mu{}^a)$ and where we have ignored an overall power of $\omega$.

Like in the Galilei case, there are now two options to proceed. First, to deal with the quadratic divergence in the first term, we rewrite the Lagrangian introducing an auxiliary field $\chi$ as follows:

$$e^{-1}\mathcal{L}_{\text{rel}} = -\frac{1}{2}\frac{1}{\omega^2}\chi^2 - \chi\tau^\mu\partial_\mu\Phi - \frac{1}{2}e^{\mu a}e^{\nu b}\delta_{ab}\partial_\mu\Phi\partial_\nu\Phi - \frac{1}{2}\epsilon M^2\Phi^2. \tag{7.13}$$

Next, choosing $\epsilon = -1$ and taking the limit $\omega \to \infty$, we see that $\chi$ has become a Lagrange multiplier and we obtain the following magnetic Carroll Lagrangian:

$$e^{-1}\mathcal{L}_{\text{magnetic Carroll}} = -\chi\tau^\mu\partial_\mu\Phi - \frac{1}{2}e^{\mu a}e^{\nu b}\delta_{ab}\partial_\mu\Phi\partial_\nu\Phi + \frac{1}{2}M^2\Phi^2. \tag{7.14}$$

The flat spacetime Lagrangian and the corresponding transformation rules are given by

$$\mathcal{L}_{\text{magnetic Carroll}}(\texttt{flat spacetime}) = -\chi(\partial_t\Phi) - \frac{1}{2}(\partial_i\Phi)^2 + \frac{1}{2}M^2\Phi^2. \tag{7.15}$$

and

$$\delta\Phi = \left( \zeta\,\partial_t + \xi^i \partial_i - \lambda^i x_i \partial_t - x^j \lambda^i{}_j\, \partial_i \right)\Phi, \qquad \delta\chi = \left( \zeta\,\partial_t + \xi^i \partial_i - \lambda^i x_i \partial_t - x^j \lambda^i{}_j\, \partial_i \right)\chi + \lambda^i(\partial_i\Phi). \tag{7.16}$$

A second option is to first redefine $\Phi = \frac{1}{\omega}\phi, M = \omega m$. We then choose $\epsilon = +1$ and take the limit $\omega \to \infty$ after which we obtain the following Lagrangian [97]:

$$e^{-1}\mathcal{L}_{\text{electric Carroll}} = +\frac{1}{2}\tau^\mu\tau^\nu\partial_\mu\phi\partial_\nu\phi - \frac{1}{2}m^2\phi^2. \tag{7.17}$$

The flat spacetime Lagrangian and the corresponding transformation rules are given by [97]

$$e^{-1}\mathcal{L}_{\text{electric Carroll}}(\texttt{flat spacetime}) = +\frac{1}{2}(\partial_t\phi)^2 - \frac{1}{2}m^2\phi^2. \tag{7.18}$$

and

$$\delta\phi = \left( \zeta\,\partial_t + \xi^i \partial_i - \lambda^i x_i \partial_t - x^j \lambda^i{}_j\, \partial_i \right)\phi. \tag{7.19}$$

This concludes our discussion of the four limits of a real massive spin-0 particle.

## 7.2. Complex Massive Spin-0.
Following [124], we now discuss the standard Bargmann limit of a complex Klein-Gordon scalar field in a curved background. In contrast to the real scalar discussed above the introduction of an extra vector gauge field has the effect that the quadratic divergences cancel and there is only one way to take the limit.

Our starting point is a Lagrangian for a relativistic massive complex scalar $\Phi$, with mass $M$, minimally coupled to an arbitrary gravitational background and the extra zero-flux $U(1)$ gauge field $M_\mu$ that we introduced in the previous section:

$$E^{-1}\mathcal{L}_{\text{rel}} = -\frac{1}{2}g^{\mu\nu}D_\mu\Phi^* D_\nu\Phi - \frac{M^2}{2}|\Phi|^2. \tag{7.20}$$

Here the covariant derivative is given by

$$D_\mu\Phi = \partial_\mu\Phi - i\,M\,M_\mu\,\Phi. \tag{7.21}$$

Apart from invariance under diffeomorphisms, the above Lagrangian is also invariant under a local U(1) symmetry given by the transformation rule

$$\delta\Phi = i\,M\,\Lambda\,\Phi\,. \tag{7.22}$$

The conserved current associated to this local U(1) symmetry, which is given by

$$j^{\mu}_{\mathrm{rel}} = \frac{M}{2i}\left(\Phi^* D^{\mu}\Phi - \Phi D^{\mu}\Phi^*\right)\,, \tag{7.23}$$

expresses conservation of the number of particles minus the number of antiparticles.

Using the redefinitions of the previous section and redefining the mass parameter $M$ as

$$M = \omega\mathfrak{m}\,, \tag{7.24}$$

one finds that the $O(\omega^2)$ contribution to the Lagrangian cancels with one contribution coming from the mass term and another one from the term that is quadratic in the U(1) gauge field. Therefore, the $\omega \to \infty$ limit is well-defined and leads to the following Lagrangian for a Schrödinger field coupled to an arbitrary Newton-Cartan background: [32] [33]

$$e^{-1}\mathcal{L}_{\mathrm{non-rel}} = \mathfrak{m}\left[\frac{i}{2}\left(\Phi^*\tilde{D}_0\Phi - \Phi\tilde{D}_0\Phi^*\right) - \frac{1}{2\mathfrak{m}}\left|\tilde{D}_a\Phi\right|^2\right]\,, \tag{7.25}$$

where we have defined

$$\tilde{D}_{\mu}\Phi = \partial_{\mu}\Phi + i\,\mathfrak{m}\,m_{\mu}\,\Phi\,. \tag{7.26}$$

The Lagrangian (7.25) is invariant under diffeomorphisms (with parameter $\xi^{\mu}$) and the local U(1) central charge transformation of the Bargmann algebra (with parameter $\sigma$), under which $\Phi$ transforms as

$$\delta\Phi = \xi^{\mu}\partial_{\mu}\Phi - i\,\mathfrak{m}\,\sigma\,\Phi\,. \tag{7.27}$$

One can then define the current associated to the central charge transformation by

$$j^{\mu}_{\mathrm{non-rel}} = \tau^{\mu}\,|\Phi|^2 + e^{\mu}{}_a\,\frac{1}{2\mathfrak{m}i}\left(\Phi^*\tilde{D}^a\Phi - \Phi\tilde{D}^a\Phi^*\right)\,. \tag{7.28}$$

When choosing a flat background

$$\tau_{\mu} = \delta^t_{\mu}\,, \qquad e_t{}^a = 0\,,\quad e_i{}^a = \delta_i^a\,, \qquad m_{\mu} = 0\,, \tag{7.29}$$

this current corresponds to the usual current of particle number or mass conservation. We thus explicitly see that, as expected for a non-relativistic limit, our NR limit procedure has suppressed antiparticles.

It is instructive to look at the action on $\Phi$ of the symmetries that are left when the flat background (7.29) is chosen. The transformation rules (7.27) then reduce to those that leave these flat background fields invariant. They are determined by the following NR Killing equations

$$\begin{aligned}
\partial_{\mu}\xi^t &= 0\,, & \partial_t\xi^i + \lambda^i &= 0\,, \\
\partial_i\xi^j + \lambda^j{}_i &= 0\,, & \partial_t\sigma &= 0\,, & \partial_i\sigma + \lambda_i &= 0\,.
\end{aligned} \tag{7.30}$$

The solution to these equations is given by

$$\xi^t(x^{\mu}) = \zeta\,, \qquad \xi^i(x^{\mu}) = \xi^i - \lambda^i\,t - \lambda^i{}_j\,x^j\,, \qquad \sigma(x^{\mu}) = \sigma - \lambda^i\,x^i\,, \tag{7.31}$$

where the parameters $\zeta$, $\xi^i$, $\lambda^i$, $\lambda^{ij}$, $\sigma$ are now constants. These correspond to the usual time translation, spatial translations, Galilean boosts, spatial rotations and central charge transformation of the rigid Bargmann algebra. One thus finds that $\Phi$ transforms as follows:

$$\delta\Phi = \left(\zeta\,\partial_t + \xi^i\,\partial_i - \lambda^i\,t\,\partial_i - x^j\lambda^i{}_j\,\partial_i - i\,\mathfrak{m}\,\sigma + i\,\mathfrak{m}\,\lambda^i\,x^i\right)\Phi\,. \tag{7.32}$$

The last term in this transformation rule corresponds to the phase factor acquired by a Schrödinger field under rigid Galilean boosts, that is necessary to show Galilei invariance of the flat space Schrödinger Lagrangian. We note that this same Schrödinger Lagrangian is also invariant under an extra dilatation and special conformal transformation, that extend the symmetries of the Bargmann algebra denoted in (7.32) to the ones of the Schrödinger algebra [129]. So, even though we started from a relativistic theory with no conformal invariance, we end up with a NR theory that is invariant under non-relativistic conformal Schrödinger symmetries.

---

[32]We have ignored an overall factor of $\omega$ coming from the redefinition of $E = \omega\,e + O(\omega^{-1})$. This factor is irrelevant as it amounts to an overall rescaling of the Lagrangian that could be compensated by a redefinition of the scalar field.

[33]We have turned curved indices into flat indices using the inverse Newton-Cartan Vierbeine. Thus $\tilde{D}_0$, $\tilde{D}_a$ are shorthand for $\tau^{\mu}\tilde{D}_{\mu}$, $e^{\mu}{}_a\tilde{D}_{\mu}$. Spatial flat indices are raised and lowered with a Kronecker delta.

7.3. **Massive Spin-1/2.** Following [124], our starting point is the Dirac Lagrangian for a $d = 3$ massive spin $1/2$ particle described by a 4-component spinor $\Psi$ coupled to an arbitrary gravitational background and the $U(1)$ gauge field $M_\mu$:

$$E^{-1}\mathcal{L}_{\text{rel}} = \bar{\Psi}\slashed{D}\Psi - M\,\bar{\Psi}\Psi + \text{h.c.}\,, \tag{7.33}$$

where the covariant derivative is given by

$$D_\mu\Psi = \partial_\mu\Psi - \frac{1}{4}\,\Omega_\mu{}^{AB}\gamma_{AB}\Psi + i\,M\,M_\mu\,\Psi\,. \tag{7.34}$$

Before taking the limit, it is convenient to define projected spinors in terms of the original spinor as follows [130]:

$$\Psi_\pm = \frac{1}{2}\left(\mathbb{1} \pm i\gamma^0\right)\Psi\,, \qquad\qquad \bar{\Psi}_\pm = \bar{\Psi}\,\frac{1}{2}\left(\mathbb{1} \pm i\gamma^0\right). \tag{7.35}$$

Besides redefining the bosonic gravitational fields we also redefine these projected spinors as follows [130]:

$$\Psi_+ = \sqrt{\omega}\,\psi_+\,, \qquad\qquad \Psi_- = \frac{1}{\sqrt{\omega}}\,\psi_-\,. \tag{7.36}$$

Using all these redefinitions and taking $M = \omega m$, one finds that the action (7.33) upon taking the $\omega \to \infty$ limit reduces to

$$e^{-1}\mathcal{L}_{\text{non-rel}} = \bar{\psi}_+\gamma^0\tilde{D}_0\psi_+ + \bar{\psi}_+\gamma^a\tilde{D}_a\psi_- + \bar{\psi}_-\gamma^a\tilde{D}_a\psi_+ - 2\,m\,\bar{\psi}_-\psi_- + \text{h.c.}\,, \tag{7.37}$$

where we have used the covariant derivatives

$$\begin{aligned}
\tilde{D}_\mu\psi_+ &= \partial_\mu\psi_+ - \frac{1}{4}\,\omega_\mu{}^{ab}\gamma_{ab}\psi_+ - i\,m\,m_\mu\,\psi_+\,, \\
\tilde{D}_\mu\psi_- &= \partial_\mu\psi_- - \frac{1}{4}\,\omega_\mu{}^{ab}\gamma_{ab}\psi_- + \frac{1}{2}\,\omega_\mu{}^a\gamma_{a0}\psi_+ - i\,m\,m_\mu\,\psi_-\,.
\end{aligned} \tag{7.38}$$

Note that all divergences have canceled. The invariance of the Lagrangian (7.37) under Galilean boosts is not manifest but can be checked by using the transformation rules

$$\begin{aligned}
\delta\psi_+ &= \frac{1}{4}\,\lambda^{ab}\gamma_{ab}\psi_+ + i\,m\,\sigma\psi_+\,, \\
\delta\psi_- &= \frac{1}{4}\,\lambda^{ab}\gamma_{ab}\psi_- - \frac{1}{2}\,\lambda^a\gamma_{a0}\psi_+ + i\,m\,\sigma\psi_-\,,
\end{aligned} \tag{7.39}$$

that are easily found by applying all field redefinitions in the relativistic transformation rules and taking the limit $\omega \to \infty$.

The equations of motion corresponding to the non-relativistic Lagrangian (7.37) are given by the Lévy–Leblond equations

$$\begin{aligned}
\gamma^0\tilde{D}_0\psi_+ + \gamma^a\tilde{D}_a\psi_- &= 0\,, \\
\gamma^a\tilde{D}_a\psi_+ - 2\,m\,\psi_- &= 0\,.
\end{aligned} \tag{7.40}$$

The second equation can be used to solve for the auxiliary spinor $\psi_-$ and eliminate it from the Lagrangian (7.37). Substituting the solution for $\psi_-$ back into the first equation we obtain the curved space generalization of the so-called Schrödinger–Pauli equation:

$$\left[\gamma^0\tilde{D}_0 + \frac{1}{2m}\tilde{D}^a\tilde{D}_a\right]\psi_+ = 0\,. \tag{7.41}$$

7.4. **NR massless spin 1.** We now consider the massless spin 1 case. Our starting point is the Lagrangian of a real, massless, relativistic vector field coupled to gravity:

$$E^{-1}\mathcal{L}_{\text{rel}} = -\frac{1}{4}\,E_A^\mu E^{\rho A}E_B^\nu E^{\sigma B}F_{\mu\nu}F_{\rho\sigma}\,, \tag{7.42}$$

where $F_{\mu\nu}$ is the usual Maxwell field strength. Like for the spin 0 case, one can take an electric or magnetic Galilean limit of electrodynamics or even an electric and magnetic Carrollian limit. We will only discuss here the magnetic Galilean limit since it shows the additional feature of an emergent symmetry, something that does not occur in the spin-0 case. Redefining the gravitational background fields like in the previous section leads to the following non-relativistic Lagrangian

$$e^{-1}\mathcal{L}_{\text{rel}} = -\frac{1}{2\omega^2}\tau^\mu\tau^\nu F_{\mu a}F_\nu{}^a - \frac{1}{4}\,F_{ab}F^{ab}\,. \tag{7.43}$$

Taking the limit $\omega \to \infty$, we obtain for a flat spacetime

$$\mathcal{L}_{\text{non-rel}} = -\frac{1}{4}\,F_{ij}F^{ij}\,. \tag{7.44}$$

Due to the absence of the field $A_0$ this Lagrangian has an emergent Stueckelberg symmetry $\delta A_0(x) = \rho(x)$ while the corresponding field equation of $A_0$ does not follow directly from the non-relativistic Lagrangian (7.44). The situation is very similar to what happens when taking the limit of Neveu-Schwarz gravity where the Poisson equation of the Newton potential is missing [131]. The missing equation of motion can be obtained by taking the limit of the relativistic equations of motion. The complete set of non-relativistic equations of motion form a reducible but indecomposable representation under Galilean boosts which means that the equation of motion corresponding to $A_0$ transforms to the equations of motion corresponding to $A_i$ but not the other way around. This shows the following connection between the equation of motion corresponding to $A_0$ and the Lagrangian (7.44): the missing equation of motion corresponding to $A_0$ is not invariant under Galilean boosts by itself but, instead transforms into the equations of motion that follow from the non-relativistic Lagrangian (7.44).

### 7.5. Massless Spin 1 with an additional scalar field.

Allowing the option to add extra fields to the Lagrangian, there is yet another way to obtain a non-relativistic Lagrangian from a relativistic one. To be precise, extending the relativistic Lagrangian with a massless scalar $\rho$ we consider the following Lagrangian [124]:

$$E^{-1}\mathcal{L}_{\rm rel} = -\frac{1}{4} E_A^\mu E^{\rho A} E_B^\gamma E^{\sigma B} F_{\mu\nu} F_{\rho\sigma} - \frac{1}{2} E_A^\mu E^{\nu A} \partial_\mu \rho \, \partial_\nu \rho \,. \tag{7.45}$$

Defining two fields A and B as follows:

$$A = E^\mu{}_0 A_\mu - \rho \,, \qquad B = E^\mu{}_0 A_\mu + \rho \,, \tag{7.46}$$

one can redefine the bosonic background fields like in the previous section supplemented with the redefinitions

$$A = \frac{1}{\omega} \tilde{A} \,, \qquad B = \omega \tilde{B} \,, \tag{7.47}$$

to obtain the following non-relativistic Lagrangian in the $\omega \to \infty$ limit [34]

$$e^{-1}\mathcal{L}_{\rm non-rel} = \frac{1}{8} \partial_0 \tilde{B} \, \partial_0 \tilde{B} + \frac{1}{2} \tilde{D}_a \tilde{A} \, \partial^a \tilde{B} - \frac{1}{4} F_{ab} F^{ab} - \frac{1}{2} \tilde{D}^a A_a \, \partial_0 \tilde{B} \,, \tag{7.48}$$

where the following derivatives were used

$$\tilde{D}_\mu \tilde{A} = \partial_\mu \tilde{A} + \omega_\mu{}^a A_a \,,$$

$$\tilde{D}_\mu A_a = \partial_\mu A_a - \omega_{\mu a}{}^b A_b + \frac{1}{2} \omega_{\mu a} \tilde{B} \,. \tag{7.49}$$

Note that the basic variables are a spatial vector $A_a$ with spatial flat indices and two extra fields $\tilde{A}$, $\tilde{B}$. These fields transform non-trivially under local spatial rotations and Galilean boosts as follows:

$$\delta \tilde{A} = -\lambda^a A_a \,, \qquad\qquad \delta \tilde{B} = 0 \,,$$

$$\delta A_a = \lambda_a{}^b A_b - \frac{1}{2} \lambda^a \tilde{B} \,, \tag{7.50}$$

while they transform as scalars under general coordinate transformations. It is with respect to these transformations that the above derivatives (7.49) are defined. The Lagrangian (7.48) is also invariant under the U(1) gauge transformation

$$\delta \tilde{A} = \tau^\mu \partial_\mu \Lambda \,, \qquad \delta A_a = e^\mu{}_a \partial_\mu \Lambda \,, \tag{7.51}$$

although this invariance is not manifest.

To get a better physical understanding of the Lagrangian (7.48), we consider the equations of motion when restricted to the flat background (7.29) (such that $i = a$):

$$\partial^i \partial_i \tilde{B} = 0 \,,$$

$$\partial_i \partial_t \tilde{B} + \partial^j F_{ji} = 0 \,, \tag{7.52}$$

$$\partial_t \partial_t \tilde{B} - 2 \, \partial^i \partial_i \tilde{A} + 2 \, \partial_t \partial^i A_i = 0 \,.$$

One can consistently set $\tilde{B}$ to zero in these equations since this constraint is invariant under all the symmetries of the theory. The remaining equations for $\tilde{A}$ and $A_i$ then coincide with the equations of Galilean Electromagnetism in the magnetic limit, where $\tilde{A}$ plays the role of the electric potential. This theory is not only invariant under the Galilei group, but also under the Galilean conformal group [129, 134–136]. The latter is the conformal extension of the Galilei group that is obtained by

---

[34] For a flat spacetime, the same Lagrangian can be obtained by a null reduction [132]. A 'T-dual way' to obtain the same Lagrangean is to take a so-called string limit of Maxwell in one dimension higher and to reduce over the spatial direction longitudinal to the string [133].

performing an Inönü-Wigner contraction of the relativistic conformal group. Since the relativistic Lagrangian we started from is conformally invariant when restricted to flat space, it is not surprising to see that the non-relativistic limit is invariant under Galilean conformal transformations.

## 8. Conclusion

In this review we summarized the basic properties of a number of non-Lorentzian theories. We first discussed the kinematical spaces and corresponding symmetry algebras of these non-Lorentzian theories. We next constructed a number of actions describing the dynamics of particles moving in these kinematical spaces. For this, we applied the method of nonlinear realisations and explained the relation between this method and the co-adjoint orbit method. We have also analysed the non-Lorentzian particles as a suitable non-relativistic limit of relativistic particles. We also discussed three types of non-Lorentzian gravity theories: Galilei gravity, Newton-Cartan gravity and Carroll gravity. We not only showed how these gravity theories can be obtained by applying a gauging procedure to an underlying non-relativistic Lie algebra but also by taking a special non-relativistic limit of general relativity. Introducing matter, we discuss the coupling of gravity to field theories describing particles of different spin. We achieved this by starting from the relativistic field theories coupled to general relativity and taking a non-relativistic limit.

There are several ways to extend the results presented in this review some of which are discussed in the other articles in this volume. As mentioned in the introduction, one could extend the degenerate geometries we considered here to geometries that are characterized by a foliation of a higher co-dimension. In particular, the geometries with a co-dimension two foliation play an important role in describing non-relativistic string theory, see the article by Oling and Yan [19].

## Acknowledgments

We acknowledge many enlightening conversations on non-lorentzian topics with the following people: Roberto Casalbuoni, Can Görmez, Ross Grassie, Jelle Hartong, Emil Have, Yannick Herfray, Axel Kleinschmidt, Johannes Landsteiner, Stefan Prohazka, Luca Romano, Jan Rosseel, Jakob Salzer, Dieter Van den Bleeken and Kevin van Helden.

The work of JG has been supported in part by MINECO FPA2016-76005-C2-1-P and PID2019-105614GB-C21 and from the State Agency for Research of the Spanish Ministry of Science and Innovation through the Unit of Excellence Maria de Maeztu 2020-203 award to the Institute of Cosmos Sciences (CEX2019-000918-M).

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

(JG) Departament de Física Cuàntica i Astrofísica and Institut de Ciències del Cosmos, Universitat de Barcelona, Martí i Franquès 1, E-08028 Barcelona, Spain

*Email address*: `gomis[at]ecm.ub.es`