# Peer review of "A non-lorentzian primer"

_SciPost Physics Lecture Notes_

## Round 1 · Referee Report · Anonymous (Referee 1) · 2023-3-7

Report

The paper under consideration is an article which gives a wide mathematical perspective on a subject of significant current relevance. The manuscript considers kinematical and dynamical aspects of theories where the underlying symmetry goes beyond the usually considered Lorentz group. These theories, especially the Galilean and Carrollian ones, have attracted a lot of recent attention due to the discovery of a wide range of applications.

I think the article is excellent, the main highlight being the detailed exposition of the algebraic and geometric structures that underlie non-Lorentzian structures which replace the (pseudo) Riemannian metric structure of the relativistic spacetimes.

This is a rapidly growing field and hence any review can only do so much. If I were to be hyper-critical, I would mention that now there is a (partial?) understanding of fermions in the Carroll limit and this is a point that the authors don't address when considering the spin 1/2 case in their field theory section. But these nitpickings aside, I think this is an article that should be published as is in this journal.

---

## Round 1 · Referee Report · Anonymous (Referee 2) · 2023-3-8

Strengths

1). Clear and concise overview of the basic mathematical notions that are required for a more mathematical approach to the subject of non-Lorentzian geometries and its basic properties such as particle dynamics.

2). The review focusses on the most important examples and is not overly encyclopedic, while at the same time the relevant papers are cited where more examples (or special cases) can be found.

Weaknesses

1). The review assumes a more than average amount of mathematical background knowledge in order to properly digest and use the results. This reduces the accessibility among an audience of predominantly theoretical physicists somewhat. This is of course also a strength in a way because one could argue that papers like this nicely complement the existing literature and open up the subject to more mathematically minded audiences.

Report

The review is rather well-structured and well-written so I have very few comments (see below). I believe the review will provide a useful source for people entering the growing subject of non-Lorentzian geometry and its applications in theoretical physics. I recommend it for publication.

Requested changes

  • item i). on page 2 is called non-relativistic holography, but the references are all in the context of the usual AdS/CFT correspondence so could the authors please clarify what is meant here.

  • Refs [8,9] are independent of flat space holography. These are just the original Carroll papers. Should Carroll not be its own item in the list, surely it features in many more places than flat space holography.

  • The authors refer to the Carroll limit as ultra-relativistic, but ultra-relativistic means v/c going to unity and so this viewpoint is really only correct if you view Carroll as originating from physics on a null hypersurface. If one does not take this higher-dimensional viewpoint it is better described as an ultra-local limit. It would be good if this is addressed in the text.

  • Please clarify the notation used in (4.36) on the LHS of the first equality. I could not make sense of what is written there in an operational sense. The RHS of the second equality misses (\alpha).

  • Below eq. (5.66) it is claimed that the equations displayed in (5.66) are related to the EOM of a magnetic Carroll scalar field theory (eq. (7.15)), but I do not agree with this. The EOM of (7.15) are not the equations in (5.66) so it was not clear to me what the authors meant by this comment. Please elaborate/clarify/modify.

---

## Editorial Decision

resubmitted